# Sample and Communication-Efficient Decentralized Actor-Critic Algorithms with Finite-Time Analysis

## Abstract

Actor-critic (AC) algorithms have been widely adopted in decentralized multi-agent systems to learn the optimal joint control policy. However, existing decentralized AC algorithms either need to share agents' sensitive information, e.g., local actions and policies , or are not sample and communication-efficient. In this work, we develop two decentralized AC and natural AC (NAC) algorithms that are sample and communication-efficient and avoid sharing agents' local actions and policies . In both algorithms, agents share only noisy rewards and adopt mini-batch local policy gradient updates to improve sample and communication efficiency. Particularly for decentralized NAC, we develop a decentralized Markovian SGD algorithm with an adaptive mini-batch size to efficiently compute the natural policy gradient. Under Markovian sampling and linear function approximation, we prove that the proposed decentralized AC and NAC algorithms achieve the state-of-the-art sample complexities $\mathcal{O}(\epsilon^{-2}\ln\epsilon^{-1})$ and $\mathcal{O}(\epsilon^{-3}\ln\epsilon^{-1})$, respectively, and achieve an improved communication complexity $\mathcal{O}(\epsilon^{-1}\ln\epsilon^{-1})$. Numerical experiments demonstrate that the proposed algorithms achieve lower sample and communication complexities than the existing decentralized AC algorithm.

## 1 Introduction

Multi-agent reinforcement learning (MARL) has achieved great success in various application domains, including control (66; 10; 51), robotics (64), wireless sensor networks (24; 67), intelligent systems (71), etc. In MARL, a set of fully decentralized agents interact with a dynamic environment following their own policies and collect local rewards, and their goal is to collaboratively learn the optimal joint policy that achieves the maximum expected accumulated reward.

Classical policy optimization algorithms have been well developed and studied, e.g., policy gradient (PG) (49), actor-critic (AC) (23) and natural actor-critic (NAC) (37; 7). In particular, AC-type algorithms are more computationally tractable and efficient as they take advantages of both policy gradient and value-based updates. However, in the multi-agent setting, decentralized AC is more challenging to design compared with the centralized AC, as the algorithm updates involve sensitive agent information, e.g., local actions, rewards and policies, which must be kept locally in the decentralized learning process. In the existing designs of decentralized AC, the agents need to share either their local actions (70; 69; 8; 36; 72; 27; 19; 26; 11) or local rewards (15; 33; 32) with their neighbors, and hence are not desired. This issue is addressed by Algorithm 2 of (70) at the cost of learning a parameterized model to estimate the averaged reward, yet this approach requires extra learning effort and the reward estimation can be inaccurate. Moreover, existing decentralized AC algorithms are not sample and communication-efficient, and do not have finite-time convergence guarantee, especially under the practical Markovian sampling setting. Therefore, we aim to address the following important question.

- *Q1: Can we develop a decentralized AC algorithm that is convergent, sample and communication-efficient, and does not require sharing agents' local actions and policies ?*

On the other hand, as an important variant of the decentralized AC, decentralized NAC algorithm has not been formally developed and rigorously analyzed in the existing literature. In particular, a major challenge is that we need to develop a fully decentralized and computationally tractable scheme to

compute the inverse of the high dimensional Fisher information matrix, and this scheme must be both sample and communication efficient. Hence, we want to ask:

- *Q2: Can we develop a computationally tractable and communication-efficient decentralized NAC algorithm that has a low finite-time sample and communication complexity?*

In this study, we provide affirmative answers to the above two questions by developing fully decentralized AC and NAC algorithms that are sample and communication-efficient, and do not reveal agents' local actions and policies. We also develop rigorous finite-time analysis of these algorithms under Markovian sampling. Our contributions are summarized as follows.

Table 1: List of complexities of the existing AC and NAC algorithms for achieving $\mathbb{E}[\|\nabla J(\omega)\|^2] \leq \epsilon$ and $\mathbb{E}[J(\omega^*) - J(\omega))] \leq \epsilon$, respectively.

| Algorithm | Papers | Share local action/policy | Sampling scheme | Sample complexity | Communication complexity |
|---|---|---|---|---|---|
| Centralized AC | (54) | – | i.i.d. | $\mathcal{O}(\epsilon^{-36})$ | – |
| | (38) | – | i.i.d. | $\widetilde{\mathcal{O}}(\epsilon^{-4})$ | – |
| | (25) | – | i.i.d. | $\mathcal{O}(\epsilon^{-2.5})$ | – |
| | (61) | – | Markovian | $\mathcal{O}(\epsilon^{-2.5}\ln^3 \epsilon^{-1})$ | – |
| | (56) | – | Markovian | $\widetilde{\mathcal{O}}(\epsilon^{-2.5})$ | – |
| | (60) | – | Markovian | $\mathcal{O}(\epsilon^{-2}\ln \epsilon^{-1})$ | – |
| Decentralized AC | (70; 69; 15) (72; 27; 26) | × | Markovian | – | – |
| | (70; 47; 33) | ✓ | Markovian | – | – |
| | This work | ✓ | Markovian | $\mathcal{O}(\epsilon^{-2}\ln \epsilon^{-1})$ | $\mathcal{O}(\epsilon^{-1}\ln \epsilon^{-1})$ |
| Centralized NAC | (54) | – | i.i.d. | $\mathcal{O}(\epsilon^{-36})$ | – |
| | (61) | – | Markovian | $\mathcal{O}(\epsilon^{-4}\ln^2 \epsilon^{-1})$ | – |
| | (60) | – | Markovian | $\mathcal{O}(\epsilon^{-3}\ln \epsilon^{-1})$ | – |
| Decentralized NAC | This work | ✓ | Markovian | $\mathcal{O}(\epsilon^{-3}\ln \epsilon^{-1})$ | $\mathcal{O}(\epsilon^{-1}\ln \epsilon^{-1})$ |

## 1.1 OUR CONTRIBUTIONS

We develop fully decentralized AC and NAC algorithms and analyze their finite-time sample and communication complexities under Markovian sampling. Our results and comparisons to existing works are summarized in Table 1 [1]. Our decentralized AC and NAC algorithms adopt the following novel designs to accurately estimate the policy gradient in an efficient way.

- *Noisy Rewards:* In a decentralized setting, local policy gradients (estimated locally by the agents) involve the average of all agents' local rewards. To help agents estimate this averaged reward without revealing the raw local rewards , we let them share Gaussian-corrupted local rewards with their neighbor, and the variance of the Gaussian noise can be adjusted by each agent to reach its desired level.

- *Mini-batch Updates:* We apply mini-batch Markovian sampling to both the decentralized actor and critic updates. This approach 1) helps the agents obtain accurate estimations of the corrupted averaged reward; 2) significantly reduces the variance of policy gradient caused by Markovian sampling; and 3) significantly reduces the communication frequency and complexity.

Moreover, for our decentralized NAC algorithm, we additionally adopt the following design to compute the inverse of the Fisher information matrix in an efficient and decentralized way.

- *Decentralized Natural Policy Gradient:* By reformulating the natural policy gradient as the minimizer of a quadratic program, we develop a decentralized SGD with Markovian sampling that allows the agents to estimate the corresponding local natural gradients by communicating only

---

[1]In this table, $\widetilde{\mathcal{O}}(\cdot)$ hides all logarithm factors. In (25), the sample complexity has been established for various AC-type algorithms, and we compare with the best one. In (70), the Algorithm 1 needs to share local actions while the Algorithm 2 does not.

scalar variables with their neighbors. In particular, in order to minimize the sample complexity of the decentralized SGD, we set the batch size to be exponentially increasing.

Theoretically, for the first time, we provide finite-time convergence analysis of decentralized AC and NAC algorithms under Markovian sampling. Specifically, we prove that our decentralized AC and NAC algorithms achieve the overall sample complexities $\mathcal{O}(\epsilon^{-2} \ln \epsilon^{-1})$ and $\mathcal{O}(\epsilon^{-3} \ln \epsilon^{-1})$, respectively, and both match the state-of-the-art complexities of their centralized versions (60). Moreover, both decentralized algorithms achieve a significantly reduced overall communication complexity $\mathcal{O}(\epsilon^{-1} \ln \epsilon^{-1})$. In particular, our analysis involves new technical developments. First, we need to characterize the bias and variance of (natural) policy gradient and stochastic gradient caused by the noisy rewards and the inexact local averaging steps, and control them with proper choices of batch sizes and number of local averaging steps. Second, when using decentralized Markovian SGD to compute the inverse Fisher information matrix, we need to use an exponentially increasing batch size to achieve an optimized sample complexity bound. Such a Markovian SGD with adaptive batch size has not been studied before and can be of independent interest.

## 1.2 RELATED WORK

**Convergence analysis of AC and NAC.** In the centralized setting, the AC algorithm was firstly proposed by (23) and later developed into the natural actor-critic (NAC) algorithm (37; 7). Specifically, (37) does not provide any convergence result, while (22; 5) and (20; 6; 7) establish the asymptotic convergence rate of centralized AC and NAC, respectively, which is weaker than our finite-time convergence results. Furthermore, (54; 25; 38; 61; 56) and (54) establish the finite-time convergence rate of centralized AC and NAC, respectively. Please refer to Table 1 for their sample complexities. Moreover, (60) improve the finite-time sample complexities of the above works to the state-of-the-art result for both centralized AC and NAC by leveraging mini batch sampling, and our sample complexities match these state-of-the-art results .

In the decentralized setting, a few works have established the almost sure convergence result of AC (15; 27; 47; 33), but they do not characterize the finite-time convergence rate and the sample complexity. To the best of our knowledge, there is no formally developed decentralized NAC algorithm.

**Decentralized TD-type algorithms.** The finite-time convergence of decentralized TD(0) has been obtained using i.i.d samples (52; 14; 53; 28) and Markovian samples (46; 53), respectively, without revealing the agents' local actions, policies and rewards . Decentralized off-policy TD-type algorithms have been studied in (34; 45; 9; 12).

**Decentralized AC in other MARL settings.** Some works apply decentralized AC to other MARL settings that are very different from ours. For example, (44; 36; 17; 11; 57) studied adversarial game. (30) studied a mixed cooperative-competitive environment where each agent maximizes its own Q function (30). (11) proposed Delay-Aware Markov Game which considers delay in Markov game. (68; 31) studied linear control system and linear quadratic regulators instead of an MDP. (55) studied sequential prisoner's dilemmas.

**Policy gradient algorithms.** Policy gradient (PG) and natural policy gradient (NPG) are popular policy optimization algorithms. (1) characterizes the iteration complexity (i.e., number of episodes) of centralized PG and NPG algorithms by assuming access to exact policy gradient. They also established a sample complexity result $\mathcal{O}(\epsilon^{-6})$ in the i.i.d. setting for NPG, which is worse than the state-of-the-art result $\mathcal{O}(\epsilon^{-3} \ln \epsilon^{-1})$ of both centralized NAC (60) and our decentralized NAC with Markovian samples. (3) proposes decentralized PG in a simple cooperative MARL setting, where all the agents share one action and the same policy, and they establish a iteration complexity in the order of $\mathcal{O}(\epsilon^{-4})$. (13; 73) apply decentralized PG to Markov games. (2) applies decentralized NPG to a different cooperative MARL setting where each agent observes its own state, takes its own action and has access to these information of its neighbors.

**Value-based algorithms.** Value-based algorithms have also been develop for MARL. Specifically, (21; 18) develop distributed Q-learning in a simplified cooperative MARL setting, where the agents share a joint action. In particular, (18) characterizes the convergence rate of a value function-based convergence error, which is a different optimality measure from that of AC-type algorithms. (35) applies distributed Q-learning to another cooperative MARL setting, where each agent observes

its own state and takes its own action. It establishes an asymptotic convergence guarantee, and no convergence rate is given. (40) develops a value propagation algorithm that uses primal-dual method to minimize a soft Bellman error in the MARL setting. Under an assumption that the variance of the stochastic gradient is uniformly bounded, it establishes a non-asymptotic convergence rate to an approximate stationary point.

## 2 REVIEW OF MULTI-AGENT REINFORCEMENT LEARNING

In this section, we first introduce some standard settings of RL. Consider an agent that starts from an initial state $s_0 \sim \xi$ and collects a trajectory of Markovian samples $\{s_t, a_t, R_t\}_t \subset \mathcal{S} \times \mathcal{A} \times \mathbb{R}$ by interacting with an underlying environment (with transition kernel $\mathcal{P}$) following a parameterized policy $\pi_\omega$ with induced stationary state distribution $\mu_\omega$. The agent aims to learn an optimal policy that maximizes the expected accumulated reward $J(\omega) = (1 - \gamma)\mathbb{E}\left[\sum_{t=0}^{\infty} \gamma^t R_t\right]$, where $\gamma \in (0, 1)$ is a discount factor. The marginal state distribution is denoted as $\mathbb{P}_\omega(s_t)$ and the visitation measure is defined as $\nu_\omega(s) := (1 - \gamma)\sum_{t=0}^{\infty} \gamma^t \mathbb{P}_\omega(s_t = s)$, both of which depend on the policy parameter $\omega \in \Omega$ and the transition kernel $\mathcal{P}$. We also define the mixed transition kernel $\mathcal{P}_\xi(\cdot|s, a) := \gamma\mathcal{P}(\cdot|s, a) + (1 - \gamma)\xi(\cdot)$, whose stationary state distribution is known to be $\nu_\omega$.

In the multi-agent RL (MARL) setting, $M$ agents are connected via a fully decentralized network and interact with a shared environment. The network topology is specified by a doubly stochastic communication matrix $W \in \mathbb{R}^{M \times M}$. At any time $t$, all the agents share a common state $s_t$. Then, every agent $m$ takes an action $a_t^{(m)}$ following its own current policy $\pi_t^{(m)}(\cdot|s_t)$ parameterized by $\omega_t^{(m)}$. After all the actions $a_t := \{a_t^{(m)}\}_{m=1}^M$ are taken, the global state $s_t$ transfers to a new state $s_{t+1}$ and every agent $m$ receives a local reward $R_t^{(m)}$. In this MARL setting, each agent $m$ can only access the global state $\{s_t\}_t$, its own actions $\{a_t^{(m)}\}_t$ and rewards $\{R_t^{(m)}\}_t$ and policy $\pi_t^{(m)}$. Next, define the joint policy $\pi_t(a_t|s_t) := \prod_{m=1}^M \pi_t^{(m)}(a_t^{(m)}|s_t)$ parameterized by $\omega_t = [\omega_t^{(1)}; \dots; \omega_t^{(M)}]$, and define the average reward $\overline{R}_t := \frac{1}{M}\sum_{m=1}^M R_t^{(m)}$. The goal of the agents is to collaboratively learn the optimal joint policy that maximizes the expected accumulated average reward $J(\omega) := (1 - \gamma)\mathbb{E}\left[\sum_{t=0}^{\infty} \gamma^t \overline{R}_t \Big| s_0 \sim \xi\right]$. Throughout, we consider the setting that the agents keep interacting with the environment and observing a trajectory of MDP transition samples, which are further used to learn the optimal joint policy.

## 3 SAMPLE AND COMMUNICATION-EFFICIENT DECENTRALIZED AC

In this section, we propose a decentralized actor-critic (AC) algorithm that is sample and communication-efficient and avoids revealing agents' local actions, policies and raw rewards .

We first consider a direct extension of the centralized AC to the decentralized case. As each agent $m$ has its own policy $\pi^{(m)}$, it aims to update the policy parameter $\omega^{(m)}$ using the local policy gradient $\nabla_{\omega^{(m)}} J(\omega)$. Under linear approximation of the value function $V_\theta(s) \approx \phi(s)^\top \theta$ where $\phi(s)$ is the feature vector, the local policy gradient has the following stochastic approximation.

$$\nabla_{\omega^{(m)}} J(\omega_t) \approx \left[\overline{R}_t + \gamma\phi(s'_{t+1})^\top\theta_t^{(m)} - \phi(s_t)^\top\theta_t^{(m)}\right]\psi_t^{(m)}(a_t^{(m)}|s_t), \tag{1}$$

$$\text{where } a_t^{(m)} \sim \pi_t^{(m)}(\cdot|s_t), s_{t+1} \sim \mathcal{P}_\xi(\cdot|s_t, a_t), s'_{t+1} \sim \mathcal{P}(\cdot|s_t, a_t). \tag{2}$$

Here, $\theta_t^{(m)}$ is agent $m$'s critic parameter and $\psi_t^{(m)}(a_t^{(m)}|s_t) = \nabla_{\omega^{(m)}} \ln \pi_t^{(m)}(a_t^{(m)}|s_t)$ is the local score function. It is clear that both $\theta_t^{(m)}$ and $\psi_t^{(m)}(a_t^{(m)}|s_t)$ can be obtained/computed by agent $m$ using the local information. However, the average reward $\overline{R}_t$ requires agent $m$ aggregating the local rewards from all the other agents, which raises concerns. In the existing literature on decentralized AC, this issue is avoided by either 1) sharing the agents' actions with each other instead (70; 69; 8; 36; 72; 27; 19; 26; 11), yet the action information is also highly sensitive; or 2) learning a parameterized model to estimate the average reward (70), which requires extra learning effort and does not provide an accurate estimation. Hence, we are motivated to develop a simpler approach that provides accurate estimation of the average reward while avoids sharing raw local rewards .

**1. Efficient Policy Gradient Estimation.** We propose a decentralized policy gradient estimation scheme that improves the sample and communication efficiency and avoids revealing the agents' local actions, policies and raw rewards . First, in order for each agent to estimate the average reward $\overline{R}_t$ in eq. (1), we let each agent $m$ generate a noisy local reward $\widetilde{R}_t^{(m)} = R_t^{(m)}(1 + e_t^{(m)})$ and share with other agents, where $e_t^{(m)} \sim \mathcal{N}(0, \sigma_m^2)$ [2]. The noise variance is determined by the agent based on its desired level. Specifically, every agent $m$ first initializes its local estimation of the averaged reward $\overline{R}_t^{(m)}$ using its own noisy reward, i.e., $\overline{R}_{t,0}^{(m)} = \widetilde{R}_t^{(m)}$. Then, each agent $m$ performs decentralized local averaging with its neighbors $\mathcal{N}_m$ for $T'$ iterations, i.e.,

$$\overline{R}_{t,\ell+1}^{(m)} = \sum_{m' \in \mathcal{N}_m} W_{m,m'} \overline{R}_{t,\ell}^{(m)}, \quad \ell = 0, 1, \ldots, T' - 1. \tag{3}$$

After that, agent $m$ obtains the final estimate $\overline{R}_t^{(m)} := \overline{R}_{t,T'}^{(m)}$. It can be shown that $\overline{R}_t^{(m)}$ converges to the averaged noisy reward $\frac{1}{M} \sum_{m=1}^M \widetilde{R}_t^{(m)}$ exponentially fast. Ideally, by averaging these noisy local rewards over the $M$ agents, the variance of the noise in the final estimation will be scaled by a factor of $\frac{1}{M}$. Therefore, to obtain an accurate estimation, the network needs to have a sufficiently large number of agents, which does not always hold in practice.

To address this issue, we let each agent $m$ collect a mini-batch of $N$ Markovian samples in each iteration $t$ to estimate the local policy gradient, which then takes the following form.

$$\widehat{\nabla}_{\omega^{(m)}} J(\omega_t) = \frac{1}{N} \sum_{i=tN}^{(t+1)N-1} \Big[ \overline{R}_i^{(m)} + \gamma \phi(s'_{i+1})^\top \theta_t^{(m)} - \phi(s_i)^\top \theta_t^{(m)} \Big] \psi_t^{(m)}(a_i^{(m)} | s_i), \tag{4}$$

where $\overline{R}_i^{(m)}$ is an estimation of $\overline{R}_i$ obtained by agent $m$ following the process described in eq. (3). Intuitively, each $\overline{R}_i^{(m)}$ is corrupted by a zero-mean noise with variance $\mathcal{O}(\frac{1}{M})$ due to averaging over the agents. Then, the mini-batch samples further help scale the noise variance by a factor of $\frac{1}{N}$. Consequently, with a sufficiently large batch size $N$, we can obtain an accurate estimation of the averaged reward and hence the policy gradient. To summarize, **our decentralized policy gradient estimation scheme has the following advantages.**

- *Avoid sharing raw rewards:* The agents share only noisy rewards $\widetilde{R}_t^{(m)}$ with their neighbors, and the noise variance can be adjusted based on the desired level such that $R_t^{(m)}$ is unknown to the other agents. This is in contrast to other decentralized AC algorithms where the agents need to either share local actions, rewards or collaboratively learn an additional parameterized reward model.

- *Sample-efficient:* The mini-batch updates help greatly suppress the noise variance of the local policy gradient in (4) and improve its estimation accuracy. On the other hand, mini-batch policy gradient also helps reduce the optimization variance caused by Markovian sampling and leads to a good finite-time sample complexity as we prove later. We note that there is no trade-off between noise variance and sample efficiency here, because for highly noisy local rewards we can choose a large batch size to suppress the overall estimation error to the desired level.

- *Communication-efficient:* The mini-batch updates also significantly reduce the communication frequency as well as the complexity as we prove later. In comparison, the existing decentralized AC requires to perform one communication round per Markovian sample.

**Remark.** *We note that the local mini-batch policy gradient update in eq. (4) can be computed in an accumulative way by the agent when observing the mini-batch of transition samples on the fly. There is no need to store all these samples and perform a large batch computation.*

**2. Fully Decentralized Critic Update.** The critic parameters of the agents are updated following the standard decentralized TD-type algorithm. Specifically, consider the $t$-th local critic update of each agent $m$. It first collects a mini-batch of $N_c$ Markovian samples. Then, starting from a fixed initialization $\theta_{t,0}^{(m)} = \theta_{-1}$, agent $m$ performs $T_c$ iterations of decentralized TD updates as follows, where $\{s_t\}_{t \in \mathbb{N}}$ follows the transition kernel $\mathcal{P}$ and $a_t^{(m)} \sim \pi_t^{(m)}(\cdot | s_t)$: for $t' = 0, 1, ..., T_c - 1$,

$$\theta_{t,t'+1}^{(m)} = \sum_{m' \in \mathcal{N}_m} W_{m,m'} \theta_{t,t'}^{(m')} + \frac{\beta}{N_c} \sum_{i=tN_c}^{(t+1)N_c-1} \Big[ R_i^{(m)} + \gamma \phi(s_{i+1})^\top \theta_{t,t'}^{(m)} - \phi(s_i)^\top \theta_{t,t'}^{(m)} \Big] \phi(s_i). \tag{5}$$

---

[2]More generally, any noise with zero mean and variance $\sigma_m^2$ will work.

Then, the updated critic parameter is set to be $\theta_t^{(m)} := \theta_{t,T_c}^{(m)}$. To further reduce the consensus error, we perform additional $T_c'$ steps of local model averaging, as also adopted in (12). The pseudo code of the entire decentralized AC algorithm is summarized in Algorithms 1 and 2 below.

---

**Algorithm 1** Decentralized Actor-Critic

**Initialize:** Actor-critic parameters $\omega_0, \theta_{-1}$.
**for** *actor iterations* $t = 0, 1, \ldots, T - 1$ **do**
  ▶ **Critic update on** $\theta_t$**:** by Algorithm 2.
  ▶ Collect $N$ Markovian samples by eq. (2).
  **for** *agents* $m = 1, \ldots, M$ *in parallel* **do**
    ▶ Send noisy local rewards and perform $T'$ local average steps following eq. (3).
    ▶ Compute the estimated local policy gradient $\widehat{\nabla}_{\omega^{(m)}} J(\omega_t)$ following eq. (4).
    ▶ **Actor update on** $\omega_t$**:**
    $\omega_{t+1}^{(m)} = \omega_t^{(m)} + \alpha \widehat{\nabla}_{\omega^{(m)}} J(\omega_t)$.
  **end**
**end**
**Output:** $\omega_{\widetilde{T}}$ with $\widetilde{T} \overset{\text{uniform}}{\sim} \{1, 2, \ldots, T\}$.

---

**Algorithm 2** Decentralized TD (critic update)

**Initialize:** Critic parameter $\theta_{t,0} = \theta_{-1}$.
**for** *critic iterations* $t' = 0, 1, \ldots, T_c - 1$ **do**
  ▶ Collect $N_c$ Markovian samples following policy $\pi_t$ and transition kernel $\mathcal{P}$.
  **for** *agents* $m = 1, \ldots, M$ *in parallel* **do**
    ▶ Send local critic parameters.
    ▶ Decentralized TD update in eq. (5).
  **end**
**end**
**for** *iterations* $t' = T_c, \ldots, T_c + T_c' - 1$ **do**
  **for** *agents* $m = 1, \ldots, M$ *in parallel* **do**
    ▶ $\theta_{t,t'+1}^{(m)} = \sum_{m' \in \mathcal{N}_m} W_{m,m'}\, \theta_{t,t'}^{(m')}$.
  **end**
**end**
**Output:** $\theta_t = \theta_{t,T_c+T_c'}$.

---

## 4 FINITE-TIME CONVERGENCE ANALYSIS OF DECENTRALIZED AC

In this section, we analyze the finite-time convergence of Algorithm 1 and characterize the sample and communication complexities. All the notations and universal constants are summarized in Appendices A & F respectively. We first introduce the following standard assumptions that have been widely adopted in the existing literature.

**Assumption 1.** *Regarding the transition kernels $\mathcal{P}, \mathcal{P}_\xi$, denote $\mu_\omega, \nu_\omega$ respectively as their stationary state distributions under policy $\pi_\omega$ and denote $\mathbb{P}, \mathbb{P}_\xi$ respectively as their marginal state distributions. Then, there exist constants $\kappa > 0$ and $\rho \in (0, 1)$ such that for all $t \geq 0$,*

$$\sup_{s \in \mathcal{S}} d_{TV}\big(\mathbb{P}\,(s_t \mid s_0 = s), \mu_\omega\big) \leq \kappa \rho^t, \quad \sup_{s \in \mathcal{S}} d_{TV}\big(\mathbb{P}_\xi\,(s_t \mid s_0 = s), \nu_\omega\big) \leq \kappa \rho^t \quad (6)$$

*where $d_{TV}(P, Q)$ denotes the total-variation distance between probability measures $P$ and $Q$.*

**Assumption 2.** *There exist constants $C_\psi, L_\psi, L_\pi > 0$ such that for all $\omega, \widetilde{\omega} \in \Omega$, $s \in \mathcal{S}$ and $a \in \mathcal{A}$, $\|\psi_\omega(a|s)\| \leq C_\psi$, $\|\psi_{\widetilde{\omega}}(a|s) - \psi_\omega(a|s)\| \leq L_\psi \|\widetilde{\omega} - \omega\|$ and $d_{TV}\big(\pi_{\widetilde{\omega}}(\cdot|s), \pi_\omega(\cdot|s)\big) \leq L_\pi \|\widetilde{\omega} - \omega\|$.*

**Assumption 3.** *There exists $R_{\max} > 0$ such that for any agent $m$ and any Markovian sample $(s, a, s')$, we have $0 \leq R^{(m)}(s, a, s') \leq R_{\max}$.*

**Assumption 4.** *The feature vectors satisfy $\|\phi(s)\| \leq 1$ for all $s \in \mathcal{S}$. There exists a constant $\lambda_\phi > 0$ such that $\lambda_{\min}\big(\mathbb{E}_{s \sim \mu_\omega}[\phi(s)\phi(s)^\top]\big) \geq \lambda_\phi$ for all $\omega$.*

**Assumption 5.** *The communication matrix $W \in \mathbb{R}^{M \times M}$ of the decentralized network is doubly stochastic, and its second largest singular value satisfies $\sigma_W \in [0, 1)$.*

Assumption 1 has been widely considered in the existing literature (4; 38; 63; 58; 42; 60; 12) and it holds for any time-homogeneous Markov chains with finite-state space and any uniformly ergodic Markov chains. Assumption 2 introduces boundedness and Lipschitzness to the policy and its associated score function (65; 60), and holds for many parameterized policies such as Gaussian policy (25) and Boltzman policy (16). Assumption 4 can always hold by normalizing the feature vector $\phi(s)$ Assumption 5 is widely used in decentralized optimization (43; 41) and multi-agent reinforcement learning (46; 53; 12), which ensures that all the decentralized agents can reach a global consensus.

With the above assumptions, we obtain the following finite-time convergence result of the decentralized AC algorithm. Throughout, we follow (60; 56) and define the critic approximation error as $\zeta_{\text{approx}}^{\text{critic}} := \sup_\omega \mathbb{E}_{s \sim \nu_\omega}(V_\omega(s) - \phi(s)^\top \theta_\omega^*)^2$ where $\theta_\omega^*$ is the optimal critic parameter (see its definition right before Lemma D.3 in Appendix D). We also define sample complexity as the total number of Markovian samples required for achieving $\mathbb{E}[\|\nabla J(\omega)\|^2] \leq \epsilon$. All the universal constants are listed in Appendix F.

**Theorem 1.** *Let Assumptions 1–5 hold and adopt the hyperparameters of the decentralized TD in Algorithm 2 following Lemma D.4. Choose $\alpha \leq \frac{1}{4L_J}$, $T' \geq \frac{\ln M}{2 \ln \sigma_W^{-1}}$. Then, the output of the decentralized AC in Algorithm 1 has the following convergence rate.*

$$\mathbb{E}\Big[\big\|\nabla J(\omega_{\widetilde{T}})\big\|^2\Big] \leq \frac{4R_{\max}}{T\alpha} + 4(c_4\sigma_W^{2T'} + c_5\beta^2\sigma_W^{2T_c'}) + 4c_6\Big(1 - \frac{\lambda_B}{8}\beta\Big)^{T_c} + \frac{4c_7}{N} + \frac{4c_8}{N_c} + 64C_\psi^2\zeta_{approx}^{critic}.$$

*Moreover, to achieve $\mathbb{E}\big[\big\|\nabla J(\omega_{\widetilde{T}})\big\|^2\big] \leq \epsilon$ for any $\epsilon \geq 128C_\psi^2\zeta_{approx}^{critic}$, we can choose $T, N, N_c = \mathcal{O}(\epsilon^{-1})$ and $T_c, T_c', T' = \mathcal{O}(\ln \epsilon^{-1})$. Consequently, the overall sample complexity is $T(T_cN_c+N) = \mathcal{O}(\epsilon^{-2}\ln\epsilon^{-1})$, and the communication complexities for synchronizing linear model parameters and rewards are $T(T_c + T_c') = \mathcal{O}(\epsilon^{-1}\ln\epsilon^{-1})$ and $TT' = \mathcal{O}(\epsilon^{-1}\ln\epsilon^{-1})$, respectively.*

**Remark.** *We note that the constraint on $\epsilon$ is naturally induced by the critic approximation error. In particular, if this approximation errors vanish, for example, when the dimension of features equals the number of states and the parameterized policy space is sufficiently expressive, then we can achieve arbitrarily small target accuracy.*

To the best of our knowledge, Theorem 1 provides the first finite-time analysis of decentralized AC under Markovian sampling. To elaborate, under any pre-specified variance $\sigma_m^2$ of the reward noise, our result shows that the gradient norm asymptotically converges to the order $\mathcal{O}(N^{-1} + N_c^{-1} + \zeta_{approx}^{critic})$, which can be made arbitrarily close to the linear model approximation error $\zeta_{approx}^{critic}$ by choosing sufficiently large batch sizes $N, N_c$. In particular, exact gradient convergence can be achieved when there is no model approximation error. The overall sample complexity of our decentralized AC is $\mathcal{O}(\epsilon^{-2}\ln\epsilon^{-1})$, matching the state-of-the-art complexity result for centralized AC (60). Moreover, with proper choices of the batch sizes $N, N_c = \mathcal{O}(\epsilon^{-1})$, the overall communication complexity is significantly reduced to $\mathcal{O}(\epsilon^{-1}\ln\epsilon^{-1})$.

The proof of Theorem 1 relies on developing several new algorithmic and technical developments to reduce the communication complexity of both the decentralized actor and critic updates while establishing tight convergence error bounds for both components. We further elaborate on these novel technical developments below.

- To achieve an overall reduced communication complexity, we adopt mini-batch updates in both the actor and critic steps to reduce the communication frequency, as opposed to the single sample-based update adopted in the existing work on decentralized TD learning (46). Specifically, in the analysis of the decentralized TD described in Algorithm 2 (see Lemma D.4), the mini-batch updates with batch size $\mathcal{O}(\epsilon^{-1})$ substantially improve the communication complexity from $\mathcal{O}(\epsilon^{-1}\ln\epsilon^{-1})$ to $\mathcal{O}(\ln\epsilon^{-1})$ while help achieve the state-of-the-art sample complexity. Eventually, this together with the mini-batch updates in the decentralized actor steps help achieve the desired overall low communication complexity.

- To achieve the state-of-the-art overall sample complexity, it is critical that the policy gradient vanishes fast, which further requires a fast convergence of the decentralized TD learning. However, although the standard $T_c$ decentralized mini-batch TD updates can yield a small convergence error for the global critic model (i.e., the average of all local critic models), it still suffers from a relatively large consensus error. To resolve this issue, we introduce an additional $T_c'$ global consensus steps in Algorithm 2 to reduce the consensus error. It is proved that a small number $\mathcal{O}(\ln\epsilon^{-1})$ of such steps suffices to yield a desired TD error.

- We inject random noises into the local raw rewards $R_t^{(m)}$ to protect the information. These noises introduce additional Markovian bias and variance to the local policy gradients in (4). Fortunately, as proved in Lemma D.6, by applying mini-batch policy gradient updates, we are able to control the bias and variance induced by the noisy rewards to an acceptable level that does not affect the overall sample and communication complexities.

## 5 DECENTRALIZED NATURAL AC

Natural actor-critic (NAC) is a popular variant of the AC algorithm. It utilizes a Fisher information matrix to perform a natural policy gradient update, which helps attain the globally optimal solution in terms of the function value convergence. In this section, we develop a fully decentralized version of the NAC algorithm that is sample and communication-efficient.

---

**Algorithm 3** Decentralized Natural Actor-Critic

---

**Initialize:** Actor-critic parameters $\omega_0, \theta_{-1}$, natural policy gradient $h_{-1}$.
**for** *actor iterations* $t = 0, 1, \ldots, T - 1$ **do**
   ▶ **Critic update on** $\theta_t$**:** by Algorithm 2.
   **for** *agents* $m = 1, ..., M$ *in parallel* **do**
      **for** *iterations* $k = 0, 1, \ldots, K - 1$ **do**
         ▶ Collect $N_k$ Markovian samples following eq. (2).
         ▶ Send $\widetilde{R}_i^{(m)}$ and $z_{i,\ell}^{(m)}$ and perform $T'$ and $T_z$ local average steps, respectively.
         ▶ Estimate local gradient $\widehat{\nabla}_{\omega^{(m)}} f_{\omega_t}(h_{t,k})$ following eqs. (8) and (4).
         ▶ Perform SGD update in eq. (9).
      **end**
      ▶ **Actor update on** $\omega_t$**:** $\omega_{t+1}^{(m)} = \omega_t^{(m)} + \alpha h_t^{(m)}$.
   **end**
**end**

**Output:** $\omega_{\widetilde{T}}$ with $\widetilde{T} \overset{\text{uniform}}{\sim} \{1, 2, \ldots, T\}$.

---

A major challenge of developing fully decentralized NAC algorithm is computing the inverse Fisher information matrix-vector product involved in the natural policy gradient update. To explain, first recall the exact natural policy gradient update of the centralized NAC algorithm, i.e., $\omega_{t+1} = \omega_t + \alpha F(\omega_t)^{-1} \nabla J(\omega_t)$, where $F(\omega_t) := \mathbb{E}_{s_t \sim \nu_{\omega_t}, a_t \sim \pi_t(\cdot|s_t)} [\psi_t(a_t|s_t)\psi_t(a_t|s_t)^\top]$ is the Fisher information matrix. However, in the multi-agent case, it is challenging to perform the natural policy gradient update in a decentralized manner. This is because the Fisher information matrix $F(\omega_t)$ is based on the concatenated multi-agent score vector $\psi_t(a_t|s_t) = [\psi_t^{(1)}(a_t^{(1)}|s_t); ...; \psi_t^{(M)}(a_t^{(M)}|s_t)]$ and the inverse matrix-vector product $F(\omega_t)^{-1} \nabla J(\omega_t)$ is not separable with regard to each agent's policy parameter dimensions. Next, we develop a **fully decentralized** scheme to implement the natural policy gradient update in the multi-agent setting.

First, note that the natural policy gradient update $h(\omega_t) := F(\omega_t)^{-1} \nabla J(\omega_t)$ is equivalent to the solution of a quadratic program, i.e.,

$$h(\omega_t) = \arg\min_h f_{\omega_t}(h) := \frac{1}{2} h^\top F(\omega_t) h - \nabla J(\omega_t)^\top h. \tag{7}$$

Therefore, we can apply $K$ steps of SGD with Markovian sampling to solve this problem and obtain an estimated natural policy gradient update. Specifically, starting from the initialization $h_{t,0} = h_{t-1}$ (obtained in the previous iteration), in the $k$-th SGD step, we sample a mini-batch $\mathcal{B}_{t,k}$ [3] of $N_k$ Markovian samples to estimate $\nabla f_{\omega_t}(h)$ as $\frac{1}{N_k} \sum_{i \in \mathcal{B}_{t,k}} \psi_t(a_i|s_i)\psi_t(a_i|s_i)^\top h_{t,k} - \widehat{\nabla} J(\omega_t; \mathcal{B}_{t,k})$, where $\widehat{\nabla} J(\omega_t; \mathcal{B}_{t,k})$ is estimated in the same decentralized way as eq. (4) using the mini-batch of samples $\mathcal{B}_{t,k}$. In particular, each agent $m$ needs to compute the corresponding local gradient $\frac{1}{N_k} \sum_{i \in \mathcal{B}_{t,k}} \psi_t^{(m)}(a_i^{(m)}|s_i)[\psi_t(a_i|s_i)^\top h_{t,k}] - \widehat{\nabla}_{\omega^{(m)}} J(\omega_t; \mathcal{B}_{t,k})$, in which $\psi_t^{(m)}(a_i^{(m)}|s_i)$ and $\widehat{\nabla}_{\omega^{(m)}} J(\omega_t; \mathcal{B}_{t,k})$ can be computed/estimated by the agent $m$. Then, it suffices to obtain an estimate of the scalar $\psi_t(a_i|s_i)^\top h_{t,k}$, which can be rewritten as $\sum_{m=1}^M \psi_t^{(m)}(a_i^{(m)}|s_i)^\top h_{t,k}^{(m)}$. This summation can be easily estimated by the decentralized agents through local averaging. Specifically, each agent $m$ locally computes $z_{i,0}^{(m)} = \psi_t^{(m)}(a_i^{(m)}|s_i)^\top h_{t,k}^{(m)}$ and performs $T_z$ steps of local averaging, i.e., $z_{i,\ell+1}^{(m)} = \sum_{m' \in \mathcal{N}_m} W_{m,m'} z_{i,\ell}^{(m')}$, $\ell = 0, 1, \ldots, T_z - 1$. After that, the quantity $M z_{i,T_z}^{(m)}$ can be proven to converge to the desired summation $\sum_{m=1}^M \psi_t^{(m)}(a_i^{(m)}|s_i)^\top h_{t,k}^{(m)}$ exponentially fast. Finally, the local gradient for agent $m$ is approximated as

$$\widehat{\nabla}_{\omega^{(m)}} f_{\omega_t}(h_{t,k}) = \frac{M}{N_k} \sum_{i \in \mathcal{B}_{t,k}} \psi_t^{(m)}(a_i^{(m)}|s_i) z_{i,T_z}^{(m)} - \widehat{\nabla}_{\omega^{(m)}} J(\omega_t; \mathcal{B}_{t,k}). \tag{8}$$

Then, the agent $m$ performs the following SGD updates to obtain $h_t^{(m)} := h_{t,K}^{(m)}$.

$$h_{t,k+1}^{(m)} = h_{t,k}^{(m)} - \eta \widehat{\nabla}_{\omega^{(m)}} f_{\omega_t}(h_{t,k}), \quad k = 0, ..., K - 1. \tag{9}$$

---

[3] Specifically, the mini-batch $\mathcal{B}_{t,k}$ contains sample indices $\{tN + \sum_{k'=0}^{k-1} N_{k'}, \ldots, tN + \sum_{k'=0}^{k} N_{k'} - 1\}$.

We emphasize that the above mini-batch SGD updates use **Markovian samples**. In particular, as shown in Section 6, we need to develop an adaptive batch size scheduling scheme for this SGD in order to reduce its sample complexity. We summarize the decentralized NAC in Algorithm 3.

# 6 FINITE-TIME CONVERGENCE ANALYSIS OF DECENTRALIZED NAC

To analyze the decentralized NAC, we introduce the following additional standard assumptions.

**Assumption 6.** *There exists a constant $\lambda_F > 0$ such that $\lambda_{\min}\big(F(\omega)\big) \geq \lambda_F > 0, \forall \omega \in \Omega$.*

**Assumption 7.** *There exists $C_* > 0$ such that for $\omega^* = \arg\max_{\omega \in \Omega} J(\omega)$ and any $\omega \in \Omega$,*

$$\mathbb{E}_{s \sim \nu_\omega, a \sim \pi_\omega(\cdot|s)}\Big[\Big(\frac{\nu_{\omega^*}(s)\pi_{\omega^*}(a|s)}{\nu_\omega(s)\pi_\omega(a|s)}\Big)^2\Big] \leq C_*^2.$$

Assumption 6 ensures that the Fisher information matrix $F(\omega)$ is uniformly positive definite, and is also considered in (65; 29; 62). Assumption 7 regularizes the discrepancy between the stationary state-action distributions $\nu_{\omega^*}(s)\pi_{\omega^*}(a|s)$ and $\nu_\omega(s)\pi_\omega(a|s)$ (54; 59).

We obtain the following finite-time convergence result of the decentralized NAC algorithm. Throughout, we follow (54; 60; 62) and define the actor approximation error as $\zeta_{\text{approx}}^{\text{actor}} := \sup_\omega \min_h \mathbb{E}_{s \sim \nu_\omega, a \sim \pi_\omega(\cdot|s)}\big[\big(\psi_\omega(a|s)^\top h - A_\omega(s,a)\big)^2\big]$. All universal constants are listed in Appendix F.

**Theorem 2.** *Let Assumptions 1–7 hold and adopt the hyperparameters of the decentralized TD in Algorithm 2 following Lemma D.4. Choose hyperparameters $\alpha \leq \min\big(1, \frac{\lambda_F^2}{4L_J C_\psi^2}, \frac{C_\psi^2}{2L_J}\big)$, $\beta \leq 1$, $T' \geq \frac{\ln M}{2 \ln \sigma_W^{-1}}$, $\eta \leq \frac{1}{2C_\psi^2}$, $T_z \geq \frac{\ln(3D_J C_\psi^2)}{\ln \sigma_W^{-1}}$, $K \geq \frac{\ln 3}{\ln(1-\eta\lambda_F/2)^{-1}}$, $N \geq \frac{2304 C_\psi^4(\kappa+1-\rho)}{\eta\lambda_F^5(1-\rho)(1-\eta\lambda_F/2)^{(K-1)/2}}$ and $N_k \propto \big(1 - \eta\lambda_F/2\big)^{-k/2}$. Then, the output of Algorithm 3 satisfies*

$$J(\omega^*) - \mathbb{E}\big[J(\omega_{\widetilde{T}})\big] \leq \frac{c_{17}}{T\alpha} + c_{18}\Big(1 - \frac{\eta\lambda_F}{2}\Big)^{(K-1)/4} + c_{19}\sigma_W^{T_z} + c_{20}\sigma_W^{T'} + c_{21}\beta\sigma_W^{T_c'} + \frac{c_{23}}{\sqrt{N_c}}$$
$$+ c_{22}\Big(1 - \frac{\lambda_B}{8}\beta\Big)^{T_c/2} + C_\psi\sqrt{c_{16}\zeta_{\text{approx}}^{\text{critic}}} + c_{24}\zeta_{\text{approx}}^{\text{critic}} + C^*\sqrt{\zeta_{\text{approx}}^{\text{actor}}}.$$

*Moreover, to achieve $J(\omega^*) - \mathbb{E}\big[J(\omega_{\widehat{T}})\big] \leq \epsilon$ for any $\epsilon \geq 2C_\psi\sqrt{c_{16}\zeta_{\text{approx}}^{\text{critic}}} + 2c_{24}\zeta_{\text{approx}}^{\text{critic}} + 2C^*\sqrt{\zeta_{\text{approx}}^{\text{actor}}}$, we can choose $T = \mathcal{O}(\epsilon^{-1})$, $N, N_c = \mathcal{O}(\epsilon^{-2})$, $T_c, T_c', T', T_z, K = \mathcal{O}(\ln \epsilon^{-1})$. Consequently, the overall sample complexity is $T(T_c N_c + N) = \mathcal{O}(\epsilon^{-3} \ln \epsilon^{-1})$, and the communication complexities for synchronizing linear model parameters and rewards are $T(T_c + T_c') = \mathcal{O}(\epsilon^{-1} \ln \epsilon^{-1})$ and $TT' = \mathcal{O}(\epsilon^{-1} \ln \epsilon^{-1})$, respectively.*

Theorem 2 provides the first finite-time analysis of fully decentralized natural AC algorithm. Our result proves that the function value optimality gap converges to the order $\mathcal{O}\big(N_c^{-1/2} + \sqrt{\zeta_{\text{approx}}^{\text{critic}}} + \sqrt{\zeta_{\text{approx}}^{\text{actor}}}\big)$, which can be made arbitrarily close to the actor and critic approximation error by choosing a sufficiently large batch size $N_c$. In particular, exact global optimum can be achieved when there is no model approximation error. We note that the overall sample complexity of our decentralized NAC is $\mathcal{O}(\epsilon^{-3} \ln \epsilon^{-1})$, matching the state-of-the-art complexity result for centralized NAC (60). Moreover, with the mini-batch updates, the overall communication complexity is significantly reduced to $\mathcal{O}(\epsilon^{-1} \ln \epsilon^{-1})$.

Similar to that of Theorem 1, our analysis of Theorem 2 also leverages the mini-batch decentralized TD updates to reduce the communication complexity and deal with the bias and variance of the local policy gradient introduced by noisy rewards. In addition, decentralized NAC uses mini-batch SGD with Markovian sampling to solve the quadratic problem in eq. (7). Here, we use a special geometrically increasing batch size scheduling scheme, i.e., $N_k \propto (1 - \eta\lambda_F/2)^{-k/2}$, to achieve the best possible convergence rate under the total sample budget that $\sum_{k=1}^K N_k = N$ and obtain the desired overall sample complexity result. Such an analysis of SGD with Markovian sampling under adaptive batch size scheduling has not been studied in the literature and can be of independent interests.

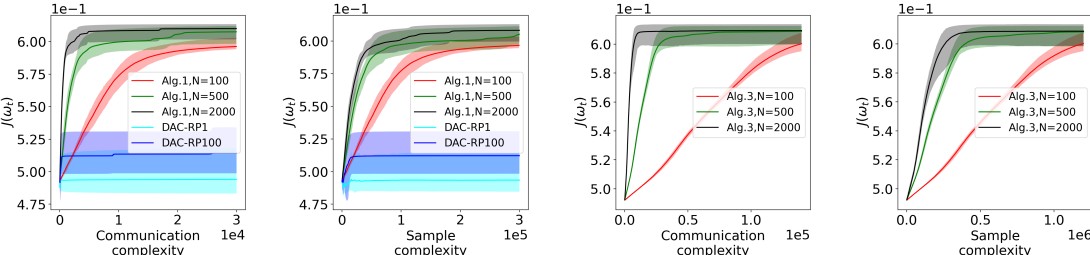

Figure 1: Comparison of accumulated discounted reward $J(\omega_t)$ among decentralized AC-type algorithms in a ring network with sparse connections .

# 7 EXPERIMENTS

We simulate a fully decentralized ring network with 6 agents. Please refer to Appendix E for detailed environment setup. We implement four decentralized AC-type algorithms and compare their performance, namely, our Algorithms 1 and 3, an existing decentralized AC algorithm (Algorithm 2 of (70)) that uses a linear model to parameterize the agents' averaged reward (we name it DAC-RP1 for decentralized AC with reward parameterization), and our proposed modified version of DAC-RP1 to incorporate minibatch, which we refer to as DAC-RP100 with batch size $N = 100$. For our Algorithm 1, we choose $T = 500$, $T_c = 50$, $T'_c = 10$, $N_c = 10$, $T' = T_z = 5$, $\beta = 0.5$, $\{\sigma_m\}_{m=1}^6 = 0.1$, and consider batch size choices $N = 100, 500, 2000$. Algorithm 3 uses the same hyperparameters as those of Algorithm 1 except that $T = 2000$ in Algorithm 3. For DAC-RP1, we set learning rates $\beta_\theta = 2(t+1)^{-0.9}$, $\beta_v = 5(t+1)^{-0.8}$ and batch size $N = 1$ as mentioned in (70). The modified DAC-RP100 adopts the same learning rates as Algorithm 1 with $N = 100$.

Figure 1 plots the accumulated reward $J(\omega_t)$ v.s. communication and sample complexity. Each curve includes 10 repeated experiments, and its upper and lower envelopes denote the 95% and 5% percentiles of the 10 repetitions, respectively. For our decentralized AC algorithm (left two figures), its communication and sample complexities for achieving a high accumulated reward are significantly reduced under a larger batch size $N$. This matches our theoretical understanding in Theorem 1 that a large $N$ helps reduce the communication frequency and policy gradient variance. In comparison, DAC-RP1 (with $N = 1$) has almost no improvement on the accumulated reward. Moreover, although the modified DAC-RP100 (with $N = 100$) outperforms DAC-RP1, its performance is much worse than our Algorithm 1 with $N = 100$. This performance gap is due to two reasons: (i) Both DAC-RP algorithms suffer from an inaccurate parameterized estimation of the averaged reward, and their mean relative reward errors are over 100%. In contrast, our noisy averaged reward estimation achieves a mean relative error in the range of $10^{-5} \sim 10^{-4}$;(ii) Both DAC-RP algorithms apply only a single TD update per-round, and hence suffers from a large mean relative TD error (about $2\%$ and $1\%$ for DAC-RP1 and DAC-RP100, respectively)whereas our algorithms perform multiple TD learning updates per-round and achieve a smaller mean relative TD error (about $0.3\%$). For our decentralized NAC algorithm (right two figures), one can make similar observations and conclusions.

# 8 CONCLUSION

We developed fully-decentralized AC and NAC algorithms that are efficient and do not reveal agents' local actions and policies . The agents share noisy reward information and adopt mini-batch updates to improve sample and communication efficiency. Under Markovian sampling and linear function approximation, we proved that our decentralized AC and NAC algorithms achieve the state-of-the-art sample complexities $\mathcal{O}(\epsilon^{-2} \ln \epsilon^{-1})$ and $\mathcal{O}(\epsilon^{-3} \ln \epsilon^{-1})$, respectively, and they both achieve a small communication complexity $\mathcal{O}(\epsilon^{-1} \ln \epsilon^{-1})$. Numerical experiments demonstrate that our algorithms achieve better sample and communication complexity than the existing decentralized AC algorithm that adopts reward parameterization.

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

# Appendix

## Table of Contents

## A NOTATIONS

**Norms:** For any vector $x$, we denote $\|x\|$ as its $\ell_2$ norm. For any matrix $X$, we denote $\|X\|, \|X\|_F$ as its spectral norm and Frobenius norm, respectively.

**Difference matrix:** $\Delta := I - \frac{1}{M}\mathbf{1}\mathbf{1}^\top$, where $\mathbf{1}$ denotes a column vector that consists of 1s.

**Moments of random vectors:** For a random vector $X$, we define its variance and covariance matrix as $\mathrm{Var}(X) := \mathbb{E}\|X - \mathbb{E}X\|^2$ and $\mathrm{Cov}(X) := \mathbb{E}\big([X - \mathbb{E}X][X - \mathbb{E}X]^\top\big)$, respectively. It is well known that $\mathbb{E}\|X\|^2 = \mathrm{Var}(X) + \|\mathbb{E}X\|^2$ and that $\mathrm{Var}(X) = \mathrm{tr}[\mathrm{Cov}(X)]$.

**Score function:** At any time $t$, The joint score function $\psi_t(a_t|s_t) := \nabla_\omega \ln \pi_t(a_t|s_t)$ can be decomposed into individual score functions $\psi_t^{(m)}(a_t^{(m)}|s_t) := \nabla_{\omega^{(m)}} \ln \pi_t^{(m)}(a_t^{(m)}|s_t)$ as $\psi_t(a_t|s_t) = [\psi_t^{(1)}(a_t^{(1)}|s_t), \ldots, \psi_t^{(M)}(a_t^{(M)}|s_t)]$.

**Reward functions:** At any time $t$, we denote $R_t^{(m)} := R^{(m)}(s_t, a_t, s_{t+1})$ and $\overline{R}_t := \overline{R}(s_t, a_t, s_{t+1})$, where $\overline{R}(s, a, s') = \frac{1}{M}\sum_{m=1}^M R^{(m)}(s, a, s')$.

**Policy gradient:** The policy gradient theorem (50) shows that

$$\nabla J(\omega) = \mathbb{E}_{\nu_\omega}\big[A_\omega(s, a)\psi_\omega(s, a)\big]. \tag{10}$$

where $A_\omega(s, a) := Q_\omega(s, a) - V_\omega(s)$ denotes the advantage function. In the decentralized case, we have the approximations $V_\omega(s_t) \approx \phi(s_t)^\top \theta$, $Q_\omega(s_t, a_t) \approx \overline{R}_t + \gamma\phi(s'_{t+1})^\top \theta$ where $s'_{t+1} \sim \mathcal{P}(\cdot|s_t, a_t)$. Therefore, we can stochastically approximate the partial policy gradient as eq. (1), i.e., for $m = 1, \ldots, M$,

$$\nabla_{\omega^{(m)}} J(\omega_t) \approx \Big[\overline{R}_t + \gamma\phi(s'_{t+1})^\top \theta_t^{(m)} - \phi(s_t)^\top \theta_t^{(m)}\Big]\psi_t^{(m)}(a_t^{(m)}|s_t).$$

We also define the following mini-batch stochastic (partial) policy gradient.

$$\widetilde{\nabla}_{\omega^{(m)}} J(\omega_t) := \frac{1}{N}\sum_{i=tN}^{(t+1)N-1}\Big[\overline{R}_i + \gamma\phi(s'_{i+1})^\top \theta_t^{(m)} - \phi(s_i)^\top \theta_t^{(m)}\Big]\psi_t^{(m)}(a_i^{(m)}|s_i).$$

$$\widetilde{\nabla} J(\omega_t) := \Big[\widetilde{\nabla}_{\omega^{(1)}} J(\omega_t); \ldots; \widetilde{\nabla}_{\omega^{(M)}} J(\omega_t)\Big].$$

**Filtrations:** We define the following filtrations for Algorithms 1 & 3.

$$\mathcal{F}_t := \sigma\big(\{\theta_{t'}^{(m)}\}_{m\in\mathcal{M},0\leq t'\leq t} \cup \{s_i,a_i,s'_{i+1},\{e_i^{(m)}\}_{m\in\mathcal{M}}\}_{i=0}^{tN-1} \cup \{s_{tN}\}\big).$$

$$\mathcal{F}'_t := \sigma\big[\mathcal{F}_t \cup \sigma\big(\{s_i,a_i,s'_{i+1}\}_{i=tN+1}^{(t+1)N-1}\big)\big].$$

$$\mathcal{F}_{t,k} = \sigma\big[\mathcal{F}_t \cup \sigma\big(\{s_i,a_i,s_{i+1},s'_{i+1},\{e_i^{(m)}\}_{m\in\mathcal{M}}\}_{i\in\cup_{k'=0}^{k-1}\mathcal{B}_{t,k'}}\big)\big].$$

## B  PROOF OF THEOREM 1

**Theorem 1.** *Let Assumptions 1–5 hold and adopt the hyperparameters of the decentralized TD in Algorithm 2 following Lemma D.4. Choose $\alpha \leq \frac{1}{4L_J}$, $T' \geq \frac{\ln M}{2\ln \sigma_W^{-1}}$. Then, the output of the decentralized AC in Algorithm 1 has the following convergence rate.*

$$\mathbb{E}\big[\|\nabla J(\omega_{\widetilde{T}})\|^2\big] \leq \frac{4R_{\max}}{T\alpha} + 4(c_4\sigma_W^{2T'} + c_5\beta^2\sigma_{W_c}^{2T'_c}) + 4c_6\Big(1 - \frac{\lambda_B}{8}\beta\Big)^{T_c} + \frac{4c_7}{N} + \frac{4c_8}{N_c} + 64C_\psi^2\zeta_{approx}^{critic}.$$

*Moreover, to achieve $\mathbb{E}\big[\|\nabla J(\omega_{\widetilde{T}})\|^2\big] \leq \epsilon$ for any $\epsilon \geq 128C_\psi^2\zeta_{approx}^{critic}$, we can choose $T,N,N_c = \mathcal{O}(\epsilon^{-1})$ and $T_c,T'_c,T' = \mathcal{O}(\ln\epsilon^{-1})$. Consequently, the overall sample complexity is $T(T_cN_c+N) = \mathcal{O}(\epsilon^{-2}\ln\epsilon^{-1})$, and the communication complexities for synchronizing linear model parameters and rewards are $T(T_c + T'_c) = \mathcal{O}(\epsilon^{-1}\ln\epsilon^{-1})$ and $TT' = \mathcal{O}(\epsilon^{-1}\ln\epsilon^{-1})$, respectively.*

*Proof.* Concatenating all the agents' actor updates in Algorithm 1, we obtain the joint actor update $\omega_{t+1} = \omega_t + \alpha\widehat{\nabla}J(\omega_t)$. Then, the item 7 of Lemma D.5 implies that

$$\begin{aligned}
J(\omega_{t+1}) &\geq J(\omega_t) + \nabla J(\omega_t)^\top(\omega_{t+1} - \omega_t) - \frac{L_J}{2}\|\omega_{t+1} - \omega_t\|^2 \\
&= J(\omega_t) + \alpha\nabla J(\omega_t)^\top\widehat{\nabla}J(\omega_t) - \frac{L_J\alpha^2}{2}\|\widehat{\nabla}J(\omega_t)\|^2 \\
&\overset{(i)}{\geq} J(\omega_t) + \alpha\|\nabla J(\omega_t)\|^2 + \alpha\nabla J(\omega_t)^\top\big(\widehat{\nabla}J(\omega_t) - \nabla J(\omega_t)\big) \\
&\quad - L_J\alpha^2\|\widehat{\nabla}J(\omega_t) - \nabla J(\omega_t)\|^2 - L_J\alpha^2\|\nabla J(\omega_t)\|^2 \\
&\overset{(ii)}{\geq} J(\omega_t) + \Big(\frac{\alpha}{2} - L_J\alpha^2\Big)\|\nabla J(\omega_t)\|^2 - \Big(\frac{\alpha}{2} + L_J\alpha^2\Big)\|\widehat{\nabla}J(\omega_t) - \nabla J(\omega_t)\|^2 \\
&\overset{(iii)}{\geq} J(\omega_t) + \frac{\alpha}{4}\|\nabla J(\omega_t)\|^2 - \alpha\|\widehat{\nabla}J(\omega_t) - \nabla J(\omega_t)\|^2
\end{aligned}$$

where (i) and (ii) use the inequalities $\|x\|^2 \leq 2\|x-y\|^2 + 2\|y\|^2$ and $x^\top y \geq -\frac{1}{2}\|x\|^2 - \frac{1}{2}\|y\|^2$ for any $x,y \in \mathbb{R}^d$, respectively, and (iii) uses the condition that $\alpha \leq \frac{1}{4L_J}$. Then, summing up the inequality above over $t = 0,1,\dots,T-1$ yields that

$$J(\omega_T) \geq J(\omega_0) + \frac{\alpha}{4}\sum_{t=0}^{T-1}\|\nabla J(\omega_t)\|^2 - \alpha\sum_{t=0}^{T-1}\|\widehat{\nabla}J(\omega_t) - \nabla J(\omega_t)\|^2.$$

Rearranging the equation above and taking expectation on both sides yields that

$$\begin{aligned}
\mathbb{E}\|\nabla J(\omega_{\widetilde{T}})\|^2 &= \frac{1}{T}\sum_{t=0}^{T-1}\mathbb{E}\|\nabla J(\omega_t)\|^2 \\
&\leq \frac{4}{T\alpha}\mathbb{E}[J(\omega_T) - J(\omega_0)] + \frac{4}{T}\sum_{t=0}^{T-1}\mathbb{E}\Big[\|\widehat{\nabla}J(\omega_t) - \nabla J(\omega_t)\|^2\Big] \\
&\overset{(i)}{\leq} \frac{4R_{\max}}{T\alpha} + 4c_4\sigma_W^{2T'} + 4c_5\beta^2\sigma_{W_c}^{2T'_c} + 4c_6\Big(1 - \frac{\lambda_B}{4}\beta\Big)^{T_c} \\
&\quad + \frac{4c_7}{N} + \frac{4c_8}{N_c} + 64C_\psi^2\zeta_{approx}^{critic},
\end{aligned} \qquad (11)$$

where (i) uses the item 4 of Lemma D.5 and eq. (39) of Lemma D.6 (The condition of Lemma D.6 that $T' \geq \frac{\ln M}{2\ln(\sigma^{-1})}$ holds). This proves the error bound of Theorem 1.

Finally, for any $\epsilon \geq 128 C_\psi^2 \zeta_{\text{approx}}^{\text{critic}}$, it can be easily verified that the following hyperparameter choices make the error bound in (11) smaller than $\epsilon$ and also satisfy the conditions of this Theorem and those in Lemma D.4 that $\beta \leq \min\left(\frac{\lambda_B}{8 C_B^2}, \frac{4}{\lambda_B}, \frac{1-\sigma}{2 C_B}\right)$, $N_c \geq \left(\frac{2}{\lambda_B} + 2\beta\right) \frac{192 C_B^2 [1 + (\kappa - 1)\rho]}{(1-\rho)\lambda_B}$.

$$\alpha = \min\left(1, \frac{1}{4 L_J}\right) = \mathcal{O}(1)$$

$$\beta = \min\left(\frac{\lambda_B}{8 C_B^2}, \frac{4}{\lambda_B}, \frac{1-\sigma}{2 C_B}\right) = \mathcal{O}(1)$$

$$T = \left\lceil \frac{48 R_{\max}}{\alpha \epsilon} \right\rceil = \mathcal{O}(\epsilon^{-1})$$

$$T' = \left\lceil \frac{1}{2 \ln(\sigma^{-1})} \max\left[\ln(48 c_4 \epsilon^{-1}), \ln M\right] \right\rceil = \mathcal{O}\left(\ln(\epsilon^{-1})\right)$$

$$T_c' = \left\lceil \frac{\ln(48 c_5 \beta^2 \epsilon^{-1})}{2 \ln(\sigma^{-1})} \right\rceil = \mathcal{O}\left(\ln(\epsilon^{-1})\right)$$

$$T_c = \left\lceil \frac{\ln(48 c_6 \epsilon^{-1})}{2 \ln[(1 - \lambda_B \beta/4)^{-1}]} \right\rceil = \mathcal{O}\left(\ln(\epsilon^{-1})\right)$$

$$N = \left\lceil \frac{48 c_7}{\epsilon} \right\rceil = \mathcal{O}(\epsilon^{-1})$$

$$N_c = \left\lceil \max\left[\frac{48 c_7}{\epsilon}, \left(\frac{2}{\lambda_B} + 2\beta\right) \frac{192 C_B^2 [1 + (\kappa - 1)\rho]}{(1-\rho)\lambda_B}\right] \right\rceil = \mathcal{O}(\epsilon^{-1}) \qquad (12)$$

$\square$

## C  Proof of Theorem 2

**Theorem 2.** *Let Assumptions 1–7 hold and adopt the hyperparameters of the decentralized TD in Algorithm 2 following Lemma D.4. Choose hyperparameters $\alpha \leq \min\left(1, \frac{\lambda_F^2}{4 L_J C_\psi^2}, \frac{C_\psi^2}{2 L_J}\right)$, $\beta \leq 1$, $T' \geq \frac{\ln M}{2 \ln \sigma_W^{-1}}$, $\eta \leq \frac{1}{2 C_\psi^2}$, $T_z \geq \frac{\ln(3 D_J C_\psi^2)}{\ln \sigma_W^{-1}}$, $K \geq \frac{\ln 3}{\ln(1 - \eta \lambda_F/2)^{-1}}$, $N \geq \frac{2304 C_\psi^4 (\kappa + 1 - \rho)}{\eta \lambda_F^5 (1-\rho)(1 - \eta \lambda_F/2)^{(K-1)/2}}$ and $N_k \propto (1 - \eta \lambda_F/2)^{-k/2}$. Then, the output of Algorithm 3 satisfies*

$$J(\omega^*) - \mathbb{E}\left[J(\omega_{\widetilde{T}})\right] \leq \frac{c_{17}}{T\alpha} + c_{18}\left(1 - \frac{\eta \lambda_F}{2}\right)^{(K-1)/4} + c_{19}\sigma_W^{T_z} + c_{20}\sigma_W^{T'} + c_{21}\beta\sigma_W^{T_c'} + \frac{c_{23}}{\sqrt{N_c}}$$

$$+ c_{22}\left(1 - \frac{\lambda_B}{8}\beta\right)^{T_c/2} + C_\psi\sqrt{c_{16}\zeta_{\text{approx}}^{\text{critic}}} + c_{24}\zeta_{\text{approx}}^{\text{critic}} + C^*\sqrt{\zeta_{\text{approx}}^{\text{actor}}}.$$

*Moreover, to achieve $J(\omega^*) - \mathbb{E}\left[J(\omega_{\widehat{T}})\right] \leq \epsilon$ for any $\epsilon \geq 2 C_\psi\sqrt{c_{16}\zeta_{\text{approx}}^{\text{critic}}} + 2 c_{24}\zeta_{\text{approx}}^{\text{critic}} + 2 C^*\sqrt{\zeta_{\text{approx}}^{\text{actor}}}$, we can choose $T = \mathcal{O}(\epsilon^{-1})$, $N, N_c = \mathcal{O}(\epsilon^{-2})$, $T_c, T_c', T', T_z, K = \mathcal{O}(\ln \epsilon^{-1})$. Consequently, the overall sample complexity is $T(T_c N_c + N) = \mathcal{O}(\epsilon^{-3} \ln \epsilon^{-1})$, and the communication complexities for synchronizing linear model parameters and rewards are $T(T_c + T_c') = \mathcal{O}(\epsilon^{-1} \ln \epsilon^{-1})$ and $T T' = \mathcal{O}(\epsilon^{-1} \ln \epsilon^{-1})$, respectively.*

*Proof.* Concatenating all the agents' actor updates in Algorithm 3, we obtain the joint actor update $\omega_{t+1} = \omega_t + \alpha h_t$. Then, the item 7 of Lemma D.5 implies that

$$J(\omega_{t+1}) \geq J(\omega_t) + \nabla J(\omega_t)^\top (\omega_{t+1} - \omega_t) - \frac{L_J}{2}\left\|\omega_{t+1} - \omega_t\right\|^2$$

$$= J(\omega_t) + \alpha \nabla J(\omega_t)^\top h_t - \frac{L_J \alpha^2}{2}\left\|h_t\right\|^2$$

$$\overset{(i)}{\geq} J(\omega_t) + \alpha \nabla J(\omega_t)^\top F(\omega_t)^{-1} \nabla J(\omega_t) + \alpha \nabla J(\omega_t)^\top [h_t - h(\omega_t)]$$

$$- L_J \alpha^2 \left\|h_t - h(\omega_t)\right\|^2 - L_J \alpha^2 \left\|F(\omega_t)^{-1} \nabla J(\omega_t)\right\|^2$$

$$\overset{(ii)}{\geq} J(\omega_t) + \left(\frac{\alpha}{C_\psi^2} - \frac{\alpha}{2C_\psi^2} - \frac{L_J\alpha^2}{\lambda_F^2}\right)\|\nabla J(\omega_t)\|^2 - \left(\frac{\alpha C_\psi^2}{2} + L_J\alpha^2\right)\big\|h_t - h(\omega_t)\big\|^2$$

$$\overset{(iii)}{\geq} J(\omega_t) + \frac{\alpha}{4C_\psi^2}\|\nabla J(\omega_t)\|^2 - \alpha C_\psi^2\big\|h_t - h(\omega_t)\big\|^2$$

where (i) uses the notation that $h(\omega_t) \overset{\triangle}{=} F(\omega_t)^{-1}\nabla J(\omega_t)$ and the inequality that $\|x\|^2 \leq 2\|x - y\|^2 + 2\|y\|^2$ for any $x, y \in \mathbb{R}^d$, (ii) uses the item 3 of Lemma D.7 and the inequality that $x^\top y \geq -\frac{1}{2C_\psi^2}\|x\|^2 - \frac{C_\psi^2}{2}\|y\|^2$ for any $x, y \in \mathbb{R}^d$, and (iii) uses the condition that $\alpha \leq \min\left(\frac{\lambda_F^2}{4L_J C_\psi^2}, \frac{C_\psi^2}{2L_J}\right)$. Taking expectation on both sides of the above inequality, summing over $t = 0, 1, \ldots, T - 1$ and rearranging, we obtain that

$$\frac{1}{T}\sum_{t=0}^{T-1}\mathbb{E}\|\nabla J(\omega_t)\|^2 \leq \frac{4C_\psi^2}{T\alpha}\mathbb{E}[J(\omega_T) - J(\omega_0)] + \frac{4C_\psi^4}{T}\sum_{t=0}^{T-1}\mathbb{E}\big\|h_t - h(\omega_t)\big\|^2$$

$$\overset{(i)}{\leq} \frac{4C_\psi^2 R_{\max}}{T\alpha} + 4C_\psi^4\Big[c_{10}\Big(1 - \frac{\eta\lambda_F}{2}\Big)^{(K-1)/2} + c_{11}\sigma^{2T_z} + c_{12}\sigma^{2T'}$$

$$+ c_{13}\beta^2\sigma^{2T_c'} + c_{14}\Big(1 - \frac{\lambda_B}{4}\beta\Big)^{T_c} + \frac{c_{15}}{N_c} + c_{16}\zeta_{\text{approx}}^{\text{critic}}\Big], \tag{13}$$

where (i) uses the item 4 of Lemma D.5 and the item 8 of Lemma D.7.

By Assumption 2, $\ln\pi_\omega(s,a)$ is an $L_\psi$-smooth function of $\omega$. Denote $\omega^* := \arg\min_{\omega\in\Omega} J(\omega)$ and denote $\mathbb{E}_{\omega^*}$ as the unconditional expectation over $s \sim \nu_{\omega^*}, a \sim \pi_{\omega^*}(\cdot|s)$. We obtain that

$$\mathbb{E}_{\omega^*}\big[\ln\pi_{t+1}(a|s) - \ln\pi_t(a|s)\big]$$

$$\geq \mathbb{E}_{\omega^*}\Big[\big(\nabla_{\omega_t}\ln\pi_t(a|s)\big)^\top(\omega_{t+1} - \omega_t)\Big] - \frac{L_\psi}{2}\mathbb{E}\|\omega_{t+1} - \omega_t\|^2$$

$$= \alpha\mathbb{E}_{\omega^*}\big[\psi_t(a|s)^\top h_t\big] - \frac{L_\psi\alpha^2}{2}\mathbb{E}\big[\|h_t\|^2\big]$$

$$\overset{(i)}{\geq} \alpha\mathbb{E}_{\omega^*}\big[\psi_t(a|s)^\top\big(h_t - h(\omega_t)\big)\big] + \alpha\mathbb{E}_{\omega^*}\big[\psi_t(a|s)^\top h(\omega_t) - A_{\omega_t}(s,a)\big] + \alpha\mathbb{E}_{\omega^*}\big[A_{\omega_t}(s,a)\big]$$

$$- L_\psi\alpha^2\mathbb{E}\big[\big\|h_t - h(\omega_t)\big\|^2\big] - L_\psi\alpha^2\mathbb{E}\big[\big\|F(\omega_t)^{-1}\nabla J(\omega_t)\big\|^2\big]$$

$$\overset{(ii)}{\geq} -\alpha C_\psi\sqrt{\mathbb{E}\big[\big\|h_t - h(\omega_t)\big\|^2\big]} - \alpha C_*\sqrt{\zeta_{\text{approx}}^{\text{actor}}}$$

$$+ \alpha\mathbb{E}\big[J(\omega^*) - J(\omega_t)\big] - L_\psi\alpha^2\mathbb{E}\big[\big\|h_t - h(\omega_t)\big\|^2\big] - L_\psi\alpha^2\lambda_F^{-2}\mathbb{E}\big[\big\|\nabla J(\omega_t)\big\|^2\big],$$

where (i) uses the inequality that $\|x\|^2 \leq 2\|x - y\|^2 + 2\|y\|^2$ for any $x, y \in \mathbb{R}^d$ and the notation that $h(\omega_t) \overset{\triangle}{=} F(\omega_t)^{-1}\nabla J(\omega_t)$, (ii) uses Cauchy-Schwarz inequality, the items 3 & 6 of Lemma D.7, the inequality that $\mathbb{E}\|X\| \leq \sqrt{\mathbb{E}\big[\|X\|^2\big]}$ for any random vector $X$ and the equality that $\mathbb{E}_{\omega^*}\big[A_{\omega_t}(s,a)\big] = \mathbb{E}\big[J(\omega^*) - J(\omega_t)\big]$ (See its proof in Lemma 3.2 of (1).). Averaging the inequality above over $t = 0, 1, \ldots, T - 1$ and rearranging it yields that

$$J(\omega^*) - \mathbb{E}\big[J(\omega_{\widetilde{T}})\big] = \frac{1}{T}\sum_{t=0}^{T-1}\mathbb{E}\big[J(\omega_t)\big]$$

$$\leq \frac{1}{T\alpha}\mathbb{E}_{\omega^*}\big[\ln\pi_T(a|s) - \ln\pi_0(a|s)\big] + C_*\sqrt{\zeta_{\text{approx}}^{\text{actor}}} + \frac{C_\psi}{T}\sum_{t=0}^{T-1}\sqrt{\mathbb{E}\big[\big\|h_t - h(\omega_t)\big\|^2\big]}$$

$$+ \frac{L_\psi\alpha}{T}\sum_{t=0}^{T-1}\mathbb{E}\big[\big\|h_t - h(\omega_t)\big\|^2\big] + \frac{L_\psi\alpha}{T\lambda_F^2}\sum_{t=0}^{T-1}\mathbb{E}\big[\big\|\nabla J(\omega_t)\big\|^2\big]$$

$$\overset{(i)}{\leq} \frac{1}{T\alpha}\mathbb{E}_{s\sim\nu_{\omega^*}}\big[\text{KL}\big(\pi_{\omega^*}(\cdot|s)\|\pi_0(\cdot|s)\big) - \text{KL}\big(\pi_{\omega^*}(\cdot|s)\|\pi_T(\cdot|s)\big)\big] + C^*\sqrt{\zeta_{\text{approx}}^{\text{actor}}}$$

$$+ C_\psi\Big[c_{10}\Big(1 - \frac{\eta\lambda_F}{2}\Big)^{(K-1)/2} + c_{11}\sigma^{2T_z} + c_{12}\sigma^{2T'} + c_{13}\beta^2\sigma^{2T_c'}$$

$$+ c_{14}\Big(1 - \frac{\lambda_B}{4}\beta\Big)^{T_c} + \frac{c_{15}}{N_c} + c_{16}\zeta_{\text{approx}}^{\text{critic}}\Big]^{1/2}$$

$$+ L_\psi \alpha \Big[ c_{10}\Big(1 - \frac{\eta\lambda_F}{2}\Big)^{(K-1)/2} + c_{11}\sigma^{2T_z} + c_{12}\sigma^{2T'} + c_{13}\beta^2\sigma^{2T'_c}$$

$$+ c_{14}\Big(1 - \frac{\lambda_B}{4}\beta\Big)^{T_c} + \frac{c_{15}}{N_c} + c_{16}\zeta_{\text{approx}}^{\text{critic}}\Big]$$

$$+ \frac{L_\psi \alpha}{\lambda_F^2}\Big\{ \frac{4C_\psi^2 R_{\max}}{T\alpha} + 4C_\psi^4\Big[ c_{10}\Big(1 - \frac{\eta\lambda_F}{2}\Big)^{(K-1)/2} + c_{11}\sigma^{2T_z} + c_{12}\sigma^{2T'} + c_{13}\beta^2\sigma^{2T'_c}$$

$$+ c_{14}\Big(1 - \frac{\lambda_B}{4}\beta\Big)^{T_c} + \frac{c_{15}}{N_c} + c_{16}\zeta_{\text{approx}}^{\text{critic}}\Big]\Big\}$$

$$\overset{(ii)}{\leq} \frac{1}{T\alpha}\mathbb{E}_{s\sim\nu_{\omega^*}}\Big[ \text{KL}\big(\pi_{\omega^*}(\cdot|s)||\pi_0(\cdot|s)\big)\Big] + C^*\sqrt{\zeta_{\text{approx}}^{\text{actor}}}$$

$$+ C_\psi\Big[ \sqrt{c_{10}}\Big(1 - \frac{\eta\lambda_F}{2}\Big)^{(K-1)/4} + \sqrt{c_{11}}\sigma^{T_z} + \sqrt{c_{12}}\sigma^{T'} + \sqrt{c_{13}}\beta\sigma^{T'_c}$$

$$+ \sqrt{c_{14}}\Big(1 - \frac{\lambda_B}{4}\beta\Big)^{T_c/2} + \sqrt{\frac{c_{15}}{N_c}} + \sqrt{c_{16}\zeta_{\text{approx}}^{\text{critic}}}\Big]$$

$$+ L_\psi\Big(1 + \frac{4C_\psi^4}{\lambda_F^2}\Big)\Big[ c_{10}\Big(1 - \frac{\eta\lambda_F}{2}\Big)^{(K-1)/4} + c_{11}\sigma^{T_z} + c_{12}\sigma^{T'} + c_{13}\beta\sigma^{T'_c}$$

$$+ c_{14}\Big(1 - \frac{\lambda_B}{4}\beta\Big)^{T_c/2} + \frac{c_{15}}{\sqrt{N_c}} + c_{16}\zeta_{\text{approx}}^{\text{critic}}\Big] + \frac{4L_\psi C_\psi^2 R_{\max}}{T\alpha\lambda_F^2}$$

$$\overset{(iii)}{=} \frac{c_{17}}{T\alpha} + c_{18}\Big(1 - \frac{\eta\lambda_F}{2}\Big)^{(K-1)/4} + c_{19}\sigma^{T_z} + c_{20}\sigma^{T'} + c_{21}\beta\sigma^{T'_c} + c_{22}\Big(1 - \frac{\lambda_B}{4}\beta\Big)^{T_c/2}$$

$$+ \frac{c_{23}}{\sqrt{N_c}} + C_\psi\sqrt{c_{16}\zeta_{\text{approx}}^{\text{critic}}} + c_{24}\zeta_{\text{approx}}^{\text{critic}} + C^*\sqrt{\zeta_{\text{approx}}^{\text{actor}}}, \tag{14}$$

where (i) uses the definition of KL divergence that $\text{KL}\big(\pi_{\omega^*}(\cdot|s)||\pi_\omega(\cdot|s)\big) = \mathbb{E}_{a\sim\pi_{\omega^*}(\cdot|s)}\big[\ln\pi_{\omega^*}(a|s) - \ln\pi_\omega(a|s)\big|s\big]$ and eqs. (13) & (54), (ii) uses the condition that $\alpha \leq 1$ and the inequality that $\sqrt{\sum_{i=1}^n x_i} \leq \sum_{i=1}^n \sqrt{x_i}$ for any $n \in \mathbb{N}^+$ and $x_1,\ldots,x_n \geq 0$, (iii) uses the notations that $c_{17} := \mathbb{E}_{s\sim\nu_{\omega^*}}\big[\text{KL}\big(\pi_{\omega^*}(\cdot|s)||\pi_0(\cdot|s)\big)\big] + \frac{4L_\psi C_\psi^2 R_{\max}}{\lambda_F^2}$, $c_{18} := C_\psi\sqrt{c_{10}} + c_{10}L_\psi\Big(1 + \frac{4C_\psi^4}{\lambda_F^2}\Big)$, $c_{19} := C_\psi\sqrt{c_{11}} + c_{11}L_\psi\Big(1 + \frac{4C_\psi^4}{\lambda_F^2}\Big)$, $c_{20} := C_\psi\sqrt{c_{12}} + c_{12}L_\psi\Big(1 + \frac{4C_\psi^4}{\lambda_F^2}\Big)$, $c_{21} := C_\psi\sqrt{c_{13}} + c_{13}L_\psi\Big(1 + \frac{4C_\psi^4}{\lambda_F^2}\Big)$, $c_{22} := C_\psi\sqrt{c_{14}} + c_{14}L_\psi\Big(1 + \frac{4C_\psi^4}{\lambda_F^2}\Big)$, $c_{23} := C_\psi\sqrt{c_{15}} + c_{15}L_\psi\Big(1 + \frac{4C_\psi^4}{\lambda_F^2}\Big)$, $c_{24} := c_{16}L_\psi\Big(1 + \frac{4C_\psi^4}{\lambda_F^2}\Big)$. This proves the error bound of Theorem 2.

Finally, for any $\epsilon \geq 2C_\psi\sqrt{c_{16}\zeta_{\text{approx}}^{\text{critic}}} + 2c_{24}\zeta_{\text{approx}}^{\text{critic}} + 2C^*\sqrt{\zeta_{\text{approx}}^{\text{actor}}}$, it can be verified that the following hyperparameter choices make the error bound in (14) smaller than $\epsilon$ and satisfy all the conditions of this Theorem and those in Lemma D.4 that $\beta \leq \min\big(\frac{\lambda_B}{8C_B^2}, \frac{4}{\lambda_B}, \frac{1-\sigma}{2C_B}\big)$, $N_c \geq \big(\frac{2}{\lambda_B} + 2\beta\big)\frac{192C_B^2[1+(\kappa-1)\rho]}{(1-\rho)\lambda_B}$.

$$\alpha = \min\Big(1, \frac{\lambda_F^2}{4L_J C_\psi^2}, \frac{C_\psi^2}{2L_J}\Big) = \mathcal{O}(1)$$

$$\beta = \min\Big(1, \frac{\lambda_B}{8C_B^2}, \frac{4}{\lambda_B}, \frac{1-\sigma}{2C_B}\Big) = \mathcal{O}(1)$$

$$\eta = \frac{1}{2C_\psi^2} = \mathcal{O}(1)$$

$$T = \Big\lceil \frac{14c_{17}}{\alpha\epsilon} \Big\rceil = \mathcal{O}(\epsilon^{-1})$$

$$K = \left\lceil \max \left[ \frac{\ln 3}{\ln[(1 - \eta\lambda_F/2)^{-1}]}, \frac{4\ln(14c_{18}\epsilon^{-1})}{\ln\left[(1 - \eta\lambda_F/2)^{-1}\right]} + 1 \right] \right\rceil = \mathcal{O}\left[\ln(\epsilon^{-1})\right]$$

$$T_z = \left\lceil \max \left[ \frac{\ln(3D_J C_\psi^2)}{\ln(\sigma^{-1})}, \frac{\ln(14c_{19}\epsilon^{-1})}{\ln(\sigma^{-1})} \right] \right\rceil = \left\lceil \mathcal{O}\left[\ln(\epsilon^{-1})\right]$$

$$T' = \left\lceil \max \left[ \frac{\ln M}{2\ln(\sigma^{-1})}, \frac{\ln(14c_{20}\epsilon^{-1})}{\ln(\sigma^{-1})} \right] \right\rceil = \mathcal{O}\left[\ln(\epsilon^{-1})\right]$$

$$T'_c = \left\lceil \frac{\ln(14c_{21}\epsilon^{-1})}{\ln(\sigma^{-1})} \right\rceil = \mathcal{O}\left[\ln(\epsilon^{-1})\right]$$

$$T_c = \left\lceil \frac{2\ln(14c_{22}\epsilon^{-1})}{\ln[(1 - \lambda_B\beta/4)^{-1}]} \right\rceil = \mathcal{O}\left[\ln(\epsilon^{-1})\right]$$

$$N = \left\lceil \frac{2304 C_\psi^4(\kappa + 1 - \rho)}{\eta\lambda_F^5(1 - \rho)(1 - \eta\lambda_F/2)^{(K-1)/2}} \right\rceil = \mathcal{O}(\epsilon^{-2})$$

$$N_c = \left\lceil \max \left[ \left(\frac{2}{\lambda_B} + 2\beta\right) \frac{192 C_B^2[1 + (\kappa - 1)\rho]}{(1 - \rho)\lambda_B}, 196c_{23}^2\epsilon^{-2} \right] \right\rceil = \mathcal{O}(\epsilon^{-2}) \tag{15}$$

$\square$

## D  SUPPORTING LEMMAS

First, we extend the Lemma F.3 of (12) to the Lemma D.1 below. The item 1 of Lemma D.1 generalizes the case $n = 1$ to any $n \in \mathbb{N}^+$, the items 2 & 3 remain unchanged, and the item 4 is added for convenience of our convergence analysis.

**Lemma D.1.** *The doubly stochastic matrix $W$ and the difference matrix $\Delta = I - \frac{1}{M}\mathbf{1}\mathbf{1}^\top$ have the following properties:*

1. $\Delta W^n = W^n\Delta = W^n - \frac{1}{M}\mathbf{1}\mathbf{1}^\top$ *for any $n \in \mathbb{N}^+$.*

2. *The spectral norm of $W$ satisfies $\|W\| = 1$.*

3. *For any $x \in \mathbb{R}^M$ and $n \in \mathbb{N}^+$, $\|W^n\Delta x\| \leq \sigma_W^n\|\Delta x\|$ ($\sigma_W$ is the second largest singular value of $W$). Hence, for any $H \in \mathbb{R}^{M \times M}$, $\|W^n\Delta H\|_F \leq \sigma_W^n\|\Delta H\|_F$.*

4. $\left\|W^n - \frac{1}{M}\mathbf{1}\mathbf{1}^\top\right\| \leq \sigma_W^n$, $\left\|W^n - \frac{1}{M}\mathbf{1}\mathbf{1}^\top\right\|_F \leq \sigma_W^n\sqrt{M}$ *for any $n \in \mathbb{N}^+$.*

*Proof.* The proof of items 2 & 3 can be found in (12). We prove the item 1 and item 4.

We prove item 1 by induction. The case $n = 1$ of the item 1 can be proved by the following two equalities, as shown in (12).

$$\Delta W = \left(I - \frac{1}{M}\mathbf{1}\mathbf{1}^\top\right)W = W - \frac{1}{M}\mathbf{1}\mathbf{1}^\top W = W - \frac{1}{M}\mathbf{1}\mathbf{1}^\top$$

$$W\Delta = W\left(I - \frac{1}{M}\mathbf{1}\mathbf{1}^\top\right) = W - \frac{1}{M}W\mathbf{1}\mathbf{1}^\top = W - \frac{1}{M}\mathbf{1}\mathbf{1}^\top$$

Suppose the case of $n = k$ holds for a certain $k \in \mathbb{N}^+$, then the following two equalities proves the case of $n = k + 1$ and thus proves the item 1.

$$\Delta W^{k+1} = (\Delta W^k)W = \left(W^k - \frac{1}{M}\mathbf{1}\mathbf{1}^\top\right)W = W^{k+1} - \frac{1}{M}\mathbf{1}\mathbf{1}^\top$$

$$W^{k+1}\Delta = W(W^k\Delta) = W\left(W^k - \frac{1}{M}\mathbf{1}\mathbf{1}^\top\right) = W^{k+1} - \frac{1}{M}\mathbf{1}\mathbf{1}^\top$$

The item 4 can be proved by the following two inequalities.

$$\left\|W^n - \frac{1}{M}\mathbf{1}\mathbf{1}^\top\right\| \overset{(i)}{=} \|W^n\Delta\| = \sup_{x:\|x\|\leq 1} \|W^n\Delta x\| \overset{(ii)}{\leq} \sup_{x:\|x\|\leq 1} \sigma_W^n\|\Delta\|\|x\| \overset{(iv)}{=} \sigma_W^n, \tag{16}$$

$$\left\|W^n - \frac{1}{M}\mathbf{1}\mathbf{1}^\top\right\|_F \overset{(i)}{=} \left\|W^n\Delta\right\|_F \overset{(iii)}{\leq} \sigma_W^n\|\Delta\|_F$$

$$\overset{(iv)}{=} \sigma_W^n\sqrt{M\left(1 - \frac{1}{M}\right)^2 + M(M-1)\left(-\frac{1}{M}\right)^2} \leq \sigma_W^n\sqrt{M}, \quad (17)$$

where (i) uses the item 1, (ii) and (iii) use the item 3 ($H = I$ in (iii)), and (iv) uses the fact that $\Delta$ has $M$ diagnoal entries $1 - \frac{1}{M}$ and $M(M-1)$ off-diagnoal entries $-\frac{1}{M}$, which implies that $\|\Delta\| = 1$. □

Next, we extend the Lemma F.2. of (12) to the Lemma D.2 below.

**Lemma D.2.** *Suppose the Markovian samples $\{s_i, a_i\}_{i\geq 0}$ are generated following the policy $\pi_\omega$ and transition kernel $\mathcal{P}'$ (can be $\mathcal{P}$ or $\mathcal{P}_\xi$), and $s'_{i+1} \sim \mathcal{P}(\cdot|s_i, a_i)$. Then, for any deterministic mapping $X : \mathcal{S} \times \mathcal{A} \times \mathcal{S} \times \mathcal{S} \to \mathbb{R}^{p\times q}$ ($p, q \in \mathbb{N}^+$ are arbitrary.) such that $\|X(s, a, s', \widetilde{s})\|_F \leq C_x$ and for any $s, s', \widetilde{s} \in \mathcal{S}, a \in \mathcal{A}$, we have*

$$\mathbb{E}\left[\left\|\frac{1}{n}\sum_{i=n'}^{n+n'-1} X(s_i, a_i, s_{i+1}, s'_{i+1}) - \overline{X}\right\|_F^2 \Big| s_{n'}\right] \leq \frac{9C_x^2(\kappa + 1 - \rho)}{n(1-\rho)}, \forall n, n' \in \mathbb{N}^+ \quad (18)$$

*where $\overline{X} = \mathbb{E}\left[X(s_i, a_i, s_{i+1}, s'_{i+1})|s_i\right]$ with $s_i \sim \mu_\omega$ (or $\nu_\omega$) when $\mathcal{P}' = \mathcal{P}$ (or $\mathcal{P}_\xi$).*

*Proof.* Denote $Y(s, a, s') := \mathbb{E}_{\widetilde{s}\sim\mathcal{P}'(\cdot|s,a)}\left[X(s, a, s', \widetilde{s})|s, a, s'\right]$ which satisfies $\|Y(s, a, s')\| \leq C_x$ and $\mathbb{E}_{s_i\sim\nu_\omega}\left[Y(s_i, a_i, s_{i+1})\right] = \overline{X}$. Hence, Lemma F.2 of (12) can be applied to $Y(s, a, s')$ and obtain the following inequality.

$$\mathbb{E}\left[\left\|\frac{1}{n}\sum_{i=n'}^{n+n'-1} Y(s_i, a_i, s_{i+1}) - \overline{X}\right\|_F^2 \Big| s_{n'}\right] \leq \frac{8C_x^2(\kappa + 1 - \rho)}{n(1-\rho)}. \quad (19)$$

Therefore, we obtain that

$$\mathbb{E}\left[\left\|\frac{1}{n}\sum_{i=n'}^{n+n'-1} X(s_i, a_i, s_{i+1}, s'_{i+1}) - \overline{X}\right\|_F^2 \Big| \{s_i, a_i, s_{i+1}\}_{i=n'}^{n+n'-1}\right]$$

$$= \left\|\mathbb{E}\left[\frac{1}{n}\sum_{i=n'}^{n+n'} X(s_i, a_i, s_{i+1}, s'_{i+1}) - \overline{X}\Big| \{s_i, a_i, s_{i+1}\}_{i=n'}^{n+n'-1}\right]\right\|_F^2$$

$$+ \mathrm{Var}\left[\frac{1}{n}\sum_{i=n'}^{n+n'-1} X(s_i, a_i, s_{i+1}, s'_{i+1})\Big| \{s_i, a_i, s_{i+1}\}_{i=n'}^{n+n'-1}\right]$$

$$\overset{(i)}{=} \left\|\frac{1}{n}\sum_{i=n'}^{n+n'-1} Y(s_i, a_i, s_{i+1}) - \overline{X}\right\|_F^2$$

$$+ \frac{1}{n^2}\sum_{i=n'}^{n+n'-1} \mathrm{Var}\left[X(s_i, a_i, s_{i+1}, s'_{i+1})\Big| \{s_i, a_i, s_{i+1}\}_{i=n'}^{n+n'-1}\right]$$

$$\overset{(ii)}{\leq} \left\|\frac{1}{n}\sum_{i=n'}^{n+n'-1} Y(s_i, a_i, s_{i+1}) - \overline{X}\right\|_F^2 + \frac{C_x^2}{n} \quad (20)$$

where (i) uses the conditional independency among $\{s'_{i+1}\}_{i=tN}^{(t+1)N-1}$ on $\{s_i, a_i, s_{i+1}\}_{i=n'}^{n+n'-1}$ and (ii) uses the fact that $\|X(s_i, a_i, s_{i+1}, s'_{i+1})\|_F \leq C_x$.

Finally, eq. (18) can be proved via the following inequality.

$$\mathbb{E}\left[\left\|\frac{1}{n}\sum_{i=n'}^{n+n'} X(s_i, a_i, s_{i+1}, s'_{i+1}) - \overline{X}\right\|_F^2 \Big| s_{n'}\right]$$

$$\overset{(i)}{\leq} \mathbb{E}\Big[\Big\|\frac{1}{n}\sum_{i=n'}^{n+n'-1} Y(s_i, a_i, s_{i+1}) - \overline{X}\Big\|_F^2 \Big| s_{n'}\Big] + \frac{C_x^2}{n}$$

$$\overset{(ii)}{\leq} \frac{8C_x^2(\kappa + 1 - \rho)}{n(1-\rho)} + \frac{C_x^2}{n} \leq \frac{9C_x^2(\kappa + 1 - \rho)}{n(1-\rho)},$$

where (i) takes the conditional expectation of eq. (20) on $s_n'$ and (ii) uses eq. (19). □

Next, we prove the following Lemmas D.3 & D.4 on the decentralized TD in Algorithm 2. We first define the following useful notations.

$\lambda_\phi := \lambda_{\min}\big(\mathbb{E}_{s\sim\mu_\omega}[\phi(s)\phi(s)^\top]\big) > 0$, see Assumption 4.

$B(s, s') := \phi(s)\big[\gamma\phi(s') - \phi(s)\big]^\top$.

$B_t := \frac{1}{N_c}\sum_{i=tN_c}^{(t+1)N_c-1} B(s_i, s_{i+1})$.

$B_\omega := \mathbb{E}_{s\sim\mu_\omega, a\sim\pi_\omega(\cdot|s), s'\sim\mathcal{P}(\cdot|s,a)}\big[B(s, s')\big]$.

$b^{(m)}(s, a, s') := R^{(m)}(s, a, s')\phi(s)$.

$b(s, a, s') := \frac{1}{M}\sum_{m=1}^M b^{(m)}(s, a, s')$.

$b_t^{(m)} := \frac{1}{N_c}\sum_{i=tN_c}^{(t+1)N_c-1} b^{(m)}(s_i, a_i, s_{i+1})$.

$b_t := \frac{1}{M}\sum_{m=1}^M b_t^{(m)}$.

$b_\omega := \mathbb{E}_{s\sim\mu_\omega, a\sim\pi_\omega(\cdot|s), s'\sim\mathcal{P}(\cdot|s,a)}\big[b(s, a, s')\big]$.

$\theta_\omega^* := B_\omega^{-1}b_\omega$, which is the optimal critic parameter under policy $\pi_\omega$.

**Lemma D.3.** *The following bounds hold for Algorithm 2.*

1. $\|B(s, s')\|_F, \|B_t\|_F, \|B_\omega\|_F \leq C_B := 1 + \gamma$,
   $\|b^{(m)}(s, a, s')\|, \|b(s, a, s')\|, \|b_t^{(m)}\|, \|b_t\|, \|b_\omega\| \leq C_b := R_{\max}$.

2. $\theta^\top B_\omega\theta \leq -\frac{\lambda_B}{2}\|\theta\|^2$ *uniformly for all* $\omega$, *where* $\lambda_B := 2(1-\gamma)\lambda_\phi > 0$.

3. $\|\theta_\omega^*\| \leq R_\theta := \frac{2C_b}{\lambda_B}$ *uniformly for all* $\omega$.

*Proof.* We first prove the item 1. Notice that for any vectors $x, y \in \mathbb{R}^d$,

$$\|xy^\top\|_F = \sqrt{\sum_{i=1}^d\sum_{j=1}^d (x_iy_j)^2} = \sqrt{\sum_{i=1}^d x_i^2}\sqrt{\sum_{j=1}^d y_j^2} = \|x\|\|y\|.$$

Hence, we obtain that

$$\|B(s, s')\|_F = \big\|\phi(s)\big(\gamma\phi(s') - \phi(s)\big)^\top\big\|_F = \|\phi(s)\|\|\gamma\phi(s') - \phi(s)\| \leq 1 + \gamma := C_B, \quad (21)$$

$$\|b(s, a, s')\| = \overline{R}(s, a, s')\|\phi(s)\| \leq R_{\max} := C_b. \quad (22)$$

The other terms listed in the item 1 can be proved by applying the Jensen's inequality to the convex function $\|\cdot\|$.

Next, we prove the item 2, where we use the underlying distribution that $s \sim \mu_\omega$, $a \sim \pi_\omega(\cdot|s)$, $s' \sim \mathcal{P}(\cdot|s, a)$. We obtain that

$$\theta^\top B_\omega\theta = \mathbb{E}_\omega\Big(\theta^\top\phi(s)\big[\gamma\phi(s') - \phi(s)\big]^\top\theta\Big)$$

$$= \gamma\mathbb{E}_\omega\Big[\big(\theta^\top\phi(s)\big)\big(\theta^\top\phi(s')\big)\Big] - \mathbb{E}_\omega\Big[\big(\theta^\top\phi(s)\big)^2\Big]$$

$$\leq \frac{\gamma}{2}\Big(\mathbb{E}_\omega\Big[\big(\theta^\top\phi(s)\big)^2\Big] + \mathbb{E}_\omega\Big[\big(\theta^\top\phi(s')\big)^2\Big]\Big) - \mathbb{E}_\omega\Big[\big(\theta^\top\phi(s)\big)^2\Big]$$

$$\overset{(i)}{=} (\gamma - 1)\mathbb{E}_\omega\Big[\big(\theta^\top\phi(s)\big)^2\Big]$$

$$= -(1-\gamma)\theta^\top\mathbb{E}_\omega[\phi(s)\phi(s)^\top]\theta$$

$$\overset{(ii)}{\leq} -\frac{\lambda_B}{2}\|\theta\|^2, \tag{23}$$

where (i) uses the fact that $s, s' \sim \mu_\omega$ which is the stationary state distribution with the transition kernel $\mathcal{P}$ and the policy $\pi_\omega$, and (ii) uses Assumption 4 and we denote $\lambda_B := 2(1-\gamma)\lambda_\phi > 0$.

Finally, the item 3 can be proved via the following inequality.

$$\|\theta_\omega^*\|^2 \overset{(i)}{\leq} -\frac{2}{\lambda_B}(\theta_\omega^*)^\top B_\omega \theta_\omega^* \leq \frac{2}{\lambda_B}\|\theta_\omega^*\|\|B_\omega\theta_\omega^*\| = \frac{2}{\lambda_B}\|\theta_\omega^*\|\|b_\omega\| \leq \frac{2C_b}{\lambda_B}\|\theta_\omega^*\|, \tag{24}$$

where (i) uses the item 2. $\qquad\square$

**Lemma D.4.** *Under Assumptions 1–5 and choosing* $\beta \leq \min\big(\frac{\lambda_B}{8C_B^2}, \frac{4}{\lambda_B}, \frac{1-\sigma_W}{2C_B}\big)$, $N_c \geq \big(\frac{2}{\lambda_B} + 2\beta\big)\frac{192C_B^2[1+(\kappa-1)\rho]}{(1-\rho)\lambda_B}$, *Algorithm 2 has the following convergence rate.*

$$\sum_{m=1}^M \mathbb{E}\big[\big\|\theta_{T_c+T_c'}^{(m)} - \theta_{\omega_t}^*\big\|^2\big|\omega_t\big] \leq \sigma_W^{2T_c'}\beta^2 c_2 + 2M\Big[c_3\Big(1 - \frac{\lambda_B}{8}\beta\Big)^{T_c} + \frac{c_1}{N_c}\Big]. \tag{25}$$

*Moreover, to achieve* $\sum_{m=1}^M \mathbb{E}\big[\big\|\theta_{T_c+T_c'}^{(m)} - \theta_{\omega_t}^*\big\|^2\big|\omega_t\big] \leq \epsilon$, *we can choose* $T_c, T_c' = \mathcal{O}\big[\ln(\epsilon^{-1})\big]$ *and* $N_c = \mathcal{O}(\epsilon^{-1})$. *Consequently, the sample complexity is* $T_c N_c = \mathcal{O}\big[\epsilon^{-1}\ln(\epsilon^{-1})\big]$ *and the communication complexity is* $T_c + T_c' = \mathcal{O}\big[\ln(\epsilon^{-1})\big]$.

*Proof.* In Algorithm 2, by averaging the TD update rule (26) over the agents $m \in \mathcal{M}$, we obtain that the averaged critic parameter $\bar{\theta}_{t,t'} := \frac{1}{M}\sum_{m=1}^M \bar{\theta}_{t,t'}^{(m)}$ follows the following update rule

$$\bar{\theta}_{t,t'+1} = \frac{1}{M}\sum_{m=1}^M \Big[\sum_{m'=1}^M W_{m,m'}\theta_{t,t'}^{(m')} + \beta\big(B_{t'}\theta_{t,t'}^{(m)} + b_{t'}^{(m)}\big)\Big]$$

$$= \frac{1}{M}\sum_{m'=1}^M \theta_{t,t'}^{(m')} + \beta\frac{1}{M}\sum_{m=1}^M \big(B_{t'}\theta_{t,t'}^{(m)} + b_{t'}^{(m)}\big)$$

$$= \bar{\theta}_{t,t'} + \beta\big(B_{t'}\bar{\theta}_{t,t'} + b_{t'}\big) \tag{26}$$

which can be viewed as a centralized TD update using the Markovian samples $\{s_i, a_i\}_i$ from the transition kernel $\mathcal{P}$ and the joint policy $\pi_t$. Therefore, Theorem 4 in (60) can be directly applied to analyze this centralized TD update and obtain the following convergence rate of $\bar{\theta}_{t,t'}$, since all the conditions of that theorem are met [4].

$$\mathbb{E}\big[\big\|\bar{\theta}_{t,T_c} - \theta_{\omega_t}^*\big\|^2\big|\omega_t\big] \leq \Big(1 - \frac{\lambda_B}{4}\beta\Big)^{T_c}\mathbb{E}\big[\big\|\bar{\theta}_{t,0} - \theta_{\omega_t}^*\big\|^2\big|\omega_t\big]$$

$$+ \Big(\frac{2}{\lambda_B} + 2\beta\Big)\frac{192\big(C_B^2 R_\theta^2 + C_b^2\big)[1 + (\kappa-1)\rho]}{(1-\rho)\lambda_B N_c}$$

$$\overset{(i)}{\leq} 2\Big(1 - \frac{\lambda_B}{4}\beta\Big)^{T_c}\big(\|\theta_{-1}\|^2 + R_\theta^2\big) + \frac{c_1}{N_c}$$

$$\overset{(ii)}{\leq} c_3\Big(1 - \frac{\lambda_B}{4}\beta\Big)^{T_c} + \frac{c_1}{N_c}. \tag{27}$$

where (i) uses the condition that $\beta \leq 4/\lambda_B$, the item 3 of Lemma D.3 and the constant that $c_1 := \frac{1920\big(C_B^2 R_\theta^2 + C_b^2\big)[1+(\kappa-1)\rho]}{(1-\rho)\lambda_B^2}$, (ii) uses the constant that $c_3 := 2\big(\|\theta_{-1}\|^2 + R_\theta^2\big)$.

---

[4]We corrected the typo $1 - \frac{\lambda_B}{8}\beta$, which should be $1 - \frac{\lambda_B}{4}\beta$.

Next, we consider the consensus error $\|\Delta\Theta_{t,t'}\|_F^2 = \sum_{m=1}^M \|\theta_{t,t'}^{(m)} - \overline{\theta}_{t,t'}\|^2$ where we define $\Theta_{t,t'} := [\theta_{t,t'}^{(1)}, \ldots, \theta_{t,t'}^{(M)}]^\top$. Note that the critic-step (26) can be rewritten into the following matrix form

$$\Theta_{t,t'+1} = W\Theta_{t,t'} + \beta(\Theta_{t,t'}B_{t'}^\top + [b_{t'}^{(1)}; \ldots; b_{t'}^{(M)}]^\top); t' = 0, 1, \ldots, T_c - 1, \tag{28}$$

which further implies that for any $t' = 0, 1, \ldots, T_c - 1$,

$$\|\Delta\Theta_{t,t'+1}\|_F \overset{(i)}{\leq} \|W\Delta\Theta_{t,t'}\|_F + \beta\|\Delta\Theta_{t,t'}B_{t'}^\top\|_F + \beta\|\Delta[b_{t'}^{(1)}; \ldots; b_{t'}^{(M)}]^\top\|_F$$

$$\overset{(ii)}{\leq} (\sigma_W + \beta C_B)\|\Delta\Theta_{t,t'}\|_F + \beta\sqrt{M\sum_{m=1}^M \|b_{t'}^{(m)}\|^2}$$

$$\overset{(iii)}{\leq} \frac{1 + \sigma_W}{2}\|\Delta\Theta_{t,t'}\|_F + \beta M C_b,$$

where (i) uses the item 1 of Lemma D.1, (ii) uses the item 3 of Lemma D.1 and the item 1 of Lemma D.3, (iii) uses the condition that $\beta \leq \frac{1-\sigma_W}{2C_B}$ and the item 1 of Lemma D.3. Telescoping the inequality above yields that

$$\|\Delta\Theta_{t,T_c}\|_F \leq \left(\frac{1+\sigma_W}{2}\right)^{T_c}\|\Delta\Theta_{t,0}\|_F + \frac{2\beta M C_b}{1-\sigma_W} \overset{(i)}{=} \frac{2\beta M C_b}{1-\sigma_W}, \tag{29}$$

where (i) uses the equality that $\Delta\Theta_0 = O$ due to the initial condition that $\Theta_{t,0} = [\theta_{-1}; \ldots; \theta_{-1}]^\top$.

On the other hand, the final $T_c'$ local average steps in Algorithm 2 can be rewritten into the following matrix form

$$\Theta_{t,t'+1} = W\Theta_{t,t'}; t = T_c, T_c + 1, \ldots, T_c + T_c' - 1.$$

Hence, the average critic parameter $\overline{\theta}_{t,t'}$ does not change in these local average steps, i.e.,

$$\overline{\theta}_{t,T_c+T_c'} = \frac{1}{M}\Theta_{t,T_c+T_c'}^\top \mathbf{1} = \frac{1}{M}\Theta_{t,T_c}^\top (W^{T_c'})^\top \mathbf{1} = \frac{1}{M}\Theta_{t,T_c}^\top \mathbf{1} = \overline{\theta}_{t,T_c}. \tag{30}$$

Therefore, we obtain that

$$\sum_{m=1}^M \|\theta_{t,T_c+T_c'}^{(m)} - \overline{\theta}_{t,T_c}\|^2 = \sum_{m=1}^M \|\theta_{t,T_c+T_c'}^{(m)} - \overline{\theta}_{t,T_c+T_c'}\|^2 = \|\Delta\Theta_{t,T_c+T_c'}\|_F^2 = \|\Delta W^{T_c'}\Theta_{t,T_c}\|_F^2$$

$$\overset{(i)}{=} \|W^{T_c'}\Delta\Theta_{t,T_c}\|_F^2 \overset{(ii)}{\leq} \sigma_W^{2T_c'}\|\Delta\Theta_{t,T_c}\|_F^2$$

$$\overset{(iii)}{\leq} \sigma_W^{2T_c'}\left(\frac{2\beta M C_b}{1-\sigma_W}\right)^2 \overset{(iv)}{=} \sigma_W^{2T_c'}\beta^2 c_2/2 \tag{31}$$

where (i) and (ii) use the items 1 and 3 of Lemma D.1 respectively, (iii) uses eq. (29), (iv) denotes that $c_2 := 2\left(\frac{2MC_b}{1-\sigma_W}\right)^2$. Combining eqs. (27) & (31) yields that

$$\sum_{m=1}^M \mathbb{E}\left[\|\theta_{t,T_c+T_c'}^{(m)} - \theta_{\omega_t}^*\|^2 | \omega_t\right] \leq 2\sum_{m=1}^M \mathbb{E}\left[\|\theta_{t,T_c+T_c'}^{(m)} - \overline{\theta}_{t,T_c}\|^2 | \omega_t\right] + 2M\mathbb{E}\left[\|\overline{\theta}_{t,T_c} - \theta_{\omega_t}^*\|^2 | \omega_t\right]$$

$$\leq \sigma_W^{2T_c'}\beta^2 c_2 + 2M\left[c_3\left(1 - \frac{\lambda_B}{4}\beta\right)^{T_c} + \frac{c_1}{N_c}\right].$$

In the inequality above, replacing $\theta_{t,T_c+T_c'}^{(m)}$ from Algorithm 2 by its corresponding variable $\theta_t^{(m)}$ from Algorithm 1 proves eq. (25). Finally, it can be easily verified that the following hyperparameter choices make the error bound in (25) smaller than $\epsilon$ and also satisfy the conditions of Lemma D.4.

$$\beta = \min\left(\frac{\lambda_B}{8C_B^2}, \frac{4}{\lambda_B}, \frac{1-\sigma_W}{2C_B}\right) = \mathcal{O}(1)$$

$$N_c = \max\left[\left(\frac{2}{\lambda_B} + 2\beta\right)\frac{192C_B^2[1 + (\kappa-1)\rho]}{(1-\rho)\lambda_B}, 6Mc_1\epsilon^{-1}\right] = \mathcal{O}(\epsilon^{-1})$$

$$T_c = \left\lceil \frac{\ln(6Mc_3\epsilon^{-1})}{\ln\left[(1-\lambda_B\beta/4)^{-1}\right]} \right\rceil = \mathcal{O}\left[\ln(\epsilon^{-1})\right]$$

$$T_c' = 2\left\lceil \frac{\ln(3\beta^2 c_2\epsilon^{-1})}{\ln(\sigma_W^{-1})} \right\rceil = \mathcal{O}\left[\ln(\epsilon^{-1})\right]$$

$\square$

**Lemma D.5.** *For any $\omega, \widetilde{\omega} \in \Omega$, $s \in \mathcal{S}$ and $a^{(m)} \in \mathcal{A}_m$ ($\mathcal{A}_m$ denotes the action space for the agent m), the following properties hold.*

1. $\|\psi_\omega^{(m)}(a^{(m)}|s)\| \leq C_\psi$, *where* $\psi_\omega^{(m)}(a^{(m)}|s) := \nabla_{\omega^{(m)}} \ln \pi_\omega^{(m)}(a^{(m)}|s)$.

2. $\|\psi_{\widetilde{\omega}}^{(m)}(a^{(m)}|s) - \psi_\omega^{(m)}(a^{(m)}|s)\| \leq L_\psi\|\widetilde{\omega}^{(m)} - \omega^{(m)}\|$.

3. $d_{TV}\left[\pi_{\widetilde{\omega}^{(m)}}^{(m)}(\cdot|s), \pi_{\omega^{(m)}}^{(m)}(\cdot|s)\right] \leq L_\pi\|\widetilde{\omega}^{(m)} - \omega^{(m)}\|$.

4. $0 \leq V_\omega(s), Q_\omega(s,a) \leq (1-\gamma)R_{\max}$, $0 \leq J(\omega) \leq R_{\max}$.

5. $d_{TV}\left[\nu_\omega(\cdot|s), \nu_{\widetilde{\omega}}(\cdot|s)\right] \leq L_\nu\|\omega' - \omega\|$ *where* $L_\nu := L_\pi[1 + \log_\rho(\kappa^{-1}) + (1-\rho)^{-1}]$.

6. $d_{TV}\left[Q_{\widetilde{\omega}}(s,a), Q_\omega(s,a)\right] \leq L_Q\|\widetilde{\omega} - \omega\|$ *where* $L_Q := \frac{2R_{\max}L_\nu}{1-\gamma}$.

7. $J(\omega)$ *is $L_J$-smooth where* $L_J := R_{\max}(4L_\nu + L_\psi)/(1-\gamma)$.

8. $\|\nabla J(\omega)\| \leq D_J := \frac{C_\psi R_{\max}}{1-\gamma}$.

9. $F(\omega)$ *is $L_F$-Lipschitz where* $L_F := 2C_\psi(L_\pi C_\psi + L_\nu C_\psi + L_\psi)$.

10. $h(\omega)$ *is $L_h$-Lipschitz where* $L_h := 2\lambda_F^{-1}(D_J\lambda_F^{-1}L_F + L_J)$.

*Proof.* For any $\omega^{(m)}, \widetilde{\omega}^{(m)} \in \Omega_m$, $s \in \mathcal{S}$ and $a^{(m)} \in \mathcal{A}_m$, arbitrarily select $\omega^{(m')} = \widetilde{\omega}^{(m')} \in \Omega_{m'}$, $a^{(m')} \in \mathcal{A}_{m'}$ for every $m' \in \{1, ..., M\}/\{m\}$. Denote $\omega = [\omega^{(1)}; \ldots; \omega^{(M)}]$, $\widetilde{\omega} = [\widetilde{\omega}^{(1)}; \ldots; \widetilde{\omega}^{(M)}]$, $a = [a^{(1)}, \ldots, a^{(M)}]$. Notice that the joint score vector has the following decomposition

$$\psi_\omega(a|s) = [\psi_\omega^{(1)}(a^{(1)}|s); \ldots; \psi_\omega^{(M)}(a^{(M)}|s)]. \tag{32}$$

Hence, the items 1 & 2 can be proved via the following two inequalities, respectively.

$$\|\psi_\omega^{(m)}(a^{(m)}|s)\| \leq \sqrt{\sum_{m'=1}^M \|\psi_\omega^{(m')}(a^{(m')}|s)\|^2} \overset{(i)}{=} \|\psi_\omega(a|s)\| \overset{(ii)}{\leq} C_\psi.$$

$$\|\psi_{\widetilde{\omega}}^{(m)}(a^{(m)}|s) - \psi_\omega^{(m)}(a^{(m)}|s)\| = \|\psi_{\widetilde{\omega}}(a|s) - \psi_\omega(a|s)\|$$
$$\overset{(i)}{\leq} L_\psi\|\widetilde{\omega} - \omega\| = L_\psi\|\widetilde{\omega}^{(m)} - \omega^{(m)}\|$$

where (i) uses Assumption 2.

Next, we prove the item 3. Notice that

$$d_{TV}\left[\pi_{\widetilde{\omega}}(\cdot|s), \pi_\omega(\cdot|s)\right]$$
$$\overset{(i)}{=} \sup_{A \subset \mathcal{A}} |\pi_{\widetilde{\omega}}(A|s) - \pi_\omega(A|s)|$$
$$\overset{(ii)}{\geq} \sup_{A_1 \subset \mathcal{A}_1, ..., A_M \subset \mathcal{A}_M} \left| \prod_{m'=1}^M \pi_{\widetilde{\omega}^{(m')}}(A_{m'}|s) - \prod_{m'=1}^M \pi_{\omega^{(m')}}(A_{m'}|s) \right|$$
$$\overset{(iii)}{=} \sup_{A_1 \subset \mathcal{A}_1, ..., A_M \subset \mathcal{A}_M} \left| \prod_{m'=1, m'\neq m}^M \pi_{\omega^{(m')}}(A_{m'}|s) \right| \left| \pi_{\widetilde{\omega}^{(m)}}(A_m|s) - \pi_{\omega^{(m)}}(A_m|s) \right|$$

$$\overset{(iv)}{=} \sup_{A_m \subset \mathcal{A}_m} \left| \pi_{\widetilde{\omega}^{(m)}}(A_m|s) - \pi_{\omega^{(m)}}(A_m|s) \right| = d_{\text{TV}}\big[ \pi_{\widetilde{\omega}^{(m)}}^{(m)}(\cdot|s), \pi_{\omega^{(m)}}^{(m)}(\cdot|s)\big],$$

where (i) denotes that $\pi_\omega(A|s) = \int_A \pi_\omega(a|s)da$, (ii) uses the relation that $\times_{m \in \mathcal{M}} A_m \subset \mathcal{A}$, (iii) uses our construction that $\omega^{(m')} = \widetilde{\omega}^{(m')} \in \Omega_{m'}, \forall m' \in \{1, ..., M\}/\{m\}$, and (iv) uses $A_{m'} = \mathcal{A}_{m'}$ to achieve the supremum. Therefore, the item 2 can be proved via the following inequality.

$$d_{\text{TV}}\big[\pi_{\widetilde{\omega}^{(m)}}^{(m)}(\cdot|s), \pi_{\omega^{(m)}}^{(m)}(\cdot|s)\big] = d_{\text{TV}}\big[\pi_{\widetilde{\omega}}(\cdot|s), \pi_\omega(\cdot|s)\big] \overset{\leq}{\underset{\widetilde{L}_\pi}{}} \|\widetilde{\omega} - \omega\| = L_\pi \|\widetilde{\omega}^{(m)} - \omega^{(m)}\|,$$

where (i) uses Assumption 2.

The item 4 can be proved by the following three inequalities that use Assumption 3.

$$0 \le V_\omega(s) = \mathbb{E}_\omega\Big[\sum_{t=0}^\infty \gamma^t \overline{R}_t\Big|s_0 = s\Big] \le \sum_{t=0}^\infty \gamma^t R_{\max} = \frac{R_{\max}}{1-\gamma},$$

$$0 \le Q_\omega(s,a) = \mathbb{E}_{s' \sim \mathcal{P}(\cdot|s,a)}[\overline{R}(s,a,s') + \gamma V_\omega(s')] \le R_{\max} + \gamma\frac{R_{\max}}{1-\gamma} = \frac{R_{\max}}{1-\gamma},$$

$$0 \le J(\omega) = (1-\gamma)\mathbb{E}_\omega\Big[\sum_{t=0}^\infty \gamma^t \overline{R}_t\Big] \le (1-\gamma)\sum_{t=0}^\infty \gamma^t R_{\max} = R_{\max}.$$

The proof of the items $5 - 7$ can be found in the proof of Lemma 3, Lemma 4 and Proposition 1 of (60), respectively.

Next, the item 8 is proved by the following inequality.

$$\big\|\nabla J(\omega)\big\| = \big\|\mathbb{E}_{s \sim \nu_\omega, a \sim \pi_\omega(\cdot|s)}\big[Q_\omega(s,a)\psi_\omega(a|s)\big]\big\|$$
$$\overset{(i)}{\le} \mathbb{E}_{s \sim \nu_\omega, a \sim \pi_\omega(\cdot|s)}\big[|Q_\omega(s,a)|\big\|\psi_\omega(a|s)\big\|\big] \overset{(ii)}{\le} \frac{C_\psi R_{\max}}{1-\gamma},$$

where (i) applies Jensen's inequality, (ii) uses Assumption 2 and the item 4.

Next, the item 9 is proved by the following inequality.

$$\big\|F(\widetilde{\omega}) - F(\omega)\big\|$$
$$= \big\|\mathbb{E}_{s \sim \nu_{\pi_{\widetilde{\omega}}}, a \sim \pi_{\widetilde{\omega}}(\cdot|s)}\big[\psi_{\widetilde{\omega}}(a|s)\psi_{\widetilde{\omega}}(a|s)^\top\big] - \mathbb{E}_{s \sim \nu_{\pi_\omega}, a \sim \pi_\omega(\cdot|s)}\big[\psi_\omega(a|s)\psi_\omega(a|s)^\top\big]\big\|$$
$$\overset{(i)}{\le} \big\|\mathbb{E}_{s \sim \nu_{\pi_{\widetilde{\omega}}}, a \sim \pi_{\widetilde{\omega}}(\cdot|s)}\big[\psi_{\widetilde{\omega}}(a|s)\psi_{\widetilde{\omega}}(a|s)^\top\big] - \mathbb{E}_{s \sim \nu_{\pi_\omega}, a \sim \pi_\omega(\cdot|s)}\big[\psi_{\widetilde{\omega}}(a|s)\psi_{\widetilde{\omega}}(a|s)^\top\big]\big\|$$
$$+ \mathbb{E}_{s \sim \nu_{\pi_\omega}, a \sim \pi_\omega(\cdot|s)}\big[\big\|[\psi_{\widetilde{\omega}}(a|s) - \psi_\omega(a|s)]\psi_{\widetilde{\omega}}(a|s)^\top\big\|\big]$$
$$+ \mathbb{E}_{s \sim \nu_{\pi_\omega}, a \sim \pi_\omega(\cdot|s)}\big[\big\|\psi_\omega(a|s)[\psi_{\widetilde{\omega}}(a|s) - \psi_\omega(a|s)]^\top\big\|\big]$$
$$\overset{(ii)}{\le} \Big\|\int_{\mathcal{S} \times \mathcal{A}}[\nu_{\widetilde{\omega}}(s)\pi_{\widetilde{\omega}}(a|s) - \nu_\omega(s)\pi_\omega(a|s)]\big[\psi_{\widetilde{\omega}}(a|s)\psi_{\widetilde{\omega}}(a|s)^\top\big]dsda\Big\| + 2C_\psi L_\psi\|\widetilde{\omega} - \omega\|$$
$$\le C_\psi^2 \int_{\mathcal{S} \times \mathcal{A}}|\nu_{\widetilde{\omega}}(s)\pi_{\widetilde{\omega}}(a|s) - \nu_\omega(s)\pi_\omega(a|s)|dsda + 2C_\psi L_\psi\|\widetilde{\omega} - \omega\|$$
$$\le C_\psi^2 \int_{\mathcal{S} \times \mathcal{A}}\nu_{\widetilde{\omega}}(s)|\pi_{\widetilde{\omega}}(a|s) - \pi_\omega(a|s)|dsda$$
$$+ C_\psi^2 \int_{\mathcal{S} \times \mathcal{A}}\pi_\omega(a|s)|\nu_{\widetilde{\omega}}(s) - \nu_\omega(s)|dsda + 2C_\psi L_\psi\big\|\widetilde{\omega} - \omega\big\|$$
$$\overset{(iii)}{\le} 2L_\pi C_\psi^2\big\|\widetilde{\omega} - \omega\big\| + 2L_\nu C_\psi^2\big\|\widetilde{\omega} - \omega\big\| + 2C_\psi L_\psi\big\|\widetilde{\omega} - \omega\big\| := L_F\big\|\widetilde{\omega} - \omega\big\|$$

where (i) applies triangle inequality and then Jensen's inequality to the norm $\|\cdot\|$, (ii) uses Assumption 2, (iii) uses the equality that $\int_{\mathcal{S}} \nu_\omega(s)ds = \int_{\mathcal{A}} \pi_\omega(a|s)da = 1$ as well as the inequlities that $\int_{\mathcal{A}}|\pi_{\widetilde{\omega}}(a|s) - \pi_\omega(a|s)|da = 2d_{\text{TV}}\big[\pi_{\widetilde{\omega}}(\cdot|s), \pi_\omega(\cdot|s)\big] \le 2L_\pi\|\widetilde{\omega} - \omega\|$ (based on Assumption 2) and that $\int_{\mathcal{S}}|\nu_{\widetilde{\omega}}(s) - \nu_\omega(s)|ds = 2d_{\text{TV}}\big[\nu_\omega(\cdot|s), \nu_{\widetilde{\omega}}(\cdot|s)\big] \le 2L_\nu\|\omega' - \omega\|$ (based on the item 5).

Finally, the item 10 is proved by the following inequality

$$
\begin{aligned}
&\big\| h(\widetilde{\omega}) - h(\omega) \big\| \\
&= \big\| F(\widetilde{\omega})^{-1} \nabla J(\widetilde{\omega}) - F(\omega)^{-1} \nabla J(\omega) \big\| \\
&\leq 2 \big\| [F(\widetilde{\omega})^{-1} - F(\omega)^{-1}] \nabla J(\widetilde{\omega}) \big\| + 2 \big\| F(\omega)^{-1} [\nabla J(\widetilde{\omega}) - \nabla J(\omega)] \big\| \\
&\overset{(i)}{\leq} 2 D_J \big\| F(\omega)^{-1} [F(\omega) - F(\widetilde{\omega})] F(\widetilde{\omega})^{-1} \big\| + 2 L_J \big\| F(\omega)^{-1} \big\| \big\| \widetilde{\omega} - \omega \big\| \\
&\overset{(ii)}{\leq} 2 D_J \lambda_F^{-2} L_F \big\| \widetilde{\omega} - \omega \big\| + 2 L_J \lambda_F^{-1} \big\| \widetilde{\omega} - \omega \big\| := L_h \big\| \widetilde{\omega} - \omega \big\|,
\end{aligned}
$$

where (i) uses the items 7 & 8, and (ii) uses the inequality that $\| F(\omega)^{-1} \| = \lambda_{\max}(F(\omega)^{-1}) = \lambda_{\min}[F(\omega)]^{-1} \leq \lambda_F^{-1}$ for all $\omega$ (since $F(\omega)$ and $F(\omega)^{-1}$ are positive definite) and the item 9. $\quad\square$

Next, we bound the approximation error of the following stochastic (partial) policy gradients.

$$
\widehat{\nabla}_{\omega^{(m)}} J(\omega_t) := \frac{1}{N} \sum_{i=tN}^{(t+1)N-1} \big[ \overline{R}_i^{(m)} + \gamma \phi(s'_{i+1})^\top \theta_t^{(m)} - \phi(s_i)^\top \theta_t^{(m)} \big] \psi_t^{(m)}(a_i^{(m)} | s_i), \quad (33)
$$

$$
\widehat{\nabla} J(\omega_t) := \big[ \widehat{\nabla}_{\omega^{(1)}} J(\omega_t); \ldots; \widehat{\nabla}_{\omega^{(M)}} J(\omega_t) \big], \quad (34)
$$

$$
\widehat{\nabla}_{\omega^{(m)}} J(\omega_t; \mathcal{B}_{t,k}) := \frac{1}{N_k} \sum_{i \in \mathcal{B}_{t,k}} \big[ \overline{R}_i^{(m)} + \gamma \phi(s'_{i+1})^\top \theta_t^{(m)} - \phi(s_i)^\top \theta_t^{(m)} \big] \psi_t^{(m)}(a_i^{(m)} | s_i), \quad (35)
$$

$$
\widehat{\nabla} J(\omega_t; \mathcal{B}_{t,k}) := \big[ \widehat{\nabla}_{\omega^{(1)}} J(\omega_t); \ldots; \widehat{\nabla}_{\omega^{(M)}} J(\omega_t) \big]. \quad (36)
$$

**Lemma D.6.** *Let Assumptions 1-5 hold and adopt the hyperparameters of the decentralized TD in Algorithm 2 following Lemma D.4. Choose $T' \geq \frac{\ln M}{2 \ln(\sigma_W^{-1})}$. Then, the following properties hold.*

1. *The estimated average reward $\overline{R}_i^{(m)}$ has the following bias and variance bound.*

$$
\sum_{m=1}^M \mathbb{E}\big[ \overline{R}_i^{(m)} - \overline{R}_i \big| R_i \big]^2 \leq M \sigma_W^{2T'} R_{\max}^2, \quad (37)
$$

$$
\sum_{m=1}^M Var\big[ \overline{R}_i^{(m)} \big| R_i \big] \leq 4 R_{\max}^2 \overline{\sigma}^2, \quad (38)
$$

*where $R_i := [R_i^{(1)}; \ldots; R_i^{(M)}]$ denotes the joint reward.*

2. *The stochastic policy gradients have the following error bound.*

$$
\mathbb{E}\big[ \big\| \widehat{\nabla} J(\omega_t) - \nabla J(\omega_t) \big\|^2 \big] \leq c_4 \sigma_W^{2T'} + c_5 \beta^2 \sigma_W^{2T'_c} + c_6 \Big( 1 - \frac{\lambda_B}{8} \beta \Big)^{T_c}
$$
$$
+ \frac{c_7}{N} + \frac{c_8}{N_c} + 16 C_\psi^2 \zeta_{approx}^{critic} \quad (39)
$$

$$
\mathbb{E}\big[ \big\| \widehat{\nabla} J(\omega_t; \mathcal{B}_{t,k}) - \nabla J(\omega_t) \big\|^2 \big| \mathcal{F}_{t,k} \big] \leq c_4 \sigma_W^{2T'} + 16 C_\psi^2 \sum_{m=1}^M \big\| \theta_t^{(m)} - \theta_{\omega_t}^* \big\|^2
$$
$$
+ \frac{c_7}{N_k} + 16 C_\psi^2 \zeta_{approx}^{critic}, \quad (40)
$$

*where $\mathcal{F}_{t,k} := \sigma\big[ \mathcal{F}_t \cup \sigma\big( \{ s_i, a_i, s_{i+1}, s'_{i+1}, \{e_i^{(m)}\}_{m \in \mathcal{M}} \}_{i \in \cup_{k'=0}^{k-1} \mathcal{B}_{t,k'}} \big) \big]$.*

*Proof.* We will first prove the item 1.

When $R_i := [R_i^{(1)}; \ldots; R_i^{(M)}]$ is given and fixed, the randomness of $\widetilde{R}_i^{(m)} := R_i^{(m)}(1 + e_i^{(m)})$ and $\widehat{\nabla}_{\omega^{(m)}} J(\omega_t)$ defined in eq. (4) only comes from the noises $\{e_i^{(m)}\}_{m=1}^M$. Since $\{e_i^{(m)}\}_{m=1}^M$ are independent noises with zero mean and variances $\sigma_1^2, \ldots, \sigma_M^2$, $\widetilde{R}_i := [\widetilde{R}_i^{(1)}; \ldots; \widetilde{R}_i^{(M)}]$ has the following moments

$$
\mathbb{E}\big[ \widetilde{R}_i | R_i \big] = R_i,
$$

$$\text{cov}\big[\widetilde{R}_i|R_i\big] = \text{diag}\big[(R_i^{(1)})^2\sigma_1^2,\ldots,(R_i^{(M)})^2\sigma_M^2\big] := \Sigma_i.$$

Hence, $\widehat{R}_i := [\overline{R}_i^{(1)},\ldots,\overline{R}_i^{(m)}]^\top = W^{T'}\widetilde{R}_i$ (the second "=" comes from eq. (3) and the notations that $\widetilde{R}_i^{(m)} := \widehat{R}_{i,0}^{(m)}$ and that $\widehat{R}_i^{(m)} := \widehat{R}_{i,T'}^{(m)}$) has the moment that $\mathbb{E}\big[\widehat{R}_i|R_i\big] = W^{T'}R_i$ and $\text{Cov}\big[\widehat{R}_i|R_i\big] = W^{T'}\Sigma_i(W^{T'})^\top$. Therefore, eq. (37) can be proved as follows

$$\sum_{m=1}^{M}\mathbb{E}\big[\overline{R}_i^{(m)} - \overline{R}_i|R_i\big]^2 = \left\|\mathbb{E}\big[\widehat{R}_i - \overline{R}_i\mathbf{1}|R_i\big]\right\|^2 = \left\|W^{T'}R_i - \frac{1}{M}\mathbf{1}\mathbf{1}^\top R_i\right\|^2$$

$$\leq \left\|W^{T'} - \frac{1}{M}\mathbf{1}\mathbf{1}^\top\right\|^2\|R_i\|^2 \overset{(i)}{\leq} M\sigma_W^{2T'}R_{\max}^2,$$

where $\mathbf{1}$ is a $M$-dim vector of 1's, (i) uses the inequality that $\|R_i\|^2 = \sum_{m=1}^{M}(R_i^{(m)})^2 \leq MR_{\max}^2$ (based on Assumption 3) and the item 4 of Lemma D.1. Then, eq. (38) can be proved as follows

$$\sum_{m=1}^{M}\text{var}\big[\overline{R}_i^{(m)}|R_i\big] = \text{Var}\big[\widehat{R}_i|R_i\big] = \text{tr}\big[(W^{T'})^\top\Sigma_i W^{T'}\big]$$

$$= \text{tr}\left[\left(W^{T'} - \frac{1}{M}\mathbf{1}\mathbf{1}^\top\right)\Sigma_i\left(W^{T'} - \frac{1}{M}\mathbf{1}\mathbf{1}^\top\right)^\top\right] + \text{tr}\left[(W^{T'})\Sigma_i\left(\frac{1}{M}\mathbf{1}\mathbf{1}^\top\right)\right]$$

$$+ \text{tr}\left[\left(\frac{1}{M}\mathbf{1}\mathbf{1}^\top\right)\Sigma_i(W^{T'})^\top\right] + \text{tr}\left[\left(\frac{1}{M}\mathbf{1}\mathbf{1}^\top\right)\Sigma_i\left(\frac{1}{M}\mathbf{1}\mathbf{1}^\top\right)\right]$$

$$\overset{(i)}{\leq} MR_{\max}^2\overline{\sigma}^2\left\|W^{T'} - \frac{1}{M}\mathbf{1}\mathbf{1}^\top\right\|^2 + \frac{2}{M}\text{tr}\big[W^{T'}\Sigma_i\mathbf{1}\mathbf{1}^\top\big] + \frac{1}{M^2}\text{tr}[\mathbf{1}(\mathbf{1}^\top\Sigma_i\mathbf{1})\mathbf{1}^\top]$$

$$\overset{(ii)}{\leq} MR_{\max}^2\overline{\sigma}^2\sigma_W^{2T'} + \frac{2}{M}\mathbf{1}^\top\Sigma_i W^{T'}\mathbf{1} + \frac{1}{M^2}(\mathbf{1}^\top\Sigma_i\mathbf{1})\text{tr}[\mathbf{1}^\top\mathbf{1}]$$

$$\overset{(iii)}{\leq} R_{\max}^2\overline{\sigma}^2 + \frac{3}{M}\mathbf{1}^\top\Sigma_i\mathbf{1}$$

$$= R_{\max}^2\overline{\sigma}^2 + \frac{3}{M}\sum_{m=1}^{M}(R_i^{(m)})^2\sigma_m^2$$

$$\overset{(iv)}{\leq} 4R_{\max}^2\overline{\sigma}^2,$$

where (i) uses the equality that $\text{tr}(Y^\top) = \text{tr}(Y)$ and the inequality (41) below in which $X = W^{T'} - \frac{1}{M}\mathbf{1}\mathbf{1}^\top$ and the $m$-th entry of $v_m \in \mathbb{R}^M$ is 1 while its other entries are 0, (ii) uses the item 4 of Lemma D.1 and the equality that $\text{tr}(xy^\top) = y^\top x$ for any $x,y \in \mathbb{R}^M$, (iii) uses the condition that $T' \geq [\ln M]/[2\ln(\sigma_W^{-1})]$ and the item 1 of Lemma D.1, (iv) uses Assumption 3.

$$\text{tr}(X\Sigma_i X^\top) = \text{tr}(X^\top X\Sigma_i) = \sum_{m=1}^{M}v_m^\top X^\top X\Sigma_i v_m \leq \sum_{m=1}^{M}\|v_m\|\|X\|^2\|\Sigma_i v_m\|$$

$$= \sum_{m=1}^{M}(R_i^{(m)})^2\sigma_m^2\|X\|^2 \leq MR_{\max}^2\overline{\sigma}^2\|X\|^2. \tag{41}$$

Next, we will prove eq. (39) in the item 2, where the error term can be decomposed as follows

$$\left\|\widehat{\nabla}J(\omega_t) - \nabla J(\omega_t)\right\|^2 \leq 4\underbrace{\left\|\widehat{\nabla}J(\omega_t) - g_t\right\|^2}_{(I)} + 4\underbrace{\left\|g_t - g_t^*\right\|^2}_{(II)}$$

$$+ 4\underbrace{\left\|g_t^* - \overline{g}_t^*\right\|^2}_{(III)} + 4\underbrace{\left\|\overline{g}_t^* - \nabla J(\omega_t)\right\|^2}_{(IV)}, \tag{42}$$

where we use the following notations that

$$g_t := [g_t^{(1)};\ldots;g_t^{(M)}], \tag{43}$$

$$g_t^{(m)} := \frac{1}{N} \sum_{i=tN}^{(t+1)N-1} \big[ \overline{R}_i + \gamma \phi(s'_{i+1})^\top \theta_t^{(m)} - \phi(s_i)^\top \theta_t^{(m)} \big] \psi_t^{(m)}(a_i^{(m)}|s_i), \tag{44}$$

$$g_t^* := \frac{1}{N} \sum_{i=tN}^{(t+1)N-1} \big[ \overline{R}_i + \gamma \phi(s'_{i+1})^\top \theta_{\omega_t}^* - \phi(s_i)^\top \theta_{\omega_t}^* \big] \psi_t(a_i|s_i), \tag{45}$$

$$\overline{g}_t^* := \mathbb{E}_{s\sim\nu_{\omega_t}, a\sim\pi_t(\cdot|s), s'\sim\mathcal{P}(\cdot|s,a)} \big[ \overline{R}(s,a,s') + \gamma \phi(s')^\top \theta_{\omega_t}^* - \phi(s)^\top \theta_{\omega_t}^* \big] \psi_t(a|s) \big| \omega_t \big]. \tag{46}$$

Conditioned on the following filtration

$$\begin{aligned}
\mathcal{F}_t' :=& \sigma\big[ \mathcal{F}_t \cup \sigma\big( \{s_i, a_i, s'_{i+1}\}_{i=tN+1}^{(t+1)N-1} \big) \big] \\
=& \sigma\big( \{\theta_{t'}^{(m)}\}_{m\in\mathcal{M}, 0\le t'\le t} \cup \{s_i, a_i, s'_{i+1}\}_{i=0}^{(t+1)N-1} \cup \{s_{(t+1)N}\} \cup \{\{e_i^{(m)}\}_{m\in\mathcal{M}}\}_{i=0}^{tN-1} \big),
\end{aligned}$$

the error term (I) can be bounded as follows.

$$\mathbb{E}\Big[ \big\| \widehat{\nabla} J(\omega_t) - g_{t,k} \big\|^2 \Big| \mathcal{F}_t' \Big]$$

$$= \mathbb{E}\Big[ \sum_{m=1}^{M} \big\| \widehat{\nabla}_{\omega^{(m)}} J(\omega_t) - g_{t,k}^{(m)} \big\|^2 \Big| \mathcal{F}_t' \Big]$$

$$\overset{(i)}{=} \sum_{m=1}^{M} \mathbb{E}\Big[ \big\| \frac{1}{N} \sum_{i=tN}^{(t+1)N-1} (\overline{R}_i^{(m)} - \overline{R}_i) \psi_t^{(m)}(a_i^{(m)}|s_i) \big\|^2 \Big| \mathcal{F}_t' \Big]$$

$$\overset{(ii)}{\le} \sum_{m=1}^{M} \big\| \mathbb{E}\Big[ \frac{1}{N} \sum_{i=tN}^{(t+1)N-1} (\overline{R}_i^{(m)} - \overline{R}_i) \psi_t^{(m)}(a_i^{(m)}|s_i) \Big| \mathcal{F}_t' \Big] \big\|^2$$

$$+ \sum_{m=1}^{M} \text{Var}\Big[ \frac{1}{N} \sum_{i=tN}^{(t+1)N-1} (\overline{R}_i^{(m)} - \overline{R}_i) \psi_t^{(m)}(a_i^{(m)}|s_i) \Big| \mathcal{F}_t' \Big]$$

$$\overset{(iii)}{\le} \sum_{m=1}^{M} \big\| \mathbb{E}\Big[ \frac{1}{N} \sum_{i=tN}^{(t+1)N-1} (\overline{R}_i^{(m)} - \overline{R}_i) \Big| \mathcal{F}_t' \Big] \psi_t^{(m)}(a_i^{(m)}|s_i) \big\|^2$$

$$+ \frac{1}{N^2} \sum_{m=1}^{M} \sum_{i=tN}^{(t+1)N-1} \text{Var}\big[ (\overline{R}_i^{(m)} - \overline{R}_i) \psi_t^{(m)}(a_i^{(m)}|s_i) \big| \mathcal{F}_t' \big]$$

$$\overset{(iv)}{\le} \sum_{m=1}^{M} \Big[ \frac{1}{N} \sum_{i=tN}^{(t+1)N-1} \mathbb{E}(\overline{R}_i^{(m)} - \overline{R}_i | \mathcal{F}_t') \Big]^2 \big\| \psi_t^{(m)}(a_i^{(m)}|s_i) \big\|^2$$

$$+ \frac{1}{N^2} \sum_{m=1}^{M} \sum_{i=tN}^{(t+1)N-1} \big\| \psi_t^{(m)}(a_i^{(m)}|s_i) \big\|^2 \text{var}\big[ \overline{R}_i^{(m)} - \overline{R}_i | \mathcal{F}_t' \big]$$

$$\overset{(v)}{\le} \frac{C_\psi^2}{N} \sum_{m=1}^{M} \sum_{i=tN}^{(t+1)N-1} \big[ \mathbb{E}(\overline{R}_i^{(m)} - \overline{R}_i | \mathcal{F}_t') \big]^2 + \frac{C_\psi^2}{N^2} \sum_{i=tN}^{(t+1)N-1} \sum_{m=1}^{M} \text{var}\big[ \overline{R}_i^{(m)} | \mathcal{F}_t' \big]$$

$$\overset{(vi)}{\le} C_\psi^2 (M\sigma_W^{2T'} R_{\max}^2) + \frac{C_\psi^2}{N} (4 R_{\max}^2 \overline{\sigma}^2)$$

$$= C_\psi^2 R_{\max}^2 \Big( M\sigma_W^{2T'} + \frac{4}{N} \overline{\sigma}^2 \Big), \tag{47}$$

where (i) uses the definitions of $\widehat{\nabla}_{\omega^{(m)}} J(\omega_t)$ and $g_t^{(m)}$ defined in eqs. (33) & (44) respectively, (ii) uses the relation that $\mathbb{E}\|X\|^2 = \text{Var}(X) + \|\mathbb{E}X\|^2$ for any random vector $X$, (iii) uses the facts that $\psi_t^{(m)}(a_i^{(m)}|s_i), \overline{R}_i \in \mathcal{F}_t'$ are fixed while $\{\overline{R}_i^{(m)}\}_{i=tN}^{(t+1)N-1}$ are random and independent given $\mathcal{F}_t'$, (iv) uses the equality that $\text{Var}(xY) = \sum_{j=1}^{d} \text{var}(xy_j) = \sum_{j=1}^{d} y_j^2 \text{var}(x) = \|y\|^2 \text{var}(x)$ for any random scalar $x$ and fixed vector $Y = [y_1, \ldots, y_d] \in \mathbb{R}^d$ (Here we denote $y = \psi_t^{(m)}(a_i^{(m)}|s_i) \in \mathcal{F}_t'$), (v) applies Jensen's inequality to the convex function $(\cdot)^2$ and uses the item 1 of Lemma D.5 as well as

the fact that $\overline{R}_i \in \mathcal{F}'_t$ is fixed, (vi) uses eqs. (37) & (38) and the fact that the conditional distribution of $\overline{R}_i^{(m)}$ on $R_i \in \mathcal{F}'_t$ is the same as that on $\mathcal{F}'_t$ since the noise $e_i^{(m)}$ is independent from any other variables.

Then we bound the error term (II) of eq. (42) as follows.

$$
\begin{aligned}
\left\|g_t - g_t^*\right\|^2 &= \sum_{m=1}^M \left\|\frac{1}{N} \sum_{i=tN}^{(t+1)N-1} \left([\gamma\phi(s'_{i+1}) - \phi(s_i)]^\top (\theta_t^{(m)} - \theta_{\omega_t}^*)\right)\psi_t^{(m)}(a_i^{(m)}|s_i)\right\|^2 \\
&\overset{(i)}{\leq} \frac{1}{N} \sum_{i=tN}^{(t+1)N-1} \sum_{m=1}^M \left\|\gamma\phi(s'_{i+1}) - \phi(s_i)\right\|^2 \left\|\theta_t^{(m)} - \theta_{\omega_t}^*\right\|^2 \left\|\psi_t^{(m)}(a_i^{(m)}|s_i)\right\|^2 \\
&\overset{(ii)}{\leq} \frac{C_\psi^2(1+\gamma)^2}{N} \sum_{i=tN}^{(t+1)N-1} \sum_{m=1}^M \left\|\theta_t^{(m)} - \theta_{\omega_t}^*\right\|^2 \\
&= 4C_\psi^2 \sum_{m=1}^M \left\|\theta_t^{(m)} - \theta_{\omega_t}^*\right\|^2,
\end{aligned}
\tag{48}
$$

where (i) applies Jensen's inequality to the convex function $\|\cdot\|^2$, (ii) uses Assumption 4 and the item 1 of Lemma D.5.

To bound the error term (III) of eq. (42), denote that

$$
X(s,a,s',\widetilde{s}) = \left[\overline{R}(s,a,\widetilde{s}) + \gamma\phi(\widetilde{s})^\top \theta_{\omega_t}^* - \phi(s)^\top \theta_{\omega_t}^*\right]\psi_t(a|s),
\tag{49}
$$

which satisfies $\|X(s,a,s',\widetilde{s})\| \leq \left[|\overline{R}(s,a,\widetilde{s})| + \|\gamma\phi(\widetilde{s}) + \phi(s)\|\|\theta_{\omega_t}^*\|\right]\|\psi_t(a|s)\| \leq C_\psi(R_{\max} + 2R_\theta)$ (the second $\leq$ uses the item 3 of Lemma D.3) and $\overline{X} = \mathbb{E}_{s_i \sim \nu_t}\left[X(s_i, a_i, s_{i+1}, s'_{i+1})|\mathcal{F}_t\right] = \overline{g}_t^*$ where $s_N, \omega_t \in \mathcal{F}_t := \sigma\left(\{\theta_{t'}^{(m)}\}_{m\in\mathcal{M}, 0\leq t'\leq t} \cup \{s_i, a_i, s'_{i+1}, \{e_i^{(m)}\}_{m\in\mathcal{M}}\}_{i=0}^{tN-1} \cup \{s_{tN}\}\right)$ are fixed. Hence, Lemma D.2 yields that

$$
\begin{aligned}
\mathbb{E}\left[\left\|g_t^* - \overline{g}_t^*\right\|^2 \big| \mathcal{F}_t\right] &= \mathbb{E}\left[\left\|\frac{1}{N} \sum_{i=tN}^{(t+1)N-1} X(s_i, a_i, s_{i+1}, s'_{i+1}) - \overline{X}\right\|^2 \Big| \mathcal{F}_t\right] \\
&\leq \frac{9C_\psi^2(R_{\max} + 2R_\theta)^2(\kappa + 1 - \rho)}{N(1-\rho)}.
\end{aligned}
\tag{50}
$$

Next, we bound the error term (IV) of eq. (42). Notice that

$$
\begin{aligned}
&\overline{g}_t^* - \nabla J(\omega_t) \\
&= \mathbb{E}_{\omega_t}\left[\left(\overline{R}(s,a,\widetilde{s}) + [\gamma\phi(\widetilde{s}) - \phi(s)]^\top \theta_{\omega_t}^* - [\overline{R}(s,a,\widetilde{s}) + \gamma V_{\omega_t}(\widetilde{s}) - V_{\omega_t}(s)]\right)\psi_t(a|s)\Big|\omega_t\right] \\
&= \mathbb{E}_{\omega_t}\left[\left(\gamma[\phi(\widetilde{s})^\top \theta_{\omega_t}^* - V_{\omega_t}(\widetilde{s})] - [\phi(s)^\top \theta_{\omega_t}^* - V_{\omega_t}(s)]\right)\psi_t(a|s)\Big|\omega_t\right].
\end{aligned}
\tag{51}
$$

Hence,

$$
\begin{aligned}
\|\overline{g}_t^* - \nabla J(\omega_t)\|^2 &= \left\|\mathbb{E}_{\omega_t}\left[\left(\gamma[\phi(\widetilde{s})^\top \theta_{\omega_t}^* - V_{\omega_t}(\widetilde{s})] - [\phi(s)^\top \theta_{\omega_t}^* - V_{\omega_t}(s)]\right)\psi_t(a|s)\Big|\omega_t\right]\right\|^2 \\
&\overset{(i)}{\leq} \mathbb{E}_{\omega_t}\left[\left\|\left(\gamma[\phi(\widetilde{s})^\top \theta_{\omega_t}^* - V_{\omega_t}(\widetilde{s})] - [\phi(s)^\top \theta_{\omega_t}^* - V_{\omega_t}(s)]\right)\psi_t(a|s)\right\|^2 \Big|\omega_t\right] \\
&\overset{(ii)}{\leq} 2C_\psi^2 \mathbb{E}_{\omega_t}\left[\gamma^2 \left\|\phi(\widetilde{s})^\top \theta_{\omega_t}^* - V_{\omega_t}(\widetilde{s})\right\|^2 + \left\|\phi(s)^\top \theta_{\omega_t}^* - V_{\omega_t}(s)\right\|^2 \Big|\omega_t\right] \\
&= 2C_\psi^2 \gamma^2 \int_{\mathcal{S}\times\mathcal{A}\times\mathcal{S}} \left\|\phi(\widetilde{s})^\top \theta_{\omega_t}^* - V_{\omega_t}(\widetilde{s})\right\|^2 \nu_t(s)\pi_t(a|s)\mathcal{P}(\widetilde{s}|s,a)dsdad\widetilde{s} \\
&\quad + 2C_\psi^2 \mathbb{E}_{\omega_t}\left[\left\|\phi(s)^\top \theta_{\omega_t}^* - V_{\omega_t}(s)\right\|^2 \Big|\omega_t\right] \\
&\overset{(iii)}{\leq} 2C_\psi^2 \gamma \int_{\mathcal{S}\times\mathcal{A}\times\mathcal{S}} \left\|\phi(\widetilde{s})^\top \theta_{\omega_t}^* - V_{\omega_t}(\widetilde{s})\right\|^2 \nu_t(s)\pi_t(a|s)\mathcal{P}_\xi(\widetilde{s}|s,a)dsdad\widetilde{s}
\end{aligned}
$$

$$+ 2C_\psi^2 \mathbb{E}_{\omega_t}\left[\left\|\phi(s)^\top\theta_{\omega_t}^* - V_{\omega_t}(s)\right\|^2 \Big| \omega_t\right]$$

$$\overset{(iv)}{=} 2C_\psi^2(\gamma+1)\mathbb{E}_{\omega_t}\left[\left\|\phi(s)^\top\theta_{\omega_t}^* - V_{\omega_t}(s)\right\|^2 \Big| \omega_t\right]$$

$$\overset{(v)}{\leq} 4C_\psi^2 \zeta_{\text{approx}}^{\text{critic}}, \tag{52}$$

where (i) applies Jensen's inequality to the convex function $\|\cdot\|^2$, (ii) uses the inequality that $\|x+y\|^2 \leq 2\|x\|^2 + 2\|y\|^2$ for any $x, y \in \mathbb{R}^d$, (iii) uses the inequality that $\mathcal{P}(s'|s,a) \leq \gamma^{-1}\mathcal{P}_\xi(s'|s,a); \forall s, s' \in \mathcal{S}, a \in \mathcal{A}$, (iv) uses the equality that $\int_{\mathcal{S}\times\mathcal{A}} \nu_t(s)\pi_t(a|s)\mathcal{P}_\xi(\widetilde{s}|s,a)dsda = \nu_t(\widetilde{s})$, and (v) uses the notation that $\zeta_{\text{approx}}^{\text{critic}} := \sup_\omega \mathbb{E}_{s\sim\nu_\omega}\left[\left|V_\omega(s) - \phi(s)^\top\theta_\omega^*\right|^2\right]$. Substituting eqs. (47),(48),(50)&(52) into eq. (42) yields that

$$\mathbb{E}\left[\left\|\widehat{\nabla}J(\omega_t) - \nabla J(\omega_t)\right\|^2 \Big| \mathcal{F}_t\right]$$

$$\leq 4C_\psi^2 R_{\max}^2\left(M\sigma_W^{2T'} + \frac{4}{N}\overline{\sigma}^2\right) + 16C_\psi^2 \sum_{m=1}^M \left\|\theta_t^{(m)} - \theta_{\omega_t}^*\right\|^2$$

$$+ \frac{36C_\psi^2(R_{\max}+2R_\theta)^2(\kappa+1-\rho)}{N(1-\rho)} + 16C_\psi^2\zeta_{\text{approx}}^{\text{critic}}$$

$$= c_4\sigma_W^{2T'} + \frac{c_7}{N} + 16C_\psi^2 \sum_{m=1}^M \left\|\theta_t^{(m)} - \theta_{\omega_t}^*\right\|^2 + 16C_\psi^2\zeta_{\text{approx}}^{\text{critic}}, \tag{53}$$

where $\theta_t^{(m)}, \omega_t \in \mathcal{F}_t$ are fixed, and we take the conditional expectation of eq. (47) on $\mathcal{F}_t \subset \mathcal{F}_t'$ and denote that $c_4 := 4MC_\psi^2 R_{\max}^2$, $c_7 := 16C_\psi^2 R_{\max}^2\overline{\sigma}^2 + \frac{36C_\psi^2(R_{\max}+2R_\theta)^2(\kappa+1-\rho)}{1-\rho}$. Substituting eq. (25) into the unconditional expectation of eq. (53) yields that

$$\mathbb{E}\left[\left\|\widehat{\nabla}J(\omega_t) - \nabla J(\omega_t)\right\|^2\right]$$

$$\leq c_4\sigma_W^{2T'} + \frac{c_7}{N} + 16C_\psi^2\left(\sigma_W^{2T_c'}\beta^2 c_2 + 2M\left[c_3\left(1 - \frac{\lambda_B}{8}\beta\right)^{T_c} + \frac{c_1}{N_c}\right]\right) + 16C_\psi^2\zeta_{\text{approx}}^{\text{critic}}$$

$$= c_4\sigma_W^{2T'} + c_5\beta^2\sigma_W^{2T_c'} + c_6\left(1 - \frac{\lambda_B}{8}\beta\right)^{T_c} + \frac{c_7}{N} + \frac{c_8}{N_c} + 16C_\psi^2\zeta_{\text{approx}}^{\text{critic}},$$

where we denote that $c_5 := 16c_2 C_\psi^2$, $c_6 := 32Mc_3 C_\psi^2$, $c_8 := 32Mc_1 C_\psi^2$. This proves eq. (39).

Equation (40) can be proved in the same way as that of proving eq. (53). There are two differences. First, $\widehat{\nabla}J(\omega_t; \mathcal{B}_{t,k})$ uses the minibatch $\mathcal{B}_{t,k}$ of size $N_k$ while $\widehat{\nabla}J(\omega_t)$ uses batchsize $N$. Second, eq. (40) is conditioned on the filtration $\mathcal{F}_{t,k} := \sigma\left[\mathcal{F}_t \cup \sigma\left(\left\{s_i, a_i, s_{i+1}, s_{i+1}', \{e_i^{(m)}\}_{m\in\mathcal{M}}\right\}_{i\in\cup_{k'=0}^{k-1}\mathcal{B}_{t,k'}}\right)\right]$ which includes not only the filtration $\mathcal{F}_t$ use by eq. (53) but also the minibatches $\cup_{k'=0}^{k-1}\mathcal{B}_{t,k'}$ used by the previous $(k-1)$ SGD steps. $\qquad\square$

**Lemma D.7.** *Implementing Algorithm 3 with $\eta \leq \frac{1}{2C_\psi^2}$, $T' \geq \frac{\ln M}{2\ln(\sigma_W^{-1})}$, $T_z \geq \frac{\ln(3D_J C_\psi^2)}{\ln(\sigma_W^{-1})}$, $K \geq \frac{\ln 3}{\ln[(1-\eta\lambda_F/2)^{-1}]}$, $N \geq \frac{2304C_\psi^4(\kappa+1-\rho)}{\eta\lambda_F^5(1-\rho)(1-\eta\lambda_F/2)^{(K-1)/2}}$ and $N_k \propto (1-\eta\lambda_F/2)^{-k/2}$, the involved quantities have the following properties, where $\mathbb{E}_\omega$ denotes the expectation under the underlying distributions that $s \sim \nu_\omega$, $a \sim \pi_\omega(\cdot|s)$.*

1. *$\lambda_F \leq \lambda_{\max}[F(\omega)] = \|F(\omega)\| \leq C_\psi^2, \forall\omega$.*

2. *$\frac{1}{2} \leq 1 - \eta C_\psi^2 \leq \|I - \eta F(\omega)\| \leq 1 - \eta\lambda_F$, so $\eta \leq \frac{1}{2\lambda_F}$.*

3. *$C_\psi^{-2} \leq \|F(\omega)^{-1}\| \leq \lambda_F^{-1}$. For any $\omega, x \in \mathbb{R}^{d_\omega}$, $x^\top F(\omega)^{-1}x \geq C_\psi^{-2}\|x\|^2$.*

4. *$\|h(\omega)\| \leq \frac{1}{\lambda_F}\|\nabla J(\omega)\| \leq \frac{D_J}{\lambda_F}$.*

5. *$h(\omega) = \arg\min_h \mathbb{E}_\omega\left[\left(\psi_\omega(a|s)^\top h - A_\omega(s,a)\right)^2\right]$, so*
   *$\mathbb{E}_\omega\left[\left(\psi_\omega(a|s)^\top h(\omega) - A_\omega(s,a)\right)^2\right] \leq \zeta_{approx}^{actor}$ where $s \sim \nu_\omega$, $a \sim \pi_\omega(\cdot|s)$.*

6. $\mathbb{E}_{\omega^*}\left[\psi_\omega(a|s)^\top h(\omega) - A_\omega(s,a)\right] \geq -C_*\sqrt{\zeta_{approx}^{actor}}, \forall \omega.$

7. $N_k = \frac{N(1-\eta\lambda_F/2)^{(K-1-k)/2}(1-\sqrt{1-\eta\lambda_F/2})}{1-(1-\eta\lambda_F/2)^{K/2}} \geq \frac{576C_\psi^4(\kappa+1-\rho)}{\lambda_F^4(1-\rho)}.$

8. $h_t$ *approximates the natural gradient* $h(\omega_t)$ *with the following error bound.*

$$\mathbb{E}\left[\left\|h_t - h(\omega_t)\right\|^2\right] \leq c_{10}\left(1 - \frac{\eta\lambda_F}{2}\right)^{(K-1)/2} + c_{11}\sigma_W^{2T_z} + c_{12}\sigma_W^{2T'} + c_{13}\beta^2\sigma_W^{2T_c'}$$
$$+ c_{14}\left(1 - \frac{\lambda_B}{8}\beta\right)^{T_c} + \frac{c_{15}}{N_c} + c_{16}\zeta_{approx}^{critic}. \tag{54}$$

*Proof.* The item 1 is proved by the following inequality.

$$\lambda_F \overset{(i)}{\leq} \lambda_{\min}[F(\omega)] \leq \lambda_{\max}[F(\omega)] \overset{(ii)}{=} \|F(\omega)\|$$
$$= \left\|\mathbb{E}_\omega\left[\psi(a|s)\psi(a|s)^\top\right]\right\| \leq \mathbb{E}_\omega\left[\left\|\psi(a|s)\right\|\left\|\psi(a|s)^\top\right\|\right] \overset{(iii)}{\leq} C_\psi^2,$$

where (i) uses Assumption 6, (ii) uses the fact that $F(\omega)$ is positive definite implied by Assumption 6, (iii) applies Jensen's inequality to the convex function $\|\cdot\|$ and (iv) uses Assumption 2.

Next we will prove the item 2. On one hand,

$$\lambda_{\min}\left[I - \eta F(\omega)\right] = 1 - \eta\lambda_{\max}\left[F(\omega)\right] \overset{(i)}{\geq} 1 - \eta C_\psi^2 \geq \frac{1}{2}, \tag{55}$$

where (i) uses the item 1, (ii) uses the condition that $\eta \leq \frac{1}{2C_\psi^2}$. On the other hand,

$$\lambda_{\min}\left[I - \eta F(\omega)\right] \leq \lambda_{\max}\left[I - \eta F(\omega)\right] \overset{(i)}{=} \|I - \eta F(\omega)\| = I - \eta\lambda_{\min}\left[F(\omega)\right] \leq 1 - \eta\lambda_F, \tag{56}$$

where (i) uses the fact that $I - \eta F(\omega)$ is positive definite based on eq. (55). Hence, eqs. (55) & (56) prove the item 2.

The item 3 can be proved by the fact that $F(\omega)^{-1}$ is positive definite with minimum eigenvalue $\lambda_{\max}[F(\omega)]^{-1} \geq C_\psi^{-2}$ and maximum eigenvalue $\lambda_{\min}[F(\omega)]^{-1} \leq \lambda_F^{-1}$ implied by the item 1.

The item 4 can be proved by the following inequality.

$$\|h(\omega)\| = \left\|F(\omega)^{-1}\nabla J(\omega)\right\| \leq \left\|F(\omega^{-1})\right\|\left\|\nabla J(\omega)\right\| \overset{(i)}{\leq} \lambda_F^{-1}\left\|\nabla J(\omega)\right\| \overset{(ii)}{\leq} \lambda_F^{-1}D_J,$$

where (i) uses the item 3 and (ii) uses the item 8 of Lemma D.5.

Next we will prove item 5.

Consider the following function of $x \in \mathbb{R}^{d_\omega}$.

$$f_\omega(x) = \frac{1}{2}\mathbb{E}_\omega\left[\left(\psi_\omega(a|s)^\top x - A_\omega(s,a)\right)^2\right]$$
$$= \frac{1}{2}x^\top\mathbb{E}_\omega\left[\psi_\omega(a|s)\psi_\omega(a|s)^\top\right]x - \mathbb{E}_\omega\left[A_\omega(s,a)\psi_\omega(a|s)\right]^\top x + \frac{1}{2}\mathbb{E}_\omega\left[A_\omega(s,a)^2\right]$$
$$= \frac{1}{2}x^\top F(\omega)x - \nabla J(\omega)^\top x + \frac{1}{2}\mathbb{E}_\omega\left[A_\omega(s,a)^2\right]$$

Since $\nabla^2 f(\omega) = F(\omega)$ is positive definite, $f$ is strongly convex quadratic and thus it has unique minimizer $h(\omega) = F(\omega)^{-1}\nabla J(\omega)$ obtained by solving $h$ from the equation $\nabla f_\omega(h) = F(\omega)h - \nabla J(\omega) = 0$. Hence,

$$\mathbb{E}_\omega\left[\left\|\psi_\omega(a|s)^\top h(\omega) - A_\omega(s,a)\right\|^2\right]$$
$$= \min_h \mathbb{E}_\omega\left[\left(\psi_\omega(a|s)^\top h - A_\omega(s,a)\right)^2\right]$$
$$\leq \sup_\omega \min_h \mathbb{E}_\omega\left[\left(\psi_\omega(a|s)^\top h - A_\omega(s,a)\right)^2\right] := \zeta_{approx}^{actor}, \tag{57}$$

which proves the item 5.

The item 6 can be proved by the following inequality.

$$
\mathbb{E}_{\omega^*}\big[A_\omega(s,a) - \psi_\omega(a|s)^\top h(\omega)\big]
$$

$$
= \int \nu_{\omega^*}(s)\pi_{\omega^*}(a|s)\big[A_\omega(s,a) - \psi_\omega(a|s)^\top h(\omega)\big]dsda
$$

$$
= \int \nu_\omega(s)\pi_\omega(a|s)\frac{\nu_{\omega^*}(s)\pi_{\omega^*}(a|s)}{\nu_\omega(s)\pi_\omega(a|s)}\big[A_\omega(s,a) - \psi_\omega(a|s)^\top h(\omega)\big]dsda
$$

$$
= \mathbb{E}_\omega\left[\frac{\nu_{\omega^*}(s)\pi_{\omega^*}(a|s)}{\nu_\omega(s)\pi_\omega(a|s)}\big[A_\omega(s,a) - \psi_\omega(a|s)^\top h(\omega)\big]\right]
$$

$$
\leq \sqrt{\mathbb{E}_\omega\left[\left(\frac{\nu_{\omega^*}(s)\pi_{\omega^*}(a|s)}{\nu_\omega(s)\pi_\omega(a|s)}\right)^2\right]}\sqrt{\mathbb{E}_\omega\big[\big(A_\omega(s,a) - \psi_\omega(a|s)^\top h(\omega)\big)^2\big]} \overset{(i)}{\leq} C_*\sqrt{\zeta_{\text{approx}}^{\text{actor}}}, \tag{58}
$$

where (i) uses Assumption 7 and the item 5. Multiplying $-1$ to the above inequality proves the item 6.

Next, the item 7 can be proved as follows.

$$
N_k \overset{(i)}{=} N\frac{(1 - \eta\lambda_F/2)^{-k/2}}{\sum_{k'=0}^{K-1}(1 - \eta\lambda_F/2)^{-k'/2}}
$$

$$
= \frac{N(1 - \eta\lambda_F/2)^{(K-1-k)/2}(1 - \sqrt{1 - \eta\lambda_F/2})}{1 - (1 - \eta\lambda_F/2)^{K/2}}
$$

$$
\overset{(ii)}{\geq} \frac{2304C_\psi^4(\kappa + 1 - \rho)}{\eta\lambda_F^5(1 - \rho)(1 - \eta\lambda_F/2)^{(K-1)/2}}\frac{(1 - \eta\lambda_F/2)^{(K-1)/2}(\eta\lambda_F/2)}{1 + \sqrt{1 - \eta\lambda_F/2}}
$$

$$
\geq \frac{576C_\psi^4(\kappa + 1 - \rho)}{\lambda_F^4(1 - \rho)},
$$

where (i) uses the conditions that $N_k \propto (1 - \eta\lambda_F/2)^{-k/2}$ and $\sum_{k=0}^{K-1}N_k = N$ and (ii) uses the condition that $N \geq \frac{2304C_\psi^4(\kappa+1-\rho)}{\eta\lambda_F^5(1-\rho)(1-\eta\lambda_F/2)^{(K-1)/2}}$

Finally, we will prove the item 8. Until the end of this proof, we use the underlying distribution that $a_i \sim \pi_t(\cdot|s_i), s_{i+1} \sim \mathcal{P}_\xi(\cdot|s_i, a_i)$ for $tN \leq i \leq (t+1)N - 1$ in the $t$-th iteration of the multi-agent NAC algorithm (Algorithm 1).

The local averaging steps of $z_{i,\ell} := [z_{i,\ell}^{(1)}, \ldots, z_{i,\ell}^{(M)}]^\top$ yield the following consensus error bound.

$$
\sum_{m=1}^{M}(z_{T_z}^{(m)} - \overline{z}_{T_z})^2 = \|\Delta z_{i,T_z}\|^2 = \|\Delta W^{T_z}z_{i,0}\|^2 \overset{(i)}{=} \|W^{T_z}\Delta z_{i,0}\|^2 \overset{(ii)}{\leq} \sigma_W^{2T_z}\|\Delta z_{i,0}\|^2
$$

$$
\overset{(iii)}{\leq} \sigma_W^{2T_z}\sum_{m=1}^{M}(z_{i,0}^{(m)})^2 = \sigma_W^{2T_z}\sum_{m=1}^{M}\big[\psi_t^{(m)}(a_i^{(m)}|s_i)^\top h_{t,k}^{(m)}\big]^2
$$

$$
\overset{(iv)}{\leq} C_\psi^2\sigma_W^{2T_z}\sum_{m=1}^{M}\big\|h_{t,k}^{(m)}\big\|^2 \leq C_\psi^2\sigma_W^{2T_z}\big\|h_{t,k}\big\|^2,
$$

where $\overline{z}_{T_z} := \frac{1}{M}\sum_{m=1}^{M}z_{i,T_z}^{(m)}$, (i) and (ii) use the items 1 and 3 of Lemma D.1 respectively, (iii) uses the equality that $\|\Delta\| = 1$, and (iv) uses the item 1 of Lemma D.5.

Then, we define the following stochastic gradients of function $f_\omega$.

$$
\widetilde{\nabla}_{\omega^{(m)}}f_{\omega_t}(h_{t,k}) := \frac{1}{N_k}\sum_{i\in\mathcal{B}_{t,k}}\psi_t^{(m)}(a_i^{(m)}|s_i)\psi_t(a_i|s_i)^\top h_{t,k} - \widehat{\nabla}_{\omega^{(m)}}J(\omega_t;\mathcal{B}_{t,k})
$$

$$
\widetilde{\nabla}f_{\omega_t}(h_{t,k}) := \frac{1}{N_k}\sum_{i\in\mathcal{B}_{t,k}}\psi_t(a_i|s_i)\psi_t(a_i|s_i)^\top h_{t,k} - \widehat{\nabla}J(\omega_t;\mathcal{B}_{t,k})
$$

$$= \big[\widetilde{\nabla}_{\omega^{(1)}} f_{\omega_t}(h_{t,k}); \ldots; \widetilde{\nabla}_{\omega^{(M)}} f_{\omega_t}(h_{t,k})\big],$$

$$\widehat{\nabla}_{\omega^{(m)}} f_{\omega_t}(h_{t,k}) := \frac{M}{N_k} \sum_{i \in \mathcal{B}_{t,k}} \psi_t^{(m)}(a_i^{(m)}|s_i) z_{i,T_z}^{(m)} - \widehat{\nabla}_{\omega^{(m)}} J(\omega_t; \mathcal{B}_{t,k}),$$

$$\widehat{\nabla} f_{\omega_t}(h_{t,k}) := \big[\widehat{\nabla}_{\omega^{(1)}} f_{\omega_t}(h_{t,k}); \ldots; \widehat{\nabla}_{\omega^{(M)}} f_{\omega_t}(h_{t,k})\big]^\top,$$

where $\widehat{\nabla}_{\omega^{(m)}} J(\omega_t; \mathcal{B}_{t,k})$ and $\widehat{\nabla} J(\omega_t; \mathcal{B}_{t,k})$ are defined in eqs. (35) & (36) respectively. Hence,

$$\big\|\widehat{\nabla} f_{\omega_t}(h_{t,k}) - \widetilde{\nabla} f_{\omega_t}(h_{t,k})\big\|^2$$

$$= \sum_{m=1}^{M} \big\|\widehat{\nabla}_{\omega^{(m)}} f_{\omega_t}(h_{t,k}) - \widetilde{\nabla}_{\omega^{(m)}} f_{\omega_t}(h_{t,k})\big\|^2$$

$$= \sum_{m=1}^{M} \Big\|\frac{1}{N_k} \sum_{i \in \mathcal{B}_{t,k}} \big[M z_{i,T_z}^{(m)} - \psi_t(a_i|s_i)^\top h_{t,k}\big] \psi_t^{(m)}(a_i|s_i)\Big\|^2$$

$$\overset{(i)}{\leq} \frac{1}{N_k} \sum_{i \in \mathcal{B}_{t,k}} \sum_{m=1}^{M} \big\|M\big(z_{i,T_z}^{(m)} - \overline{z}_{T_z}\big)\psi_t^{(m)}(a_i|s_i)\big\|^2$$

$$\overset{(ii)}{\leq} \frac{M^2 C_\psi^2}{N_k} \sum_{i \in \mathcal{B}_{t,k}} \sum_{m=1}^{M} (z_{i,T_z}^{(m)} - \overline{z}_{T_z})^2 \leq M^2 C_\psi^4 \sigma_W^{2T_z} \big\|h_{t,k}\big\|^2. \tag{59}$$

where (i) uses the equality that $\psi_t(a_i|s_i)^\top h_{t,k} = \sum_{m \in \mathcal{M}} z_{i,T_z}^{(m)} = M\overline{z}_{T_z}$, (ii) uses the item 1 of Lemma D.5.

Since, $\omega_t, h_{t,k} \in \mathcal{F}_{t,k}$ while $\{s_i, a_i\}_{i \in \mathcal{B}_{t,k}}$ are random. Hence,

$$\mathbb{E}\big[\big\|\widehat{\nabla} f_{\omega_t}(h_{t,k}) - \nabla f_{\omega_t}(h_{t,k})\big\|^2\big|\mathcal{F}_{t,k}\big]$$

$$= \mathbb{E}\Big[\Big\|\frac{1}{N_k}\sum_{i \in \mathcal{B}_{t,k}}\big[\psi_t(a_i|s_i)\psi_t(a_i|s_i)^\top\big]h_{t,k} - \widehat{\nabla} J(\omega_t; \mathcal{B}_{t,k}) - F(\omega_t)h_{t,k} + \nabla J(\omega_t)\Big\|^2\Big|\mathcal{F}_{t,k}\Big]$$

$$\overset{(i)}{\leq} 2\mathbb{E}\Big[\Big\|\frac{1}{N_k}\sum_{i \in \mathcal{B}_{t,k}}\big[\psi_t(a_i|s_i)\psi_t(a_i|s_i)^\top\big] - F(\omega_t)\Big\|^2\|h_{t,k}\|^2\Big|\mathcal{F}_{t,k}\Big]$$

$$\quad + 2\mathbb{E}\big[\big\|\widehat{\nabla} J(\omega_t; \mathcal{B}_{t,k}) - \nabla J(\omega_t)\big\|^2\big|\mathcal{F}_{t,k}\big]$$

$$\overset{(ii)}{=} 2\mathbb{E}\Big[\Big\|\frac{1}{N_k}\sum_{i \in \mathcal{B}_{t,k}}\big[\psi_t(a_i|s_i)\psi_t(a_i|s_i)^\top\big] - F(\omega_t)\Big\|^2\Big|\mathcal{F}_{t,k}\Big]\|h_{t,k}\|^2$$

$$\quad + 2\mathbb{E}\big[\big\|\widehat{\nabla} J(\omega_t; \mathcal{B}_{t,k}) - \nabla J(\omega_t)\big\|^2\big|\mathcal{F}_{t,k}\big]$$

$$\overset{(iii)}{\leq} \frac{18 C_\psi^4 (\kappa + 1 - \rho)}{N_k(1-\rho)}\|h_{t,k}\|^2 + 2c_4\sigma_W^{2T'} + \frac{2c_7}{N_k} + 32C_\psi^2 \sum_{m=1}^{M}\big\|\theta_t^{(m)} - \theta_{\omega_t}^*\big\|^2 + 32C_\psi^2 \zeta_{\text{approx}}^{\text{critic}}, \tag{60}$$

where (i) uses the inequalities that $\|x + y\|^2 \leq 2\|x\|^2 + 2\|y\|^2$ for any $x, y \in \mathbb{R}^d$, (ii) uses the fact that $h_{t,k} \in \mathcal{F}_{t,k}$, and (iii) uses eq. (40) and applies Lemma D.2 to the quantity that $X(s, a, s', \widetilde{s}) = \psi_t(a|s)\psi_t(a|s)^\top$ in which $\omega_t \in \mathcal{F}_{t,k}$ is fixed and $\|X(s, a, s', \widetilde{s})\|_F \leq C_\psi^2$.

Combining eqs. (59) & (60) yields that

$$\mathbb{E}\big[\big\|\widehat{\nabla} f_{\omega_t}(h_{t,k}) - \nabla f_{\omega_t}(h_{t,k})\big\|^2\big|\mathcal{F}_{t,k}\big]$$

$$\leq 2\mathbb{E}\big[\big\|\widehat{\nabla} f_{\omega_t}(h_{t,k}) - \widetilde{\nabla} f_{\omega_t}(h_{t,k})\big\|^2\big|\mathcal{F}_{t,k}\big] + 2\mathbb{E}\big[\big\|\widetilde{\nabla} f_{\omega_t}(h_{t,k}) - \nabla f_{\omega_t}(h_{t,k})\big\|^2\big|\mathcal{F}_{t,k}\big]$$

$$\leq C_\psi^4\Big[2M^2\sigma_W^{2T_z} + \frac{36(\kappa + 1 - \rho)}{N_k(1-\rho)}\Big]\|h_{t,k}\|^2 + 4c_4\sigma_W^{2T'}$$

$$\quad + \frac{4c_7}{N_k} + 64C_\psi^2 \sum_{m=1}^{M}\big\|\theta_t^{(m)} - \theta_{\omega_t}^*\big\|^2 + 64C_\psi^2 \zeta_{\text{approx}}^{\text{critic}}. \tag{61}$$

Therefore,

$$\mathbb{E}\big[\big\|h_{t,k+1} - h(\omega_t)\big\|^2 \big| \mathcal{F}_{t,k}\big]$$

$$= \mathbb{E}\big[\big\|h_{t,k} - \eta\widehat{\nabla}f_{\omega_t}(h_{t,k}) - h(\omega_t)\big\|^2 \big| \mathcal{F}_{t,k}\big]$$

$$\overset{(i)}{\leq} (1 + \eta\lambda_F)\mathbb{E}\big[\big\|h_{t,k} - \eta\nabla f_{\omega_t}(h_{t,k}) - h(\omega_t)\big\|^2 \big| \mathcal{F}_{t,k}\big]$$

$$+ \big[1 + (\eta\lambda_F)^{-1}\big]\mathbb{E}\big[\big\|\eta\big[\widehat{\nabla}f_{\omega_t}(h_{t,k}) - \nabla f_{\omega_t}(h_{t,k})\big]\big\|^2 \big| \mathcal{F}_{t,k}\big]$$

$$\overset{(ii)}{=} (1 + \eta\lambda_F)\big\|h_{t,k} - \eta F(\omega_t)\big[h_{t,k} - h(\omega_t)\big] - h(\omega_t)\big\|^2$$

$$+ \eta\big(\eta + \lambda_F^{-1}\big)\mathbb{E}\big[\big\|\widehat{\nabla}f_{\omega_t}(h_{t,k}) - \nabla f_{\omega_t}(h_{t,k})\big\|^2 \big| \mathcal{F}_{t,k}\big]$$

$$= (1 + \eta\lambda_F)\big\|\big[I - \eta F(\omega_t)\big]\big[h_{t,k} - h(\omega_t)\big]\big\|^2$$

$$+ \eta\big(\eta + \lambda_F^{-1}\big)\mathbb{E}\big[\big\|\big[\widehat{\nabla}f_{\omega_t}(h_{t,k}) - \nabla f_{\omega_t}(h_{t,k})\big]\big\|^2 \big| \mathcal{F}_{t,k}\big]$$

$$\overset{(iii)}{\leq} (1 + \eta\lambda_F)(1 - \eta\lambda_F)^2\big\|h_{t,k} - h(\omega_t)\big\|^2$$

$$+ \frac{2\eta}{\lambda_F}\Big(C_\psi^4\Big[2M^2\sigma_W^{2T_z} + \frac{36(\kappa + 1 - \rho)}{N_k(1-\rho)}\Big]\|h_{t,k}\|^2 + 4c_4\sigma_W^{2T'}$$

$$+ \frac{4c_7}{N_k} + 64C_\psi^2\sum_{m=1}^{M}\big\|\theta_t^{(m)} - \theta_{\omega_t}^*\big\|^2 + 64C_\psi^2\zeta_{\text{approx}}^{\text{critic}}\Big)$$

$$\leq (1 - \eta\lambda_F)\big\|h_{t,k} - h(\omega_t)\big\|^2$$

$$+ \frac{2\eta}{\lambda_F}\Big(2C_\psi^4\Big[2M^2\sigma_W^{2T_z} + \frac{36(\kappa + 1 - \rho)}{N_k(1-\rho)}\Big](\|h_{t,k} - h(\omega_t)\|^2 + \|h(\omega_t)\|^2)$$

$$+ 4c_4\sigma_W^{2T'} + \frac{4c_7}{N_k} + 64C_\psi^2\sum_{m=1}^{M}\big\|\theta_t^{(m)} - \theta_{\omega_t}^*\big\|^2 + 64C_\psi^2\zeta_{\text{approx}}^{\text{critic}}\Big)$$

$$\overset{(iv)}{\leq} \Big(1 - \frac{\eta\lambda_F}{2}\Big)\big\|h_{t,k} - h(\omega_t)\big\|^2 + \frac{2\eta}{\lambda_F}\Big(2C_\psi^4\Big[2M^2\sigma_W^{2T_z} + \frac{36(\kappa + 1 - \rho)}{N_k(1-\rho)}\Big]\frac{D_J^2}{\lambda_F^2}$$

$$+ 4c_4\sigma_W^{2T'} + \frac{4c_7}{N_k} + 64C_\psi^2\sum_{m=1}^{M}\big\|\theta_t^{(m)} - \theta_{\omega_t}^*\big\|^2 + 64C_\psi^2\zeta_{\text{approx}}^{\text{critic}}\Big)$$

$$\overset{(v)}{\leq} \Big(1 - \frac{\eta\lambda_F}{2}\Big)\big\|h_{t,k} - h(\omega_t)\big\|^2 + \frac{8\eta}{\lambda_F}\Big(C_\psi^4 M^2\sigma_W^{2T_z} + \frac{c_9}{N_k}$$

$$+ c_4\sigma_W^{2T'} + 16C_\psi^2\sum_{m=1}^{M}\big\|\theta_t^{(m)} - \theta_{\omega_t}^*\big\|^2 + 16C_\psi^2\zeta_{\text{approx}}^{\text{critic}}\Big)$$

where (i) uses the inequality that $\|x+y\|^2 \leq (1+\eta\lambda_F)\|x\|^2 + [1+(\eta\lambda_F)^{-1}]\|y\|^2$ for any $x, y \in \mathbb{R}^d$, (ii) uses the notation that $\nabla f_{\omega_t}(h) = F(\omega_t)h - \nabla J(\omega_t) = F(\omega_t)[h - h(\omega_t)]$ and the fact that $\omega_t, h_{t,k} \in \mathcal{F}_{t,k}$, (iii) uses eq. (61) and the item 2 of this Lemma, (iv) uses the conditions that $T_z \geq \frac{\ln(3D_J C_\psi^2)}{\ln(\sigma_W^{-1})}$ and the item 7 of this Lemma, and (v) uses the notation that $c_9 := \frac{18C_\psi^4 D_J^2(\kappa+1-\rho)}{\lambda_F^2(1-\rho)} + c_7$.

Then, taking unconditional expectation of the above inequality and iterating it over $k = 0, 1, \ldots, K-1$ yield that

$$\mathbb{E}\big[\big\|h_t - h(\omega_t)\big\|^2\big] = \mathbb{E}\big[\big\|h_{t,K} - h(\omega_t)\big\|^2\big]$$

$$\leq \Big(1 - \frac{\eta\lambda_F}{2}\Big)^K \mathbb{E}\big[\big\|h_{t,0} - h(\omega_t)\big\|^2\big] + \frac{8\eta}{\lambda_F}\sum_{k=0}^{K-1}\Big(1 - \frac{\eta\lambda_F}{2}\Big)^{K-1-k}$$

$$\Big(C_\psi^4 M^2\sigma_W^{2T_z} + \frac{c_9}{N_k} + c_4\sigma_W^{2T'} + 16C_\psi^2\sum_{m=1}^{M}\mathbb{E}\big[\big\|\theta_t^{(m)} - \theta_{\omega_t}^*\big\|^2\big] + 16C_\psi^2\zeta_{\text{approx}}^{\text{critic}}\Big)$$

$$\overset{(i)}{\leq} \Big(1 - \frac{\eta\lambda_F}{2}\Big)^K \mathbb{E}\big[\big\|h_{t-1} - h(\omega_t)\big\|^2\big]$$

$$+ \frac{16}{\lambda_F^2} \Big( C_\psi^4 M^2 \sigma_W^{2T_z} + c_4 \sigma_W^{2T'} + 16 C_\psi^2 \sum_{m=1}^M \mathbb{E}\big[\|\theta_t^{(m)} - \theta_{\omega_t}^*\|^2\big] + 16 C_\psi^2 \zeta_{\text{approx}}^{\text{critic}} \Big)$$

$$+ \frac{8\eta c_9 [1 - (1 - \eta\lambda_F/2)^{K/2}]}{N\lambda_F(1 - \sqrt{1 - \eta\lambda_F/2})} \sum_{k=0}^{K-1} \Big( 1 - \frac{\eta\lambda_F}{2} \Big)^{(K-1-k)/2}$$

$$\overset{(ii)}{\leq} \Big( 1 - \frac{\eta\lambda_F}{2} \Big)^K \mathbb{E}\big[\|h_{t-1} - h(\omega_t)\|^2\big] + \frac{16}{\lambda_F^2} \big( C_\psi^4 M^2 \sigma_W^{2T_z} + c_4 \sigma_W^{2T'} + 16 C_\psi^2 \zeta_{\text{approx}}^{\text{critic}} \big)$$

$$+ \frac{256 C_\psi^2}{\lambda_F^2} \Big( \sigma_W^{2T_c'}\beta^2 c_2 + 2M \Big[ c_3 \Big( 1 - \frac{\lambda_B}{8}\beta \Big)^{T_c} + \frac{c_1}{N_c} \Big] \Big) + \frac{8\eta c_9}{N\lambda_F(1 - \sqrt{1 - \eta\lambda_F/2})^2}$$

$$\overset{(iii)}{\leq} \Big( 1 - \frac{\eta\lambda_F}{2} \Big)^K \mathbb{E}\big[\|h_{t-1} - h(\omega_t)\|^2\big] + \frac{16}{\lambda_F^2} \big( C_\psi^4 M^2 \sigma_W^{2T_z} + c_4 \sigma_W^{2T'} + 16 C_\psi^2 \zeta_{\text{approx}}^{\text{critic}} \big)$$

$$+ \frac{256 C_\psi^2}{\lambda_F^2} \Big( \sigma_W^{2T_c'}\beta^2 c_2 + 2M \Big[ c_3 \Big( 1 - \frac{\lambda_B}{8}\beta \Big)^{T_c} + \frac{c_1}{N_c} \Big] \Big) + \frac{128 c_9}{N\eta\lambda_F^3} \qquad (62)$$

$$\overset{(iv)}{\leq} 3 \Big( 1 - \frac{\eta\lambda_F}{2} \Big)^K \mathbb{E}\big[\|h_{t-1} - h(\omega_{t-1})\|^2 + \|h(\omega_{t-1})\|^2 + \| - h(\omega_t)\|^2\big]$$

$$+ \frac{16}{\lambda_F^2} \big( C_\psi^4 M^2 \sigma_W^{2T_z} + c_4 \sigma_W^{2T'} + 16 C_\psi^2 \zeta_{\text{approx}}^{\text{critic}} \big)$$

$$+ \frac{256 C_\psi^2}{\lambda_F^2} \Big( \sigma_W^{2T_c'}\beta^2 c_2 + 2M \Big[ c_3 \Big( 1 - \frac{\lambda_B}{8}\beta \Big)^{T_c} + \frac{c_1}{N_c} \Big] \Big) + \frac{128 c_9}{N\eta\lambda_F^3}$$

$$\overset{(v)}{\leq} 3 \Big( 1 - \frac{\eta\lambda_F}{2} \Big)^K \mathbb{E}\big[\|h_{t-1} - h(\omega_{t-1})\|^2\big] + \frac{6D_J^2}{\lambda_F^2} \Big( 1 - \frac{\eta\lambda_F}{2} \Big)^K$$

$$+ \frac{16}{\lambda_F^2} \big( C_\psi^4 M^2 \sigma_W^{2T_z} + c_4 \sigma_W^{2T'} + 16 C_\psi^2 \zeta_{\text{approx}}^{\text{critic}} \big)$$

$$+ \frac{256 C_\psi^2}{\lambda_F^2} \Big( \sigma_W^{2T_c'}\beta^2 c_2 + 2M \Big[ c_3 \Big( 1 - \frac{\lambda_B}{8}\beta \Big)^{T_c} + \frac{c_1}{N_c} \Big] \Big) + \frac{128 c_9}{N\eta\lambda_F^3},$$

where (i) uses the notation that $h_{t,0} = h_t$, the item 7 of this Lemma and the inequality that $\sum_{k=0}^{K-1} \big( 1 - \frac{\eta\lambda_F}{2} \big)^{K-1-k} \leq \frac{2}{\eta\lambda_F}$, (ii) uses Lemma D.4, (iii) uses the inequality that $\frac{1}{(1 - \sqrt{1 - \eta\lambda_F/2})^2} = \frac{(1 + \sqrt{1 - \eta\lambda_F/2})^2}{(\eta\lambda_F/2)^2} \leq \frac{16}{(\eta\lambda_F)^2}$ implied by the item 2 of this Lemma, (iv) uses the inequality that $\|x + y + z\|^2 \leq 3\|x\|^2 + 3\|y\|^2 + 3\|z\|^2, \forall x, y, z \in \mathbb{R}^d$, and (v) uses the items 4 of this Lemma. Taking unconditional expectation of the above inequality and iterating it over $t$ yield that

$$\mathbb{E}\big[\|h_t - h(\omega_t)\|^2\big]$$

$$\overset{(i)}{\leq} \Big[ 3 \Big( 1 - \frac{\eta\lambda_F}{2} \Big)^K \Big]^t \mathbb{E}\big[\|h_0 - h(\omega_0)\|^2\big] + \frac{12 D_J^2}{\lambda_F^2} \Big( 1 - \frac{\eta\lambda_F}{2} \Big)^K$$

$$+ \frac{32}{\lambda_F^2} \big( C_\psi^4 M^2 \sigma_W^{2T_z} + c_4 \sigma_W^{2T'} + 16 C_\psi^2 \zeta_{\text{approx}}^{\text{critic}} \big)$$

$$+ \frac{512 C_\psi^2}{\lambda_F^2} \Big( \sigma_W^{2T_c'}\beta^2 c_2 + 2M \Big[ c_3 \Big( 1 - \frac{\lambda_B}{8}\beta \Big)^{T_c} + \frac{c_1}{N_c} \Big] \Big) + \frac{256 c_9}{N\eta\lambda_F^3}$$

$$\overset{(ii)}{\leq} \Big[ 3 \Big( 1 - \frac{\eta\lambda_F}{2} \Big)^K \Big]^t \Big[ \Big( 1 - \frac{\eta\lambda_F}{2} \Big)^K \mathbb{E}\big[\|h_{-1} - h(\omega_0)\|^2\big]$$

$$+ \frac{16}{\lambda_F^2} \big( C_\psi^4 M^2 \sigma_W^{2T_z} + c_4 \sigma_W^{2T'} + 16 C_\psi^2 \zeta_{\text{approx}}^{\text{critic}} \big)$$

$$+ \frac{256 C_\psi^2}{\lambda_F^2} \Big( \sigma_W^{2T_c'}\beta^2 c_2 + 2M \Big[ c_3 \Big( 1 - \frac{\lambda_B}{8}\beta \Big)^{T_c} + \frac{c_1}{N_c} \Big] \Big) + \frac{128 c_9}{N\eta\lambda_F^3} \Big]$$

$$+ \frac{12 D_J^2}{\lambda_F^2} \Big( 1 - \frac{\eta\lambda_F}{2} \Big)^K + \frac{32}{\lambda_F^2} \big( C_\psi^4 M^2 \sigma_W^{2T_z} + c_4 \sigma_W^{2T'} + 16 C_\psi^2 \zeta_{\text{approx}}^{\text{critic}} \big)$$

$$+ \frac{512 C_\psi^2}{\lambda_F^2} \Big( \sigma_W^{2T_c'} \beta^2 c_2 + 2M \Big[ c_3 \Big( 1 - \frac{\lambda_B}{8} \beta \Big)^{T_c} + \frac{c_1}{N_c} \Big] \Big) + \frac{256 c_9}{N \eta \lambda_F^3}$$

$$\overset{(iii)}{\leq} 2 \Big( 1 - \frac{\eta \lambda_F}{2} \Big)^K \Big( \|h_{-1}\|^2 + \frac{D_J^2}{\lambda_F^2} \Big)$$

$$+ \frac{12 D_J^2}{\lambda_F^2} \Big( 1 - \frac{\eta \lambda_F}{2} \Big)^K + \frac{48}{\lambda_F^2} \big( C_\psi^4 M^2 \sigma_W^{2T_z} + c_4 \sigma_W^{2T'} + 16 C_\psi^2 \zeta_{\text{approx}}^{\text{critic}} \big)$$

$$+ \frac{768 C_\psi^2}{\lambda_F^2} \Big( \sigma_W^{2T_c'} \beta^2 c_2 + 2M \Big[ c_3 \Big( 1 - \frac{\lambda_B}{8} \beta \Big)^{T_c} + \frac{c_1}{N_c} \Big] \Big)$$

$$+ \frac{384 c_9}{\eta \lambda_F^3} \frac{\eta \lambda_F^5 (1 - \rho)(1 - \eta \lambda_F/2)^{(K-1)/2}}{2304 C_\psi^4 (\kappa + 1 - \rho)}$$

$$\overset{(iv)}{\leq} c_{10} \Big( 1 - \frac{\eta \lambda_F}{2} \Big)^{(K-1)/2} + c_{11} \sigma_W^{2T_z} + c_{12} \sigma_W^{2T'} + c_{13} \beta^2 \sigma_W^{2T_c'}$$

$$+ c_{14} \Big( 1 - \frac{\lambda_B}{8} \beta \Big)^{T_c} + \frac{c_{15}}{N_c} + c_{16} \zeta_{\text{approx}}^{\text{critic}}$$

where (i) uses the inequality that $3(1 - \eta \lambda_F/2)^K \leq 1$ implied by the condition that $K \geq \frac{\ln 3}{\ln[(1 - \eta \lambda_F/2)^{-1}]}$, (ii) uses eq. (62) with $t = 0$, (iii) uses the condition that $N \geq \frac{2304 C_\psi^4 (\kappa + 1 - \rho)}{\eta \lambda_F^5 (1 - \rho)(1 - \eta \lambda_F/2)^{(K-1)/2}}$ as well as the inequalities that $\|h_{-1} - h(\omega_0)\|^2 \leq 2\|h_{-1}\|^2 + 2\|h(\omega_0)\|^2 \overset{*}{\leq} 2\|h_{-1}\|^2 + 2D_J^2 \lambda_F^{-2}$ (* uses the item 4 of this Lemma) and that $3(1 - \eta \lambda_F/2)^K \leq 1$, (iv) denotes that $c_{10} := 2\|h_{-1}\|^2 + \frac{14 D_J^2}{\lambda_F^2} + \frac{c_9 \lambda_F^2}{C_\psi^4}$, $c_{11} := \frac{48 C_\psi^4 M^2}{\lambda_F^2}$, $c_{12} := \frac{48 c_4}{\lambda_F^2}$, $c_{13} := \frac{768 c_2 C_\psi^2}{\lambda_F^2}$, $c_{14} := \frac{1536 M c_3 C_\psi^2}{\lambda_F^2}$, $c_{15} := \frac{1536 M c_1 C_\psi^2}{\lambda_F^2}$, $c_{16} := \frac{768 C_\psi^2}{\lambda_F^2}$. This proves the item 8 of this Lemma. $\square$

# E  EXPERIMENT SETUP AND ADDITIONAL RESULTS

In this section, we present all the details of experiment setup and additional experiment results.

## E.1  EXPERIMENT SETUP

We simulate a fully decentralized ring network with 6 fully decentralized agents, using communication matrix with diagonal entries 0.4 and off-diagonal entries 0.3. The shared state space contains 5 states and each agent can take 2 actions. We adopt the softmax policy $\pi_\omega(a|s) \propto e^{\omega_{s,a}}$. The entries of the transition kernel and the reward functions are independently generated from the standard Gaussian distribution (with proper normalization of the absolute value for the transition kernel). We use the rows of a 5-dimensional identity matrix as state features. We set the discount factor $\gamma = 0.95$.

We implement and compare four decentralized AC-type algorithms in this multi-agent MDP: our decentralized AC in Algorithm 1, our decentralized NAC in Algorithm 3, an existing decentralized AC algorithm (Algorithm 2 of (70)) that uses a linear model to parameterize the agents' averaged reward $\overline{R}(s, a, s') = \sum_i \lambda_i f_i(s, a, s')$ (we name it DAC-RP1 for decentralized AC with reward parameterization) [5], and our proposed modified version of DAC-RP1 to incorporate minibatch, which we refer to as DAC-RP100 with batch size $N = 100$. For our Algorithm 1, we choose $T = 500$, $T_c = 50$, $T_c' = 10$, $N_c = 10$, $T' = T_z = 5$, $\beta = 0.5$, $\{\sigma_m\}_{m=1}^6 = 0.1$, and consider batch size choices $N = 100, 500, 2000$. Algorithm 3 uses the same hyperparameters as those of Algorithm 1 except that $T = 2000$ in Algorithm 3. We select $\alpha = 10, 50, 200$ for Algorithm 1 with $N = 100, 500, 2000$ respectively, and $T_z = 5$, $\alpha = 0.1, 0.5, 2$, $\eta = 0.04, 0.2, 0.8$, $K = 50, 100, 200$, $N_k \equiv 2, 5, 10$ for Algorithm 3 with $N = 100, 500, 2000$, respectively. For DAC-RP1 that was originally designed for discount factor $\gamma = 1$, we slightly adjust it to fit our setting where $0 < \gamma < 1$ [6].

---

[5] The original algorithm in (70) uses the parameterization $\overline{R}(s, a) = \sum_i \lambda_i f_i(s, a)$, and we extend to our setting where the rewards also depend on the next state $s'$.

[6] (70) defined the Q-function $Q_\theta(s, a) = \mathbb{E}\big[\overline{r}_{t+1} - J(\theta)\big]$ for policy parameter $\theta$ and used the temporal differences $\delta_t^i = r_{t+1}^i - \mu_t^i + V_{t+1}(v_t^i) - V_t(v_t^i)$ and $\widetilde{\delta}_t^i = \overline{R}_t(\lambda_t^i) - \mu_t^i + V_{t+1}(v_t^i) - V_t(v_t^i)$ for critic

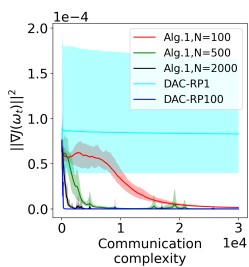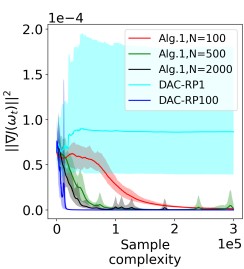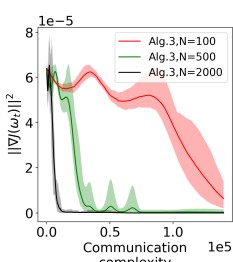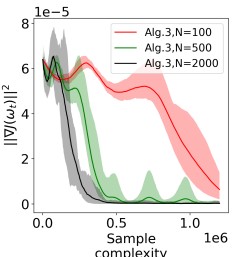

Figure 2: Comparison of $\|\nabla J(\omega_t)\|^2$ among decentralized AC-type algorithms for ring network .

For this adjusted DAC-RP1, we select diminishing stepsizes $\beta_\theta = 2(t+1)^{-0.9}$, $\beta_v = 5(t+1)^{-0.8}$ as recommended in (70) and use the rows of a 1600-dimensional identity matrix as the reward features $\{f_i(s, a, s') : s, s' \in \mathcal{S}, a \in \mathcal{A}\}$ $(i = 1, 2, \ldots, 1600)$ to fully express $\overline{R}(s, a, s')$ over all the $5 \times 2^6 \times 5 = 1600$ triplets $(s, a, s')$. DAC-RP100 has batchsizes 100 and 10 for actor and critic updates respectively, and selects constant stepsizes $\beta_v = 0.5$, $\beta_\theta = 10$. This setting is similar to Algorithm 1 with $N = 100$ to inspect the reason of performance difference between Algorithm 1 and DAC-RP1. All the algorithms are repeated 10 times using initial state 0 and the same initial actor parameter $\omega_0$ generated from standard Gaussian distribution.

### E.2 GRADIENT NORM CONVERGENCE RESULTS IN RING NETWORK

Figure 2 plots $\|\nabla J(\omega_t)\|^2$ v.s. communication complexity $(t(T_c + T_c + T') = 65t$, $t(T_c + T_c + T' + T_z) = 70t$ and $2t$ for Algorithms 1 & 3, and both DAC-RP algorithms, respectively)[7] and sample complexity $(t(T_c N_c + N)$, $2t$ and $110t$ for both of our AC-type algorithms, DAC-RP1 and DAC-RP100, respectively).[8] For each curve, its upper and lower envelopes denote the 95% and 5% percentiles of the 10 repetitions, respectively.

Similar to the result of accumulative reward $J(\omega_t)$ shown in Figure 1, it can be seen from Figure 2 that the communication and sample efficiency of both our decentralized AC and NAC algorithms improve with larger batchsize due to reduced gradient variance, which matches our understanding in Theorems 1 & 2. Our decentralized AC and NAC algorithms significantly outperform DAC-RP1 which has batchsize 1. Using mini-batch, DAC-RP100 outperforms a lot than DAC-RP1, and converges to critical points earlier than Algorithm 1. However, it can be seen from Figure 1 that such early convergence turns out to have much lower $J(\omega_t)$ than Algorithm 1 with $N = 100$ and $N_c = 10$. Such a performance gap is caused by two reasons: (i) Both DAC-RP1 and DAC-RP100 suffer from an inaccurate parameterized estimation of the averaged reward, and the mean relative estimation errors of both DAC-RP1 and DAC-RP100 are over 100% [9]. In contrast, our noisy averaged reward estimation achieves a mean relative error in the range of $10^{-5} \sim 10^{-4}$. [10] ; (ii) Both DAC-RP1

---

update and actor update respectively. To fit $0 < \gamma < 1$, we use $\delta_t^i = r_{t+1}^i + \gamma V_{t+1}(v_t^i) - V_t(v_t^i)$ and $\widetilde{\delta}_t^i = \overline{R}_t(\lambda_t^i) + \gamma V_{t+1}(v_t^i) - V_t(v_t^i)$ where $\mu_t^i \approx J(\theta_t)$ is removed since $Q_\theta(s, a) = \mathbb{E}(\overline{r}_{t+1})$. In addition, we used two different chains generated from transition kernels $\mathcal{P}$, $\mathcal{P}_\xi$ respectively for critic update and actor update as in our Algorithm 1.

[7]Each update of our decentralized AC uses $T_c + T_c'$ and $T'$ communication rounds for synchronizing critic model and rewards, respectively. Each update of our decentralized NAC uses $T_c + T_c'$, $T'$, $T_z'$ communication rounds for synchronizing critic model, rewards and scalar $z$, respectively. Each update of both DAC-RP1 and DAC-RP100 uses 1 communication round for synchronizing $v$ and $\lambda$ respectively.

[8]DAC-RP1 uses 1 sample for actor and critic updates respectively. DAC-RP100 uses 100 and 10 samples for actor and critic updates respectively.

[9]The relative reward estimation error at the $t$-th iteration of both DAC-RP1 and DAC-RP100 is defined as $A/B$ where $A = \frac{1}{M|\mathcal{S}|^2|\mathcal{A}|} \sum_{m=1}^{M} \sum_{s,s' \in \mathcal{S}} \sum_{a \in \mathcal{A}} [\overline{R}(s, a, s') - \sum_i \lambda_i^{(m)} f_i(s, a, s')]^2$ and $B = \frac{1}{|\mathcal{S}|^2|\mathcal{A}|} \sum_{s,s' \in \mathcal{S}} \sum_{a \in \mathcal{A}} \overline{R}(s, a, s')^2$.

[10]At the $t$-th iteration of Algorithms 1 & 3, we focus on $\overline{r}_t^{(m)} = \frac{1}{N} \sum_{i=tN}^{(t+1)N-1} \overline{R}_i^{(m)}$ as the estimation of the batch-averaged reward $\overline{r}_t = \frac{1}{N} \sum_{i=tN}^{(t+1)N-1} \overline{R}_i$ since its estimation error affects the accuracy of the policy gradient (4). The relative estimation error is defined as $\frac{1}{M\overline{r}_t^2} \sum_{m=1}^{M} (\overline{r}_t^{(m)} - \overline{r}_t)^2$.

and DAC-RP100 apply only a single TD update per-round, and hence suffers from a larger mean TD learning error (about $2\%$ and $1\%$ for DAC-RP1 and DAC-RP100, respectively), whereas our algorithms perform multiple TD learning updates per-round and achieve a smaller mean relative error (about $0.3\%$ and $0.07\%$ for our decentralized AC and NAC respectively) [11]. All these relative errors are averaged over iterations.

### E.3 ADDITIONAL EXPERIMENTS IN FULLY CONNECTED NETWORK

To investigate the effect of network topology on the performance of our algorithms, we also conduct the above experiments on a fully connected network with 6 fully decentralized agents, using communication matrix with diagonal entries 0.4 and all the other entries 0.12. The MDP environment and all the hyperparameters are the same as the above experiments for ring network. Figures 3 & 4 plot the learning curves of the optimality gap $J^* - J(\omega_t)$ and $\|\nabla J(\omega_t)\|^2$ respectively for fully connected network. To make comparison, we plot $J^* - J(\omega_t)$ and $\|\nabla J(\omega_t)\|$ in Figures 5 & 2 respectively for the above experiments with ring network. It can be seen by comparing these figures that network topology does not much affect the performance of these algorithms, so the conclusions for ring network that we summarized right before this subsection also holds for fully connected network.

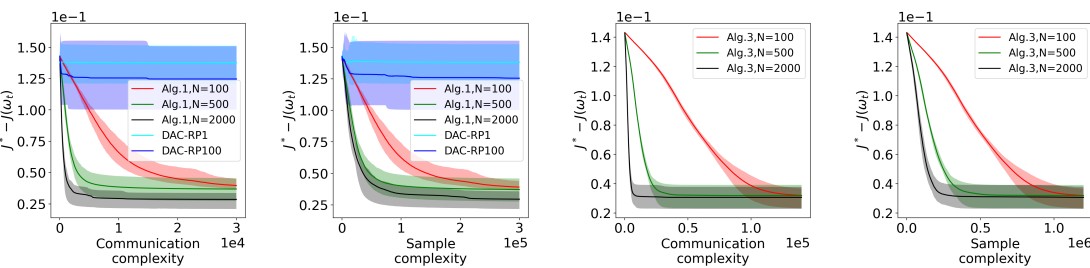

Figure 3: Comparison of optimality gap $J(\omega^*) - J(\omega_t)$ among decentralized AC-type algorithms in fully connected network.

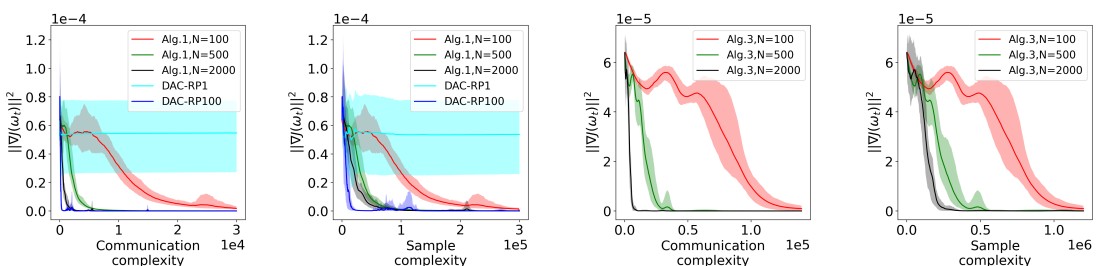

Figure 4: Comparison of $\|\nabla J(\omega_t)\|^2$ among decentralized AC-type algorithms in fully connected network.

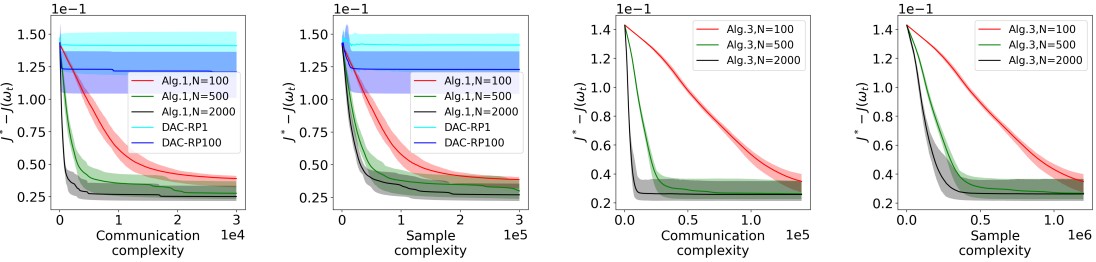

Figure 5: Comparison of optimality gap $J(\omega^*) - J(\omega_t)$ among decentralized AC-type algorithms in ring network.

---

[11]The TD error at the $t$-th iteration is defined as $\frac{1}{M\|\theta^*_{\omega_t}\|^2} \sum_{m=1}^{M} \|\theta_t^{(m)} - \theta^*_{\omega_t}\|^2$.

### E.4 Two-agent Cliff Navigation

In this subsection, we test our algorithms in solving a two-agent Cliff Navigation problem (39) in a grid-world environment. This problem is adapted from its single-agent version (see Example 6.6 of (48)). As illustrated in Figure 6, two agents start from the starting point "S" on a $3 \times 4$ grid and aim to reach the destination "D". Here, global state is defined as the joint location of the two agents, and there are in total $(3 \times 4)^2 = 144$ global states. In most states, an agent can choose to move up, down, left or right by one step and receives $-1$ reward. However, once an agent falls into the cliff "X", it will return to the starting point "S" and receives $-100$ reward. When an agent reaches "D", it will always stay at "D", and receives $0$ reward if the other agent also reaches/stays at "D", or receives $-0.5$ reward otherwise. If an agent is not at "X" or "D" and selects a direction that points outside the grid, then it stays in the previous location and receives $-1$ reward. The optimal path for both agents is the red path shown in Figure 6, which has the minimum accumulative reward $J^* = -0.1855$ under the discount factor $\gamma = 0.95$.

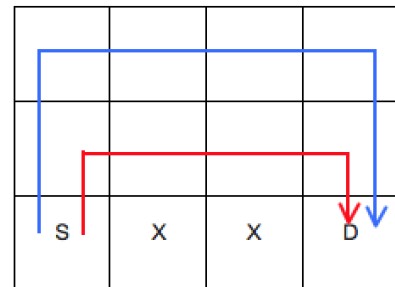

Figure 6: Two-agent cliff navigation. ("S", "X", "D" denote starting point, cliff and destination respectively. The optimal path is shown in red.)

For our Algorithm 1, we choose $T = 500$, $T_c = 50$, $T'_c = 10$, $N_c = 10$, $T' = T_z = 5$, $\beta = 0.5$, $\{\sigma_m\}_{m=1}^6 = 0.1$, and consider batch size choices $N = 100, 500, 2000$. Our Algorithm 3 uses the same hyperparameters as those of Algorithm 1 except that we choose $T = 2000$. We select $\alpha = 1, 5, 20$ for Algorithm 1 with $N = 100, 500, 2000$ respectively, and $T_z = 5$, $\alpha = 0.002, 0.01, 0.04$, $\eta = 0.002, 0.01, 0.04$, $K = 50, 100, 200$, $N_k \equiv 2, 5, 10$ for Algorithm 3 with $N = 100, 500, 2000$, respectively. For DAC-RP1, we select $T = 10000$, $\beta_v = 10(t+1)^{-0.6}$ and $\beta_\theta = 5(t+1)^{-0.6}$. For DAC-RP100, we use $T = 2000$ and batchsizes $100$ and $10$ for actor and critic updates respectively, and selects constant stepsizes $\beta_v = 0.5$, $\beta_\theta = 1$. This setting is similar to Algorithm 1 with $N = 100$ to inspect performance difference between Algorithm 1 and DAC-RP1.

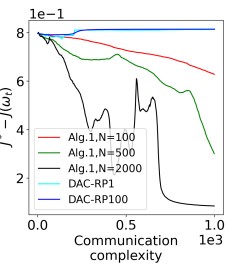 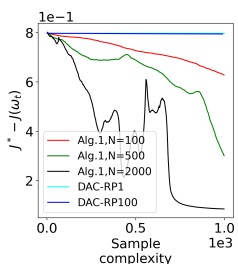 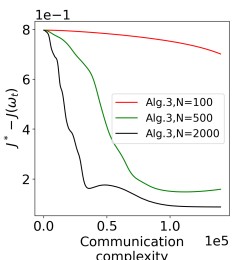 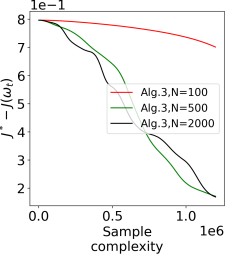

Figure 7: Comparison of optimality gap $J(\omega^*) - J(\omega_t)$ among decentralized AC-type algorithms on cliff navigation.

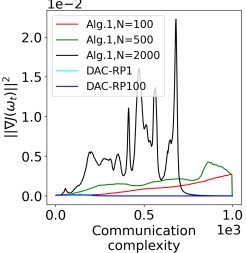 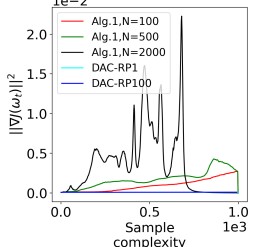 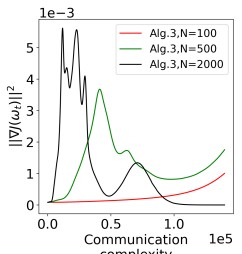 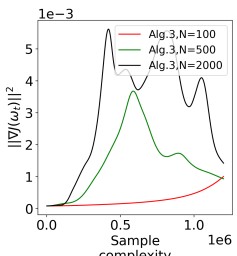

Figure 8: Comparison of optimality gap $J(\omega^*) - J(\omega_t)$ among decentralized AC-type algorithms on cliff navigation.

We plot $J^* - J(\omega_t)$ and $\|\nabla J(\omega_t)\|$ in Figures 7 & 8 respectively. It can be seen from these figures that both our Algorithm 1 & Algorithm 3 significantly reduce the function value gap $J^* - J(\omega_t)$, and their convergence is faster with a larger batchsize. In contrast, the function value gaps of DAC-RP1 and DAC-RP100 do not decrease sufficiently and converge to a high value. In particular, since DAC-RP100 achieves a larger function value gap than our Algorithm 1 with $N = 100$ while their hyperparameter choices are similar, we attribute this performance gap to the inaccurate average reward estimation and TD error, as we analyzed in Appendix E.2.

## F    CONSTANT SCALARS

The following global constants are frequently used.

$M$: The number of agents.

$\gamma$: Discount rate.

$R_{\max}$: The reward bound such that $0 \leq R^{(m)}(s, a, s') \leq R_{\max}$ for any $s, s' \in \mathcal{S}$ and $a \in \mathcal{A}$ (Assumption 3). Hence, $0 \leq \overline{R}^{(m)}(s, a, s'), R_i^{(m)}, \overline{R}_i \leq R_{\max}$.

$\sigma_W \in [0, 1)$: The second largest singular value of $W$.

$\omega^* := \max_\omega J(\omega)$ denotes the optimal policy parameter.

The following constants are defined in Lemma D.3.

$C_B := 1 + \gamma$.

$C_b := R_{\max}$.

$\lambda_\phi := \lambda_{\min}\big(\mathbb{E}_{s \sim \mu_\omega}[\phi(s)\phi(s)^\top]\big) > 0$ satisfies Assumption 4.

$\lambda_B := 2(1 - \gamma)\lambda_\phi > 0$. (Assumption 4 implies that $\lambda_\phi > 0$.)

$R_\theta := \frac{2C_b}{\lambda_B}$.

The policy-related norm bounds and Lipschitz parameters are defined as follows.

$C_\psi, L_\psi, L_\pi > 0$ defined in Assumption 2: For all $s \in \mathcal{S}$, $a \in \mathcal{A}$ and $\omega, \widetilde{\omega}$, $\|\psi_\omega(a|s)\| \leq C_\psi$, $\|\psi_{\widetilde{\omega}}(a|s) - \psi_\omega(a|s)\| \leq L_\psi\|\widetilde{\omega} - \omega\|$ and $d_{\mathrm{TV}}\big(\pi_{\widetilde{\omega}}(\cdot|s), \pi_\omega(\cdot|s)\big) \leq L_\pi\|\widetilde{\omega} - \omega\|$.

$L_\nu := L_\pi[1 + \log_\rho(\kappa^{-1}) + (1 - \rho)^{-1}]$.

$L_Q := \frac{2R_{\max}L_\nu}{1-\gamma}$.

$L_J := R_{\max}(4L_\nu + L_\psi)/(1 - \gamma)$.

$D_J := \frac{C_\psi R_{\max}}{1-\gamma}$.

$L_F := 2C_\psi(L_\pi C_\psi + L_\nu C_\psi + L_\psi)$.

$L_h := 2\lambda_F^{-1}(D_J \lambda_F^{-1} L_F + L_J)$ where $\lambda_F := \inf_{\omega \in \Omega} \lambda_{\min}[F(\omega)] > 0$ ($\lambda_{\min}$ denotes the minimum eigenvalue) which satisfies Assumption 6.

The following constants are defined to simplify the notations in the proof.

$c_1 := \frac{1920\big(C_B^2 R_\theta^2 + C_b^2\big)[1 + (\kappa - 1)\rho]}{(1-\rho)\lambda_B^2}$.

$c_2 := 2\big(\frac{2MC_b}{1-\sigma_W}\big)^2$.

$c_3 := 2\big(\|\theta_{-1}\|^2 + R_\theta^2\big)$ where $\theta_{-1}$ is the initial parameter of decentralized TD (Algorithm 2).

$c_4 := 4MC_\psi^2 R_{\max}^2$.

$c_5 := 16c_2 C_\psi^2$.

$c_6 := 32Mc_3 C_\psi^2$.

$c_7 := 16C_\psi^2 R_{\max}^2 \overline{\sigma}^2 + \frac{36C_\psi^2 (R_{\max}+2R_\theta)^2 (\kappa+1-\rho)}{1-\rho}$.

$c_8 := 32Mc_1 C_\psi^2$.

$c_9 := \frac{18C_\psi^4 D_J^2 (\kappa+1-\rho)}{\lambda_F^2 (1-\rho)} + c_7$.

$c_{10} := 2\|h_{-1}\|^2 + \frac{14D_J^2}{\lambda_F^2} + \frac{c_9 \lambda_F^2}{C_\psi^4}$ where $h_{-1}$ is the initial natural gradient of Algorithm 3.

$c_{11} := \frac{48C_\psi^4 M^2}{\lambda_F^2}$.

$c_{12} := \frac{48c_4}{\lambda_F^2}$.

$c_{13} := \frac{768c_2 C_\psi^2}{\lambda_F^2}$.

$c_{14} := \frac{1536Mc_3 C_\psi^2}{\lambda_F^2}$.

$c_{15} := \frac{1536Mc_1 C_\psi^2}{\lambda_F^2}$.

$c_{16} := \frac{768C_\psi^2}{\lambda_F^2}$.

$c_{17} := \mathbb{E}_{s \sim \nu_{\omega^*}} \left[ \text{KL}\big(\pi_{\omega^*}(\cdot|s) || \pi_0(\cdot|s)\big) \right] + \frac{4L_\psi C_\psi^2 R_{\max}}{\lambda_F^2}$.

$c_{18} := C_\psi \sqrt{c_{10}} + c_{10} L_\psi \left(1 + \frac{4C_\psi^4}{\lambda_F^2}\right)$.

$c_{19} := C_\psi \sqrt{c_{11}} + c_{11} L_\psi \left(1 + \frac{4C_\psi^4}{\lambda_F^2}\right)$.

$c_{20} := C_\psi \sqrt{c_{12}} + c_{12} L_\psi \left(1 + \frac{4C_\psi^4}{\lambda_F^2}\right)$.

$c_{21} := C_\psi \sqrt{c_{13}} + c_{13} L_\psi \left(1 + \frac{4C_\psi^4}{\lambda_F^2}\right)$.

$c_{22} := C_\psi \sqrt{c_{14}} + c_{14} L_\psi \left(1 + \frac{4C_\psi^4}{\lambda_F^2}\right)$.

$c_{23} := C_\psi \sqrt{c_{15}} + c_{15} L_\psi \left(1 + \frac{4C_\psi^4}{\lambda_F^2}\right)$.

$c_{24} := c_{16} L_\psi \left(1 + \frac{4C_\psi^4}{\lambda_F^2}\right)$.

