# OpenReview forum: "Sample and Communication-Efficient Decentralized Actor-Critic Algorithms with Finite-Time Analysis"
_ICLR.cc/2022/Conference — ICLR 2022 Submitted_

### Official Review · Reviewer_ZhdW · 2021-10-29

**Correctness:** 4
**Technical Novelty And Significance:** 4
**Empirical Novelty And Significance:** 3
**Recommendation:** 8
**Confidence:** 4

**Main Review:**

The paper is professionally written and well-organized. This is the first work that derives sample complexity bounds for decentralized AC-type algorithms, and the obtained sample complexities match the existing ones of centralized AC and NAC. Moreover, the corresponding communication complexities are orderwise lower.

Pros:

The developed algorithms are fully decentralized. Especially for the decentralized NAC algorithm, the agents use decentralized SGD to compute the inverse of the fisher information matrix via solving a quadratic problem. This is proven to be computation-efficient.

The algorithm designs of AC and NAC use mini-batch sampling to substantially reduce the communication complexity (by an order of $\epsilon$). Also, the algorithm updates avoid sharing local actions. To accurately estimate the average reward while preserving privacy of the local rewards, the authors propose to add large Gaussian noises to each of the local rewards, and the total noise variance is shown to be substantially suppressed after averaging. This is because the mini-batch samples and local averaging help suppress the noise variance.

For the decentralized NAC, decentralized SGD is used to solve for the natural gradient update. This decentralized SGD uses Markovian samples and an adapted batch size scheme. Such a design helps optimize the error bound and achieve the desired complexity.

Questions:

The decentralized NAC algorithm involves multi-level loops, is there a more effective way to implement it in practice?

For the same algorithm, is it possible that we only update the inner mini-batch SGD for a single iteration and still achieve the desired complexity results?

In the critic update, the algorithm performs multiple steps of local model averaging after the main critic updates. What is the purpose of introduing these additional consensus steps? Is it necessary?

**Summary Of The Paper:**

In this work, the authors propose two decentralized policy gradient-type algorithms for multi-agent reinforcement learning, namely, a decentralized actor-critic algorithm and a decentralized natural actor-critic algorithm. The stochastic updates of both algorithms preserve the agents' privacy information, including their local actions and local rewards. The authors analyze the finite-time convergence rate of both algorithms under Markovian sampling and linear function approximation, and prove that both algorithms achieve the state-of-the-art sample complexities and improved communication complexities.

**Summary Of The Review:**

The paper is professionally written and well-organized. This is the first work that derives sample complexity bounds for decentralized AC-type algorithms, and the obtained sample complexities match the existing ones of centralized AC and NAC. Moreover, the corresponding communication complexities are orderwise lower. I think the developed decentralized AC and NAC algorithms are good complements to the existing multi-agent RL literature, and hence I recommend accepting this work.

---

> ### Author Response · Authors · 2021-11-16
> **Response to Review**
>
> Thank you very much for reviewing our manuscript and providing valuable feedback. Below is a point-to-point response. We have submitted a revised version with all revisions marked in `red`. Please let us know if further clarifications are needed.
>
> **Q1:** Is there a more effective way to implement multi-loop decentralized NAC algorithm in practice?
>
> **A:** Great question. In fact, in our decentralized NAC, the inner SGD iterations for solving the quadratic problem can be carried out by the agents when they observe the mini-batch samples on the fly. So this inner loop actually does not cost much extra time. After these SGD updates, the agents perform the local actor updates.
>
> **Q2:** Is it possible to obtain the desired complexity results of decentralized NAC with only one SGD iteration?
>
> **A:** Great question. We believe this scheme will not work because a single inner iteration will lead to an inaccurate estimate of the stochastic natural policy gradient and cannot be compensated by running more outer iterations. In addition, the upper bound in Theorem 2 cannot be arbitrarily small when $K=1$.
>
> **Q3:** Why do we need additional consensus steps in decentralized TD algorithm? Is it necessary?
>
> **A:** Great question. The additional consensus steps is necessary because the decentralized critic updates cannot achieve an arbitrary small consensus error. The additional local model averaging steps are introduced to help reach the desired global consensus, and it turns out that only a small number of local averaging steps $T_c'= \mathcal{O}(\ln\epsilon^{-1})$ are required.

---

> > ### Comment · Reviewer_ZhdW · 2021-12-01
> > **Thanks for your responses**
> >
> > I want to thank the authors for making great effort to improve the draft. In the revision, they included more clarifications in response to the review questions and added more related work and comparisons with the existing literature. More experiments under different topology and RL environments are also provided.
> >
> > I am satisfied with the authors' response and the revised draft and will keep my score for acceptance. In general, this is a solid and comphrehensive theoretical work on decentralized actor-critic and natural actor-critic algorithms, whose sample complexity and communication complexity have not been established in prior works. I found the assumptions and settings are reasonable and follow the standard ones that are adopted in the existing literature on finite-sample analysis of (decentralized) RL. In particular, they propose the first decentralized NAC algorithm, and the analysis introduces a special Markovian decentralized SGD algorithm with adaptive mini-batch sampling to efficiently compute the Fisher matrix-vector product. I generally think this is a complete work and is a good complement to the existing literature on decentralized policy optimization.

---

### Official Review · Reviewer_8K9V · 2021-10-31

**Correctness:** 3
**Technical Novelty And Significance:** 3
**Empirical Novelty And Significance:** 2
**Recommendation:** 5
**Confidence:** 4

**Main Review:**

1.	In the abstract, the authors mentioned that the proposed algorithms were developed with linear function approximation. That way, how to apply the theory to scenarios where typically deep nets are utilized?
2.	In Table 1, though this paper is focused on AC type of algorithms, other decentralized algorithms, such as Q learning, value-prop, etc., should be discussed here for comparison as well.
3.	When the authors mentioned Markovian sample, would it refer to a trajectory or just one transition? If it was a trajectory, then it might be enough in practice. As a mini-batch in practice could be time-consuming for updates, though it did really reduce the variance.
4.	In the proposed algorithm, each agent shares Gaussian-corrupted local reward. Then the differential privacy should be involved in this work? Also, why not sharing the parameters of actor or critic, which has been popular in decentralized optimization domain?
5.	The authors only put related works together without any discussion on their advantages and disadvantages, which renders the weak motivation. I would suggest the authors discuss these in detail to provide better comparison.
6.	In page 5, the authors mentioned that they could choose a large batch size to suppress the overall estimation error to the desired level. A large batch will definitely affect the accuracy and computational time in practice based on existing conclusions from the study in deep learning using large batch. Additionally, the authors cannot just always increase the size of mini-batch.
7.	In Algorithm 2, extra local model averaging steps are taken. Would this cause additional communication overhead in practice?
8.	In Theorem 1, $\epsilon$ is required to satisfy a condition. Similarly in Theorem 2. Why do you need such conditions? Can you not just define it as an arbitrarily small constant as in other works? If $\epsilon$ is too big, the optimality would be poor. For the finite-time analysis, can you not just make the step size and other constants satisfy some conditions w.r.t. time $T$ and derive the complexity? That way, typically $T$ needs to be sufficiently larger than one constant.
9.	In the discussion for Theorem 1, the author mentioned that “different from the existing decentralized TD (39) that uses a single sample to update the value function parameters per communication round, we use a mini-batch of $N_c$ samples instead to significantly reduce the communication complexity of decentralized TD learning while maintaining the same level of sample complexity.” Basically existing decentralized TD algorithm uses a single sample to update the value function parameters per communication round, the authors used a mini-batch of samples to reduce the communication complexity. Why is this true? Basically, when a mini-batch of samples is used, the variance can be used. But if the authors keep the same communication frequency, the number of communication rounds remains the same. Or after the authors use the mini-batch of samples, they do multiple updates before next communication round? When the authors mentioned the same level of sample complexity, I think they should give more detail on this. As in their case, I am not very clear about what sample complexity refers to. Is it the number of data samples required for update? Or the number of gradient oracles required? Or else?
10.	In the decentralized NAC, it requires to solve a quadratic programming. Would it increase the computational complexity when solving a quadratic programming problem? The approximation to the inverse Fisher information matrix looks complicated. In practice, what is computational time cost? Also, the Fisher information matrix has to be uniformly positive definite for the analysis. Such an assumption seems to be too strong.
11.	After Theorem 2, one statement is that “the result proves that the function value optimality gap asymptotically converges to the order…”. Finite-time typically corresponds to the non-asymptotic convergence, correct? If the authors require asymptotic behavior, why are those different parameters required to satisfy conditions w.r.t. $\epsilon$? Or did I misunderstand?
12.	At the bottom of Page 8, the authors mentioned that a sample budget was set in the analysis. Why is such a total sample budget constraint there? When $k$ is very large, $N_k$ will be huge. How would you implement this in practice?
13.	Though the experimental results make sense and sort of validate the algorithms, they are not convincing. First, the benchmark environment is too simple, only a few states with a couple actions. We have no idea how the proposed methods would perform under some popular benchmarks, such as gridworld, which has many more states. Second, more algorithms are required for comparison. It is understandable that this paper focuses significantly on the AC type of algorithms, while other existing algorithms have been proposed, and one or two of them should be used for the comparison. Third, for decentralized algorithms, different topologies are critical to be shown for demonstrate the effectiveness.
14.	In this paper, the authors used partial policy gradient. Why not directly using local policy gradient? The authors also mentioned that “…to obtain an accurate estimation, the network needs to have a sufficiently large number of agents, which can be unrealistic in practice.” But typically in practice, this can happen and remains a challenge. How to scale decentralized algorithms to large networks?
15.	What is $(\{s_t\})_t$ in page 5?

****************************************
I really appreciate the detailed responses from the authors and they did address some confusions regarding the theoretical analysis. More experimental results also provided better validation for the proposed algorithms. However, there are still some confusions in the revised draft. For example, in Table 1, why is the sample complexity for paper (54) much much larger than others? If $\epsilon = 0.001$, then the complexity is completely intractable. I checked the paper and don't think the complexity given in the Table 1 for paper (54) is correct. Additionally, after carefully reviewing comments from other reviewers, I decided to keep the score and believe with some more work, the paper will be technically solid and sound and a nice publication.

**Summary Of The Paper:**

This paper developed two sample and communication efficient decentralized actor-critic algorithms for multi-agent reinforcement learning. Specifically, the authors proposed decentralized AC and natural AC algorithms that can be private and efficient for different agents to learn. They added noise the local rewards of agents, which were shared among different agents. Moreover, the authors adopted mini-batch updates for the actors and critics in these two algorithms. Theoretically, the authors showed that both algorithms were able to achieve the state-of-the-art sample complexities. To validate the proposed algorithms, they used a simple environment to show the superiority over the existing decentralized AC algorithm. Overall, the investigated topic in this paper is absolutely interesting. The sample and communication complexities of decentralized MARL algorithms remain active research areas. While the mathematical analysis in this work looks good, the numerical results to me don't look very convincing. Additionally, there are quite a few points I have in mind regarding the proposed algorithm and the relevant analysis.

**Summary Of The Review:**

Overall, I think the current form of the paper still requires substantial work to make it technically solid and sound. In particular, the numerical results are not very convincing, and there are numerous confusions in the analysis, which have weakened the theoretical contributions. I hope my comments can help out.

---

> ### Author Response · Authors · 2021-11-16
> **Response to Review: Part I**
>
> Thank you very much for reviewing our manuscript and providing valuable feedback. Below is a point-to-point response. We have submitted a revised version with all revisions marked in `red`. Please let us know if further clarifications are needed.
>
> **Q1:** Algorithms were developed with linear function approximation. How can the theorems apply to deep nets?
>
> **A:** Good question. We use linear model to approximate the value function so that the analysis of the decentralized TD part is technically tractable. Note that this part of the analysis is not the main body of this study. Such a linear model can be technically generalized to neural network models by following some existing works, e.g.,
>
> [1] Xu et.al. A Finite-Time Analysis of Q-Learning with Neural Network Function Approximation, ICML, 2019.
>
> [2] Cai et.al. Neural temporal difference learning converges to global optima, NeurIPS, 2019.
>
> In these works, the authors use a deep ReLU network to model the value functions, and the idea is to approximate this nonlinear model by a class of its locally linearized models.
> We also note that, in our formulation, the policies are allowed to be parameterized by nonlinear models.
>
>
> **Q2:** Should compare with other MARL algorithms like Q-learning.
>
> **A:** Thanks for the great suggestion. In the related work section of the revised version, we have cited and discussed more papers on decentralized policy gradient-type and decentralized value-based algorithms, as also suggested by other reviewers. We note that the complexity results of these algorithms are generally not comparable to our sample complexity due to two reasons: First, the convergence criterion of value-based algorithms (e.g., Q-learning) typically involves the Bellman error and the optimal Q function, while the convergence criterion of our AC-type algorithms is defined in terms of the expected accumulated award. These convergence criteria are not comparable. Second, the sample complexity of policy gradient-type algorithms is usually expressed in terms of number of episodes (by assuming access to exact policy gradient), while the sample complexity of our AC-type algorithms is defined in terms of number Markovian transition samples. These quantities cannot be compared directly. In addition to these, we also added more details to elaborate the comparison between our work and the existing art on AC-type algorithms, and included more sample complexity results on AC and NAC in Table 1.
>
> **Q3:** What does Markovian samples mean? Large batchsize could be time-consuming.
>
> **A:**  To clarify, our setting assumes that the agents keep interacting with the environment and observing a trajectory of MDP transition samples, which are correlated through a Markov train and we refer to them as Markovian samples.
>
> We want to clarify that the mini-batch update can be implemented in an efficient and practical way. To explain, note that the existing decentralized AC algorithm takes one local update per MDP transition sample. As a comparison, our algorithms take one local update per mini-batch of MDP transition samples. This means that while the agents follow their current policies and keep observing MDP transition samples over time, our algorithms require less frequent updates (and hence less frequent communication as well). Also, the local mini-batch policy gradient in eq.(4) can be computed in an accumulative way by the agent when observing the mini-batch of transition samples on the fly. Hence, there is no need to store and load all these samples at once. We have added these clarifications to the Remark on Page 5.
>
> **Q4:** Why not use differential privacy? Why not share actor or critic parameters?
>
> **A:** We are really sorry that our tone of writing confuses the reviewer. To clarify, in this work, we call a decentralized RL algorithm 'private' if the agents do not need to share their local information (e.g., local actions, local policies, etc.). Our work focuses on characterizing the communication and sample complexities of the proposed decentralized RL algorithms, and quantifying the privacy guarantee does not fall into the scope of this study. In the revision, we have replaced all the keywords 'privacy/private' by the precise statement 'without sharing local actions and local policies'.
>
> The agents generally do not want to share their actor parameters as that would reveal their own policies. On the other hand, the critic parameters are shared among the agents in the decentralized TD learning phase, see eq. (5). This is reasonable because the agents share a global state and therefore they collaborate together to learn a shared global critic model.
>
>
> **Q5:** We should compare with the papers in related works.
>
> **A:** Thanks for the suggestion. In the related work section of the revision, we added more details to elaborate the comparison between our work and the existing art on AC-type algorithms, and included more sample complexity results on AC and NAC in Table 1.

---

> > ### Author Response · Authors · 2021-11-16
> > **Response to Review: Part II**
> >
> > **Q6:** The authors cannot just always increase the size of mini-batch, it is time-consuming.
> >
> > **A:** Based on our theorem, we will pre-determine a desired accuracy level $\epsilon$, and then the batch size is fixed to be a constant $O(\epsilon^{-1})$. As we explained in the response to Q3, the mini-batch update can be computed on the fly when the agents keep querying samples from the MDP and is not time-consuming. Also, our experiments demonstrate that using a large batch size can reduce the variance in the training and achieve a comparable or even higher accumulated reward compare with using a small batch size.
> >
> >
> >
> > **Q7:** Does extra local model averaging steps in Algorithm 2 cause additional communication overhead in practice?
> >
> > **A:** Great question. In our experiments, in a sparsely-connected ring network, we found that 8-10 extra local model averaging steps suffices to achieve a global consensus. Hence, less extra model averaging steps are needed for denser networks.
> >
> >
> > **Q8:** Why do we need the constraints on $\epsilon$ in theorems? Why not express hyperparameters including stepsizes in terms of the number of iterations $T$.
> >
> > **A:** Good question. Those constraints on $\epsilon$ are naturally induced by the critic and actor approximation errors involved in the convergence bounds in Theorems 1 and 2. That is, the target accuracy $\epsilon$ cannot be smaller than these fundamental approximation errors. On the other hand, if both approximation errors vanish, for example, when the dimension of features equals the number of states and the parameterized policy space is sufficiently expressive, then we can achieve arbitrarily small target accuracy. We have clarified these points in our revision, see the remark on Page 6.
> >
> >
> > Regarding the hyperparameters, they can be equivalently expressed in terms of iteration $T$. Specifically, we just need to substitute $\epsilon=\mathcal{O}(T^{-1})$ (because $T=\mathcal{O}(\epsilon^{-1})$) into all the expressions of the choices of hyperparameters.
> >
> >
> > **Q9:** What does sample complexity mean? Why can mini batch reduce the communication complexity while maintaining the sample complexity in decentralized TD?
> >
> > **A:** Good question. Following the convention of the existing works on RL theory, we define sample complexity as the total number of MDP transition samples required for achieving an $\epsilon$ convergence error, as we explained in the caption of Table I. We have also clarified this definition before Theorem 1.
> >
> > We are sorry for the confusion of the sentence '...while maintaining the sample complexity in decentralized TD', we actually mean 'while maintaining the state-of-the-art sample complexity of decentralized TD'. Therefore, while our algorithms achieve the state-of-the-art sample complexity, they require less number of iterations as well as communication rounds thanks to the use of mini-batch updates. Please see our response to Q3 for more explanation. We have updated this in the revision. Please let us know if this answers your question.
> >
> >
> > **Q10:** The quadratic programming in decentralized NAC requires approximating inverse Fisher information matrix. Does it increase computational complexity? In practice, what is computational time cost? The assumption of uniformly positive definite Fisher information matrix looks strong.
> >
> > **A:** Great question. We want to first clarify that the quadratic programming problem is developed to approximate only the inverse Fisher matrix-vector product $F(\omega_t)^{-1}\nabla J(\omega_t)$, which is simpler than approximating the whole inverse Fisher matrix. We have replaced 'inverse Fisher information matrix' by 'inverse matrix-vector product' in the revision.
> >
> > We note that the SGD steps for solving this problem are cheap, as the stochastic gradient in eq. (8) only needs to compute one scalar-vector product per sample. In our numerical experiment, solving such a quadratic problem using SGD takes about 0.16 seconds of each update of the algorithm, which is much less than the total run time 1.23 seconds of each update of the algorithm.
> >
> > Regarding the uniformly positive definite assumption on the Fisher information matrix, it is widely adopted in the existing analysis of AC-type algorithms (see [1-3] listed below) and is satisfied for many policies, e.g., Gaussian-type policies (see appendix B.2 of [1]).
> >
> > [1] Y. Liu, K. Zhang, T. Basar, and W. Yin. An improved analysis of (variance-reduced) policy gradient and natural policy gradient methods. In Proc. Advances in Neural Information Processing Systems (NeurIPS), volume 33, pages 7624–7636, 2020.
> >
> > [2] T. Xu, Z. Yang, Z. Wang, and Y. Liang. Doubly robust off-policy actor-critic: Convergence and optimality. ArXiv:2102.11866, 2021.
> >
> > [3] L. Yang, Q. Zheng, and G. Pan. Sample complexity of policy gradient finding second-order stationary points. ArXiv:2012.01491, 2020.

---

> > > ### Author Response · Authors · 2021-11-16
> > > **Response to Review: Part III**
> > >
> > > **Q11:** Do we need asymptotic behavior?
> > >
> > > **A:** Sorry for the confusion. Our convergence bounds are non-asymptotic (finite-time) since the convergence bounds are expressed in terms of number of iterations $T$. 'Asymptotically converge' simply means 'converge'. We have removed `asymptotically' in the revision.
> > >
> > > **Q12:** How to set total sample size budget $N$ in practice as $N_k$ grows with $k$?
> > >
> > > **A:** Great question. We first note that although $N_k$ scales with $k$ exponentially, it does not grow to infinity as $k = 0,1,...,K-1$ is bounded (see eq.(9)) by a small number $K$. To explain, in every NAC update we solve the quadratic problem (7) by using the queried  mini-batch of $N$ samples. Therefore the total sample budget $N$ for solving the quadratic problem is essentially the batch size of the algorithm. Then, when apply SGD to solve this quadratic problem, we distributed these $N$ samples unevenly to the SGD iterations $k=0,1,...,K-1$ following the proposed exponential scaling scheme, which turns out to tighten the convergence bounds in the technical proof. In our experiments, we find that the scheduling of $N_k$ can be flexible, and a constant $N_k$ works well.
> > >
> > >
> > > **Q13:** The experiments should compare with more kinds of algorithms under popular benchmark environments and different network topologies.
> > >
> > > **A:** We thank the reviewer for the great suggestion. In our revision, we have added more experiments on a fully connected network typology in Appendix E.3. We are now conducting additional experiments in the OpenAI Gym environment and adding more baseline algorithms such as the policy gradient. We will keep the reviewer updated during the rebuttal period.
> > >
> > > **Q14:** Why use 'partial policy gradient'? Why say 'The network needs to have a sufficiently large number of agents, which can be unrealistic in practice.'?
> > >
> > > **A:** In our revision, we have changed 'partial policy gradient' to 'local policy gradient' to denote an agent's own policy gradient. We have also changed 'which can be unrealistic in practice.' to 'which does not always hold in practice.'. This means that by using mini-batch updates, we do not necessarily need to rely on a large number of agents to reduce the reward estimation variance.
> > >
> > >
> > > **Q15:** The meaning of {s_t}_t in page 5.
> > >
> > > **A:** {s_t}_t means {s_t}_{$t \in\mathbb{N}$}. We have updated this in the revision.

---

> ### Author Response · Authors · 2021-11-30
> **Follow-up response**
>
> Thank you very much for the update and we would like to provide a follow-up response.
>
> Regarding the complexity of the paper (54) given in Table 1, we are very sorry that we made a mistake in the calculation during the rebuttal period. We have updated the complexity to be $O(\epsilon^{-18})$ for their AC algorithm and $O(\epsilon^{-14})$ for their NAC algorithm. We attached our calculation at the end of this response for your reference and will clarify their settings in the revision. While their sample complexity results seem worse than other works', they actually considered a very different over-parameterized neural network model and their convergence bounds turn out to have a heavy dependence on the number of neurons, which explodes the overall complexity.
>
> The reviewer also comments that '...with some more work, the paper will be technically solid and sound.' We believe that our theoretical analysis is rigorous and sound, and all our theoretical settings are also widely adopted in the existing literature on theory of multi-agent RL and actor-critic algorithms. We have also clarified the questions raised by the other reviewers. We would greatly appreciate it if the reviewer can provide more precise comments about the soundness of our technical proof and help us understand the weakness, as this is very important from a theorist point of view and our entire team has spent tremendous effort on understanding, discussing and responding to the review comments over the past two weeks.
>
>
> **Calculation of the sample complexity for (54)**
>
> In Theorem 4.7 of (54) about centralized AC, $\rho_i$ corresponds to our $\nabla J(\pi_{\theta_i})$ when $\theta_i\in\mathcal{B}$. They proved the convergence bound
> $$\min_i \mathbb{E}||\rho_i||^2\le \frac{8}{\sqrt{T}}\mathbb{E}[J(\pi_{\theta_{T+1}})-J(\pi_{\theta_1})]+\frac{8\sigma_{\xi}^2}{B}+\kappa\mathcal{O}(R^{5/2}m^{-1/4}T^{1/2}+R^{9/4}m^{-1/8}T^{1/2}),
> $$
> where $m$ denotes the number of neurons.
> To achieve an $\epsilon$ convergence error,
> we need to choose $T=\mathcal{O}(\epsilon^{-2})$ iterations in Algorithm 1. In each iteration, the actor update steps (3.7) and (3.9) require $B=\mathcal{O}(\epsilon^{-1})$ samples, and Algorithm 2 requires $T_{TD}=\mathcal{O}(m)=\mathcal{O}\big(\max(T^2\epsilon^{-4}, T^4\epsilon^{-8})\big)=\mathcal{O}(\epsilon^{-16})$ critic updates with one sample per update. Hence, the overall sample complexity is $T(T_{TD}+B)=\mathcal{O}(\epsilon^{-18})$.
>
> In Theorem A.4 of (54) about centralized NAC, they proved a convergence bound
> $$\min_{i\in[T]}\mathbb{E}[J(\pi^*)-J(\pi_{\theta_i})] \le \mathcal{O}(T^{-1/2}) +\mathcal{O}(B^{-1/4}) + \mathcal{O}(T^{-1/2}m^{-1/4}) \sum_{i\in [T]}\tau_i + \mathcal{O}(m^{-1/8})$$
> $$= \mathcal{O}(T^{-1/2}) +\mathcal{O}(B^{-1/4}) + \mathcal{O}(TT_{TD}^{-1/4}) + \mathcal{O}(T_{TD}^{-1/8}),
> $$
> where we used their hyper-parameter choices $\tau_{i+1}=i\eta=i\mathcal{O}(T^{-1/2})$ and $m=\mathcal{O}(T_{TD})$. To achieve an $\epsilon$ convergence error, we need to choose $T=\mathcal{O}(\epsilon^{-2})$, $B=\mathcal{O}(\epsilon^{-4})$, $T_{TD}=\mathcal{O}(\epsilon^{-12})$. Consequently, the overall sample complexity is $T(T_{TD}+B)=\mathcal{O}(\epsilon^{-14})$.

---

### Official Review · Reviewer_hsKp · 2021-11-01

**Correctness:** 2
**Technical Novelty And Significance:** 2
**Empirical Novelty And Significance:** 3
**Recommendation:** 5
**Confidence:** 5

**Main Review:**

Strengths:
1.	The proposed two decentralized multi-agent actor-critic algorithms achieve the state-of-the-art sample and communication complexities. The proposed algorithms also preserve privacy which seems important in practice.
2.	Experiments have demonstrated that the proposed algorithms achieve lower sample and communication complexities than the existing decentralized AC algorithms.

Weakness:
1.	The paper does not put this work into context. In general, efforts over the past year/two have been made to develop different policy-based and value-based algorithms for MARL, some of which have also considered communication efficiency. Authors have missed many works in this field. Even for related works mentioned in this paper, the authors did not discuss their advantages and disadvantages, which makes hard to evaluate the real contributions of this work.

2.	In the beginning of this paper, the authors mention privacy as the main motivation and the selling point of this paper. However, there is not explicitly definition of privacy nor the theoretical guarantee on privacy. In addition, there is a growing line of literature of differential privacy in RL, which has not been discussed in this work.

3.	The proposed algorithm seems impossible to be implemented in practice. There are two reasons behind this. One is that the number of samples N and N_c need to grow with accuracy epsilon^-1, which will eventually blow up as the number of iterations increase. More importantly, different from existing decentralized AC algorithm that uses one transition per update (e.g., [63]), the proposed algorithms require N or N_c transitions involving global states and actions per local update. Note that different from the centralized AC and PG, in the decentralized MARL setting, unless the agent has a accurate simulator to simulate the entire MARL systems, it is not possible to obtain more than one transition per update in MARL.

4.	The optimality gap in Theorem 2 can be very large. Specifically, there are three terms that depend on the unknown actor and critic approximation errors. Are they necessary to be in the bound? Any lower bound to justify this? Is it possible to quantify this?

5.	Numerical experiments seem very limited and not thorough enough considering the number of works published on a daily basis in this field. Specifically, only randomly generated MDPs have been tested and only one baseline [63] has been compared. Furthermore, in Theorem 2, the global convergence has been proved. It would be great to validate the optimality gap in this simulation.



**Summary Of The Paper:**

This paper introduces a decentralized multi-agent actor-critic algorithm. Different from existing algorithms, the new algorithm enjoys both the privacy and the communication efficiency.


**Summary Of The Review:**

Two new decentralized AC algorithms have been developed with guaranteed  sample and communication complexities. However, some claims are not justified and the implementation is not realistic.

---

> ### Author Response · Authors · 2021-11-16
> **Response to Review**
>
> Thank you very much for reviewing our manuscript and providing valuable feedback. Below is a point-to-point response. We have submitted a revised version with all revisions marked in `red`. Please let us know if further clarifications are needed.
>
> **Q1:** This paper does not compare with papers in related works and those about other kinds of MARL algorithms.
>
> **A:** Thanks for the great suggestion. In the related work section of the revised version, we added more details to elaborate the comparison between our work and the existing art on AC-type algorithms. We also included more sample complexity results on AC and NAC in Table 1. Moreover, we have cited and discussed more papers on decentralized policy gradient-type and decentralized value-based algorithms in the related work, as also suggested by other reviewers. We note that the complexity results of these algorithms are generally not comparable to our sample complexity due to two reasons: First, the convergence criterion of value-based algorithms (e.g., Q-learning) typically involves the Bellman error and the optimal Q function, while the convergence criterion of our AC-type algorithms is defined in terms of the expected accumulated award. These convergence criteria are not comparable. Second, the sample complexity of policy gradient-type algorithms is usually expressed in terms of number of episodes (by assuming access to exact policy gradient), while the sample complexity of our AC-type algorithms is defined in terms of number Markovian transition samples. These quantities cannot be compared directly.
>
> **Q2:** There is no explicit definition or theoretical guarantee of privacy. Differential privacy is not mentioned.
>
> **A:** We are really sorry that our tone of writing confuses the reviewers. To clarify, in this work, we call a decentralized RL algorithm 'private' if the agents do not need to share their local information (e.g., local actions, local policies, etc.). To avoid confusion, we have replaced all the keywords 'privacy/private' by the precise statement `without sharing local actions and local policies'. Our work focuses on characterizing the communication and sample complexities of the proposed decentralized RL algorithms, and quantifying the privacy guarantee does not fall into the scope of this study.
>
>
> **Q3:** The algorithm is impossible to implement for two reasons: 1) batch sizes need to grow with accuracy $\epsilon^{-1}$, which will eventually blow up; 2) algorithms require a mini-batch of MDP transitions per local update.
>
> **A:** Our algorithms can be implemented in a practical way and we clarify as follows. Regarding the reviewer's first comment, we can pre-determine a desired accuracy level $\epsilon$, and then the batch size is fixed to be $O(\epsilon^{-1})$.
>
> Regarding the reviewer's second comment, we want to clarify that our setting assumes that the agents keep interacting with the environment and observing MDP transition samples, they do not have access to any simulator of the MDP. This is a standard setting that is widely adopted in the convergence analysis of many RL algorithms.
> Note that the existing decentralized AC algorithm takes one local update per MDP transition sample. As a comparison, our algorithms take one local update per mini-batch of MDP transition samples. This means that while the agents follow their current policies and keep observing MDP transition samples over time, our algorithms require less frequent updates (and hence less frequent communication as well). Also, we note that the local mini-batch policy gradient in eq.(4) can be computed in an accumulative way by the agent when observing the mini-batch of transition samples on the fly. There is no need to store all these samples. We have added these clarifications to the Remark on Page 5.

---

> > ### Author Response · Authors · 2021-11-16
> > **continue response**
> >
> > **Q4:** Are actor and critic approximation errors necessary? Are there lower bound to justify this?
> >
> > **A:** Yes, the actor and critic approximation errors
> > are widely adopted in the analysis of AC-type algorithms, e.g., see the references [1-4] listed below. Note that the critic approximation error corresponds to the discrepancy between the true value function and the linear function class, while the actor approximation error represents the approximation error induced by the insufficient expressive power of the parameterized policy class. Both are fundamental quantities. We also note that these approximation errors vanish when the value function model class is compete, for example, when the dimension of features equals the number of states and the parameterized policy space is sufficiently expressive.
> >
> > [1] L. Wang, Q. Cai, Z. Yang, and Z. Wang. Neural policy gradient methods: Global optimality and rates of convergence. ArXiv:1909.01150, 2019.
> >
> > [2] Y. F. Wu, W. ZHANG, P. Xu, and Q. Gu. A finite-time analysis of two time-scale actor-critic methods. In Proc. Advances in Neural Information Processing Systems (NeurIPS), volume 33, pages 17617–17628, 2020.
> >
> > [3] T. Xu, Z. Wang, and Y. Liang. Improving sample complexity bounds for (natural) actor-critic algorithms. In Proc. Advances in Neural Information Processing Systems (NeurIPS), volume 33, 2020.
> >
> > [4] T. Xu, Z. Yang, Z. Wang, and Y. Liang. Doubly robust off-policy actor-critic: Convergence and optimality. ArXiv:2102.11866, 2021.
> >
> > **Q5:** Only randomly generated MDPs have been tested and only baseline [63] has been compared. It would be great to validate the optimality gap in this simulation.
> >
> > **A:** We thank the reviewer for the great suggestion. In our revision, we have changed the optimality measure to the optimality gap in all experiments. Please see the Figure 5 in Appendix E.3. We have also added more experiments on a fully connected network typology in Appendix E.3. We are now conducting additional experiments in the OpenAI Gym environment and adding more baseline algorithms such as the policy gradient. We will keep the reviewer updated during the rebuttal period.

---

### Official Review · Reviewer_RWPJ · 2021-11-02

**Correctness:** 3
**Technical Novelty And Significance:** 2
**Empirical Novelty And Significance:** 2
**Recommendation:** 6
**Confidence:** 3

**Main Review:**

Strengths:

1. The sample complexity of the proposed decentralized algorithms is comparable to their single-agent state-of-the-art results.

2. The decentralized NAC algorithm is probably the first one in the literature.

3. Using a separate distributed averaging procedure to estimate team average policy gradient is an interesting idea to avoid sharing/broadcasting local actions.

Weaknesses:

1. Although the idea of using a separate distributed averaging procedure to estimate team average policy gradient is interesting, it requires additional coordination among the networked agents. The value of T' needs to be carefully designed and shared among all the agents.

2. The paper claims privacy-preserving. But it reads like a immediate, lower-level privacy concern. Does it involve any qualitative support?

3. The paper claims the technical challenges due to mini-batch updates. But the description at the end of Section 4 is still vague. What is the difference/novelty compared with the techniques used in "Finite-sample analysis for decentralized batch multi-agent reinforcement learning with networked agents" by Zhang et al. in TAC 2021?

4. The proposed idea involves additional T' iterations for distributed averaging in each time step. Is it still fair to compare with the single-agent case without such inner iterations?

5. Why in Table 1 (63) is marked as both private and non-private?



**Summary Of The Paper:**

The paper proposes two decentralized AC algorithms, one for actor-critic and the other for natural actor-critic, which are sample and communication efficient. The paper also provides their finite time performance, which is comparable with their corresponding single-agent state-of-the-art results in sample complexity.

**Summary Of The Review:**

The paper proposes a new decentralized AC framework which combines existing decentralized techniques including distributed averaging and TD learning. The idea is interesting but not sophisticated. The NAC part is probably the first decentralized NAC algorithm. Combining existing tools may lead to new analysis challenges, which needs to be further clarified. Thus I think the paper is on the board line.

---

> ### Author Response · Authors · 2021-11-16
> **Response to Review**
>
> Thank you very much for reviewing our manuscript and providing valuable feedback. Below is a point-to-point response. We have submitted a revised version with all revisions marked in `red`. Please let us know if further clarifications are needed.
>
> **Q1:** Privacy concern looks immediate and lower-level. Does it involve any qualitative support?
>
> **A:** We are really sorry that our tone of writing confuses the reviewers. To clarify, in this work, we call a decentralized RL algorithm 'private' if the agents do not need to share their local information (e.g., local actions, local policies, etc.). To avoid confusion, we have replaced all the keywords 'privacy/private' by the precise statement 'without sharing local actions and local policies'. Our work focuses on characterizing the communication and sample complexities of the proposed decentralized RL algorithms, and quantifying the privacy guarantee does not fall into the scope of this study.
>
> **Q2:** Technical challenges in Section 4 are vague. What is the difference/novelty compared with the techniques used in Zhang, et al. in TAC 2021?
>
> **A:** Thanks for the great suggestion. We have provided more elaborations on the technical challenges in Section 4 in the revision.
>
> Regarding the comparison to Zhang et al., their algorithm for cooperative MARL is based on a very different type of algorithm named fitted-Q iteration. Moreover, they use mini-batch in a different way from us. Specifically, in each iteration, they need to solve a finite-sum minimization problem that consists of a batch of fitted-Q loss, but their algorithm solves this problem using stochastic updates with batch size 1 (not mini-batch). As a comparison, our algorithms use mini-batch stochastic updates.
>
> **Q3:** Is it fair to compare with the single-agent case without inner iterations of local averaging?
>
> **A:** We compare the sample complexities of all the algorithms, i.e., the total number of samples required for achieving an $\epsilon$ convergence error. We think this metric is fair and fundamental to all algorithms. For the single-agent algorithms, they can be viewed as centralized algorithms and hence do not need any local averaging for achieving consensus.
>
>
> **Q4:** Why [1] is marked as both private and non-private in Table 1?
>
> **A:** Great question. Algorithm 1 of [1] (listed below) needs to share local actions while Algorithm 2 of [1] does not, as described in the second paragraph of the introduction. We have added a footnote to clarify this in our revision. Thank you for pointing this out.
>
> [1] K. Zhang, Z. Yang, H. Liu, T. Zhang, and T. Basar. Fully decentralized multi-agent reinforcement learning with networked agents. In Proc. International Conference on Machine Learning (ICML), pages 5872–5881, 2018.
>
> As suggested by other reviewers, we have also included more experiments on a different network topology in Appendix E.3.

---

### Author Response · Authors · 2021-11-22
**Update on experiments**

Dear Reviewers,

We have updated our revision and included more experiments on a gridworld benchmark in the Appendix E.4. These additional experiment results also demonstrate the advantage of our decentralized AC and NAC algorithms. Thank you all again for the great effort in reviewing this work and for helping us improve the quality.

---

### Decision · Program_Chairs · 2022-01-20

**Decision:**

Reject

**Comment:**

While authors have updated the draft to address reviewers' concerns, some parts are still not clear enough and the presentation needs further improvements. I encourage authors to revise the draft accordingly and resubmit in the future venues.